# LONG-HORIZON MODEL-BASED OFFLINE REINFORCE-MENT LEARNING WITHOUT CONSERVATISM

## ABSTRACT

Popular offline reinforcement learning (RL) methods rely on *conservatism*, either by penalizing out-of-dataset actions or by restricting rollout horizons. In this work, we question the universality of this principle and instead revisit a complementary one: a *Bayesian* perspective. Rather than enforcing conservatism, the Bayesian approach tackles epistemic uncertainty in offline data by modeling a posterior distribution over plausible world models and training a history-dependent agent to maximize expected rewards, enabling test-time generalization. We first illustrate, in a bandit setting, that Bayesianism excels on low-quality datasets where conservatism fails. We then scale this principle to realistic tasks, identifying key design choices, such as layer normalization in the world model and adaptive long-horizon planning, that mitigate compounding error and value overestimation. These yield our practical algorithm, NEUBAY, grounded in the NEUtral BAYesian principle. On D4RL and NeoRL benchmarks, NEUBAY generally matches or surpasses leading conservative algorithms, achieving new state-of-the-art on 7 datasets. Notably, it succeeds with rollout horizons of several hundred steps, contrary to dominant practice. Finally, we characterize datasets by quality and coverage, showing when NEUBAY is preferable to conservative methods. Together, we argue NEUBAY lays the foundation for a new practical direction in offline and model-based RL.

## 1 INTRODUCTION

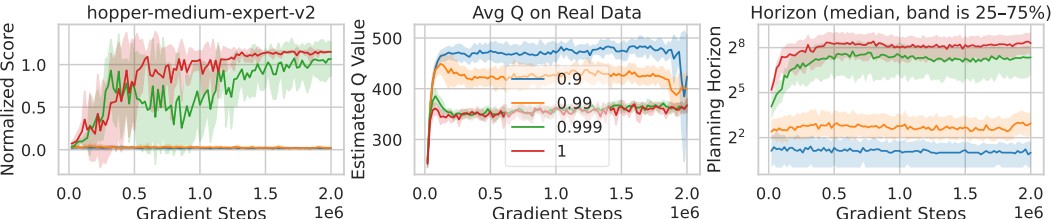

Figure 1: Our algorithm NEUBAY's result on a D4RL dataset. From left to right: normalized score on the real environment, estimated Q-value on the offline dataset, and **rollout horizon statistics** over 100 training rollouts (median with interquartile range). Here we vary the uncertainty quantile $\zeta \in \{0.9, 0.99, 0.999, 1.0\}$ for the rollout truncation threshold, **without using conservatism**.

Reinforcement learning (RL) often assumes direct interaction with the environment, which we refer to as online RL (Sutton & Barto, 2018). While successful in simulation, deploying it in real-world settings such as robotics or healthcare is limited by time-consuming and risky data collection (Dulac-Arnold et al., 2021). A more practical alternative is offline RL (Lange et al., 2012), which learns from pre-collected datasets (e.g., from human demonstrations or prior agents) without further environment access (Levine et al., 2020; Fu et al., 2020; Gulcehre et al., 2020). This decoupling enables safe, scalable training and the potential to outperform the behavior policies that produced the data.

Most offline RL algorithms adopt a conservative principle by penalizing the policy and value function on out-of-dataset state-action pairs (Levine et al., 2020; Prudencio et al., 2023) and using short rollout horizons (Lu et al., 2021). In theory, these algorithms enforce robustness, either strict (Jin et al., 2021; Uehara & Sun, 2022) or soft (Zhang et al., 2024b), over the uncertainty set of possible MDPs consistent with the dataset. The trade-off is clear: conservatism reduces value overestimation and unsafe extrapolation, but can also suppress average-case performance and limit generalization, since policies are discouraged from exploring potentially high-reward but underrepresented actions.

Bayesian RL optimizes average-case performance under epistemic uncertainty (Duff, 2002). Its application to offline RL was pioneered by Ghosh et al. (2022), who formalized the problem as an *epistemic POMDP*, where partial observability arises from limited coverage. This formulation enables test-time generalization through history-dependent Bayes-optimal policies. In this work, we first revisit and extend this Bayesian principle through the lens of *data quality*. Using a two-armed bandit with skewed data, we show that conservative algorithms, with sufficient uncertainty penalty, are guaranteed to commit to the seen arm regardless of test-time conditions. In contrast, Bayesian algorithms can adaptively explore and commit to the better arm at test time, a clear advantage in low-quality datasets.

However, scaling the Bayesian principle to realistic tasks is challenging, as it requires solving an *approximate* epistemic POMDP. We identify *three key challenges*: (1) compounding error in world models, where inaccuracies grow rapidly with horizon (Lambert et al., 2022); (2) value overestimation, since Bayesian RL lacks explicit penalties for out-of-dataset actions and is vulnerable to extrapolation error (Fujimoto et al., 2019); and (3) training agents with long-term memory to enable test-time adaptation (Ni et al., 2023). Each aligns with a major research area: model-based RL, offline RL, and partially observable RL. This helps explain why prior Bayesian-inspired algorithms often *reintroduce* conservatism, through uncertainty penalties, short horizons, or reductions to model-free RL (Ghosh et al., 2022; Chen et al., 2021c; Jeong et al., 2023).

We build a practical algorithm, NEUBAY, to address these challenges. First, we show that layer normalization (Ba et al., 2016), effective in model-free RL (Ball et al., 2023), also helps world models mitigate compounding error. Second, we observe that model-generated rewards are less biased than bootstrapped values, allowing Bayesian RL to amortize overestimation risk through long-horizon planning. To keep such rollouts reliable, we truncate them using epistemic uncertainty as a threshold (Frauenknecht et al., 2024; Zhan et al., 2021), an *alternative* use of uncertainty beyond penalties. Finally, we leverage advances in recurrent RL (Morad et al., 2024; Luo et al., 2024a) to stabilize training and support long-term memory. Together, these components adapt insights from online RL to make Bayesian offline RL practical.

We evaluate NEUBAY on the D4RL (Fu et al., 2020) and NeoRL (Qin et al., 2022) benchmarks, covering 33 datasets. Overall, NEUBAY matches or surpasses leading conservative algorithms and outperforms other Bayesian-inspired methods, establishing new state-of-the-art results on 7 datasets. In realistic tasks, NEUBAY performs best on low-quality datasets and on medium-quality datasets with moderate coverage. Our sensitivity study validates a key insight: **adaptive long-horizon planning** is a primary driver of NEUBAY's success. Whereas dominant model-based RL practice favors short horizons, our Bayesian approach uncovers a new role for long horizons: they **suppress value overestimation**. NEUBAY routinely plans **64-512 steps** (e.g., Fig. 1), while short-horizon variants fail due to severe overestimation. These results position NEUBAY as a practical direction for model-based and offline RL from a Bayesian perspective, with future advances in world modeling to push the limits further.

## 2 BACKGROUND ON OFFLINE RL

In the standard offline RL setting, a static dataset $\mathcal{D}$ is collected by interacting with an MDP $\mathcal{M}^*$, which we refer to as the *true MDP*. Formally, $\mathcal{M}^* = (\mathcal{S}, \mathcal{A}, \gamma, T, f_{\text{term}}, m^*, \rho)$ where $\mathcal{S}$ and $\mathcal{A}$ are state and action spaces, $\gamma \in (0, 1)$ is the discount factor, $T \in \mathbb{N}$ is the maximum episode length, and $f_{\text{term}} : \mathcal{S} \times \mathcal{A} \times \mathcal{S} \to \{0, 1\}$ is the terminal function. These components are assumed to be **known** (Puterman, 2014; Yu et al., 2020). The *joint reward-transition function* is $m^* : \mathcal{S} \times \mathcal{A} \to \Delta([-r_{\max}, r_{\max}] \times \mathcal{S})$, consisting of reward function $R^* : \mathcal{S} \times \mathcal{A} \to \Delta([-r_{\max}, r_{\max}])$ (here, $r_{\max}$ is a positive constant) and dynamics $P^* : \mathcal{S} \times \mathcal{A} \to \Delta(\mathcal{S})$, both **unknown** to the agent and learned in model-based methods. The initial state distribution $\rho \in \Delta(\mathcal{S})$ is also unknown, but we do not model it explicitly since initial states can be directly sampled from $\mathcal{D}$ (Janner et al., 2019).

The static dataset of trajectories[1] $\mathcal{D} = \{\tau^i\}_{i=1}^{\text{num\_traj}}$ is collected by an unknown (possibly) history-dependent behavior policy $\pi_\beta : \mathcal{H}_t \to \Delta(\mathcal{A})$, where $\mathcal{H}_t$ is the space of state-action-reward sequences up to timestep $t$. Define $h_t = (s_{0:t}, a_{0:t-1}, r_{1:t}) \in \mathcal{H}_t$ for $t \geq 1$ with the convention that $h_0 = s_0$. Each trajectory $\tau = (s_0, a_0, r_1, d_1, s_1, a_1, \dots)$ is generated by: $s_0 \sim \rho, a_t \sim \pi_\beta(h_t), (r_{t+1}, s_{t+1}) \sim m^*(s_t, a_t), d_{t+1} = f_{\text{term}}(s_t, a_t, s_{t+1})$. A trajectory ends either when $d_t = 1$ (termination) or when $t = T$ (truncation). We stress this distinction: *termination* implies absorbing states with zero future rewards, whereas *truncation* preserves continuation and thus allows bootstrapping.

---

[1]While offline RL datasets are often expressed as transition tuples, the trajectory format is available in common benchmarks (Fu et al., 2020) and required by sequence-based algorithms (Chen et al., 2021a).

*The ideal objective* in offline RL is to find a possibly history-dependent policy $\pi : \mathcal{H}_t \to \Delta(\mathcal{A})$ that maximizes the expected discounted return under the true MDP $\mathcal{M}^*$:

$$\max_\pi J(\pi, m^*) := \mathbb{E}\left[\sum_{t=0}^\infty \gamma^t r_{t+1} \,\Big|\, s_0 \sim \rho, a_t \sim \pi(h_t), (r_{t+1}, s_{t+1}) \sim m^*(s_t, a_t)\right]. \quad (1)$$

The defining constraint of offline RL is that the agent cannot interact with $m^*$, making it intractable to direct optimize Eq. 1. This leads to the following discussion about epistemic uncertainty on $m^*$.

**Empirical model and epistemic uncertainty.** From the agent's view, knowledge of $m^*$ is well-defined only on the state-action support of the dataset $\mathcal{D}$: $\mathrm{supp}_{\mathcal{S} \times \mathcal{A}}(\mathcal{D}) := \{(s, a) \mid (s, a, r, s') \in \mathcal{D}\}$. Let $\mathfrak{M}_{\mathrm{in}}$ denote a model class whose domain is restricted to $\mathrm{supp}_{\mathcal{S} \times \mathcal{A}}(\mathcal{D})$. The *empirical model* (Fujimoto et al., 2019) is then obtained by maximum likelihood estimation (MLE):

$$m_\mathcal{D} = \mathrm{argmax}_{m \in \mathfrak{M}_{\mathrm{in}}} \mathbb{E}_{(s,a,r,s') \sim \mathcal{D}}[\log m(r, s' \mid s, a)]. \quad (2)$$

Thus, $m_\mathcal{D}$ is uniquely determined in-support by empirical frequencies, but remains *undefined* for $(s, a) \notin \mathrm{supp}_{\mathcal{S} \times \mathcal{A}}(\mathcal{D})$, giving rise to substantial epistemic uncertainty (Gal, 2016). A common way to formalize this uncertainty is through an *uncertainty set* $\mathfrak{M}_\mathcal{D}$, the set of plausible models on $\mathcal{S} \times \mathcal{A}$ that agree with $\mathcal{D}$ on $\mathrm{supp}_{\mathcal{S} \times \mathcal{A}}(\mathcal{D})$ (i.e, close to $m_\mathcal{D}$). Offline policy learning then incorporates $\mathfrak{M}_\mathcal{D}$ into optimization, typically via two paradigms: *conservatism* (or *pessimism*), which optimizes against worst-case models in $\mathfrak{M}_\mathcal{D}$, or *Bayesianism*, which leverages a posterior distribution over $\mathfrak{M}_\mathcal{D}$.

**Conservative principle: robust MDPs.** Conservative RL methods commonly optimize return under the *worst* model of the uncertainty set $\mathfrak{M}_\mathcal{D}$ (Uehara & Sun, 2022):

$$\max_\pi J(\pi; \mathfrak{M}_\mathcal{D}) := \max_\pi \min_{m \in \mathfrak{M}_\mathcal{D}} J(\pi, m). \quad (3)$$

This robust MDP formulation (Wiesemann et al., 2013) covers both model-free (Fujimoto et al., 2019) and model-based algorithms (Rigter et al., 2022; Yu et al., 2020), differing only in the choice of uncertainty set $\mathfrak{M}_\mathcal{D}$ and the degree of robustness (Zhang et al., 2024b). We provide formal connections between conservatism and robustness in Sec. B.

**Bayesian principle: epistemic POMDPs.** Bayesianism instead treats the true model $m^*$ as a random variable (Ghavamzadeh et al., 2015), maintaining a posterior distribution $\mathbb{P}_\mathcal{D}(m)$ over plausible models. Bayesian offline RL (Ghosh et al., 2022; Uehara & Sun, 2022; Fellows et al., 2025) then optimizes the *expected* return under this posterior, known as *ambiguity-neutrality* (Ellsberg, 1961):

$$\max_\pi J(\pi; \mathbb{P}_\mathcal{D}) := \max_\pi \mathbb{E}_{m \sim \mathbb{P}_\mathcal{D}}[J(\pi, m)]. \quad (4)$$

The posterior naturally induces an uncertainty set via its support. If $\mathfrak{M}_\mathcal{D} = \mathrm{supp}(\mathbb{P}_\mathcal{D})$, the Bayesian objective (Eq. 4) is less pessimistic than the conservative one (Eq. 3), since for all policies $\pi$, $J(\pi; \mathbb{P}_\mathcal{D}) \geq J(\pi; \mathfrak{M}_\mathcal{D})$. By averaging over $\mathbb{P}_\mathcal{D}$, the Bayesian approach places more weight on likely models and less on unlikely ones, effectively yielding a *soft-robust* optimization (Derman et al., 2018). Eq. 4 is equivalent to solving a Bayes-adaptive MDP (BAMDP) (Duff, 2002), also known as an epistemic POMDP; see Ghosh et al. (2022) and Sec. C.1 for a connection with POMDPs.

## 3 ILLUSTRATIVE EXAMPLE: WHEN IS BAYESIANISM BETTER?

To illustrate *when* the Bayesian principle is preferred over conservatism, we construct a skewed bandit dataset below. A formal proof comparing the robust return with the Bayes one can be found in Sec. D, which includes the existence of similar offline RL problems with skewed data samples.

**The bandit dataset.** We consider a sequential two-armed bandit with $\mathcal{A} = \{0, 1\}$. Each arm $a \in \mathcal{A}$ yields a Bernoulli reward $r \sim R^*(a) = \mathcal{B}(p_a^*)$, so the true MDP $m^*$ is specified by reward parameters $p^* \in [0, 1] \times [0, 1]$. We fix $p_0^* = 0.5$ in our setup. The optimal policy $\pi^*$ in $m^*$ is deterministic and memoryless, always choosing $\mathrm{argmax}_{a \in \mathcal{A}} p_a^*$. To highlight out-of-support challenges, we construct a dataset that only covers arm 0. Specifically, the dataset $\mathcal{D} = \{(a_{0:T-1}^i, r_{1:T}^i)\}_i$ is collected by a deterministic behavior policy $\pi_\beta(a) = \mathbb{1}(a = 0)$. Thus, for each $t < T = 100$, $a_t = 0$, $r_{t+1} \sim \mathcal{B}(p_{a_t}^*)$, and $\mathcal{D}$ contains no data on $a = 1$.

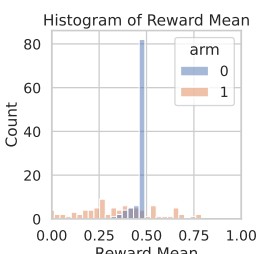

Figure 2: Histogram of estimated reward means $p_0, p_1$ across ensemble members.

This skewed dataset $\mathcal{D}$ leaves the true reward parameter $p_1^*$ for arm 1 *completely* unobserved, inducing substantial epistemic uncertainty on $p_1^*$. Theoretically, under an uninformative prior for Bernoulli rewards, this uncertainty corresponds to the entropy of $\mathrm{Unif}[0, 1]$. In practice, we approximate the posterior by fitting a reward-model ensemble

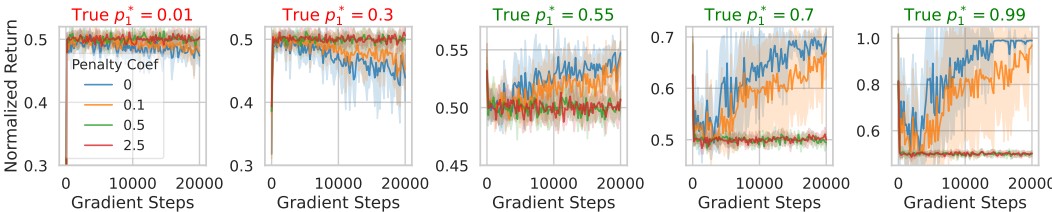

Figure 3: Average return (normalized by $T$) on **test-time bandits** with $p_1^* \in \{0.01, 0.3, 0.55, 0.7, 0.99\}$. Since the observed arm has $p_0^* = 0.5$, cases with $p_1^* < 0.5$ are *worse* and those with $p_1^* > 0.5$ are *better*.

using Gaussian outputs (Chua et al., 2018). As shown in Fig. 2, after pretraining on $\mathcal{D}$, the ensemble predictions concentrate around the observed arm 0 but sharply disagree on the unobserved arm 1, with estimated uncertainty about $10\times$ larger than on arm 0. Next, we test the Bayesian method from Eq. 4, where planning begins at $t = 0$ and truncates at $t = T$. Dynamics are not modeled due to the bandit structure, and there is no concern about compounding error.

**Uncertainty penalty hurts generalization, but Bayesian agents adapt to worse *and* better cases.** We study conservatism by adding an uncertainty penalty to the Bayesian objective. Such a design is common in empirical algorithms (Yu et al., 2020), and can be viewed as risk-sensitive Bayesian RL (Rigter et al., 2021) or a soft robust MDP (Zhang et al., 2024b) (see Sec. B). Since the estimated uncertainty on arm 1 is much larger than arm 0, a sufficiently large penalty coefficient $\lambda$ forces the resulting algorithm to always select arm 0, regardless of the test-time environment.

In contrast, the Bayesian agent ($\lambda = 0$) trains on a posterior $\mathbb{P}_{\mathcal{D}}(p_1)$ that spans both worse and better values relative to $p_0^*$ (see Fig. 2). It adopts a history-dependent policy that explores the unseen arm at test time before committing, similarly to Bayesian meta-RL (Zintgraf et al., 2020). Thus, as shown in Fig. 3, the Bayesian agent's return is slightly worse than conservatism when $p_1^* < p_0^* = 0.5$, because it briefly explores arm 1. When $p_1^* > 0.5$, Bayesianism is a clear win, as it identifies and exploits the better unseen action, whereas heavy conservatism remains stuck on arm 0. We can thus deduce our main insight: *Bayesianism excels with **low-quality datasets** lack of optimal actions, while remaining competitive on **high-quality datasets** by test-time adaptation with a cost.*

## 4 NEUBAY: A PRACTICAL BAYESIAN-PRINCIPLED ALGORITHM

The Bayesian objective (Eq. 4) is conceptually simple and, as seen in the bandit example, clearly outperforms conservatism on low-quality data. Extending it to MDPs raises **three challenges**: compounding model error, value overestimation, and instability in training history-dependent agents. We address these with ensembles for posterior approximation (Sec. 4.1), adaptive long-horizon planning (Sec. 4.2), and stabilized recurrent training (Sec. 4.3). Together these yield our practical algorithm, NEUBAY (Sec. 4.4), whose design choices we analyze in Sec. 5.2.

### 4.1 CONTROLLING COMPOUNDING ERRORS IN EPISTEMIC POMDP MODELING

A prerequisite for optimizing Eq. 4 is to approximate the epistemic POMDP $\mathbb{P}_{\mathcal{D}}$ with a learned posterior $\hat{\mathbb{P}}_{\mathcal{D}}$. Following popular model-based RL methods (Chua et al., 2018; Yu et al., 2020), we model the posterior with deep ensembles (Lakshminarayanan et al., 2017), a simple method that balances predictive accuracy with uncertainty quantification (Ovadia et al., 2019; Gustafsson et al., 2020). Our deep ensemble is a set of neural networks $\mathbf{m}_{\boldsymbol{\theta}} = \{m_{\theta^n}\}_{n=1}^N$ where each model outputs a Gaussian distribution over next state $s'$ and reward $r$: $m_{\theta^n}(s, a) = \mathcal{N}(\mu_{\theta^n}(s, a), \sigma_{\theta^n}(s, a))$, with parameters $\boldsymbol{\theta} = \{\theta^n\}_{n=1}^N$ independently initialized at random. Each $m_{\theta^n}$ is trained on the same dataset using the maximum likelihood (MLE) loss and evaluated by mean squared error (MSE) on a shared validation set (Yu et al., 2020). Deep ensembles provide a practical approximation to Bayesian neural networks, as they capture diverse modes of the Bayesian posterior (Fort et al., 2019; Wilson & Izmailov, 2020). The induced posterior is $\hat{\mathbb{P}}_{\mathcal{D}}(m) = \frac{1}{N} \sum_{n=1}^N \mathbb{1}(m = m_{\theta^n})$ forms a uniform distribution over ensemble members. Epistemic uncertainty can be estimated as the standard deviation of the *mean* predictions (Lakshminarayanan et al., 2017) of the ensemble members: $U_{\boldsymbol{\theta}}(s, a) = \text{std}(\{\mu_{\theta^n}(s, a)\}_{n=1}^N)$. This simple design enables direct comparison with popular methods, although multi-step prediction (Lin et al., 2025) and richer uncertainty quantification (Qiao et al., 2025) may further improve performance.

**Design choice 1: larger ensemble size $N$ for posterior fidelity.** While deep ensembles approximate Bayesian neural networks, their epistemic fidelity relies on member diversity. With horizons extending hundreds of steps, compounding errors make faithful posteriors essential, rendering small ensembles (e.g., $N = 5$ in MBPO (Janner et al., 2019)) inadequate.

**Design choice 2: layer normalization in the world model to control compounding errors.** A fundamental problem with world models is that small prediction errors compound during multi-step planning (Talvitie, 2014). To mitigate this, we apply *layer normalization* (LN) (Ba et al., 2016) to each world model $m_{\theta^n}$. Similar to its role in reducing value overestimation (Ball et al., 2023), LN mitigates compounding model error. To address this, we structure the world model as a delta predictor and apply *layer normalization* (LN) (Ba et al., 2016) to its hidden features. Concretely, the model predicts the next state as $\mathbb{E}[\hat{s}'] = s + \mathbf{W}^\top \mathrm{ReLU}(\mathrm{LN}(\psi(s,a)))$, with features $\psi(s,a) \in \mathbb{R}^k$ and output weights $\mathbf{W} \in \mathbb{R}^{k \times |\mathcal{S}|}$. For LN without affine parameters, under $\ell_2$ norm:

$$\|\mathbb{E}[\hat{s}'] - s\| \le \|\mathbf{W}\|\|\mathrm{ReLU}(\mathrm{LN}(\psi(s,a))\| \le \|\mathbf{W}\|\|\mathrm{LN}(\psi(s,a))\| = \sqrt{k}\|\mathbf{W}\|, \qquad (5)$$

using that fact that $\|\mathrm{LN}(x)\| = \sqrt{k}$ for any $x \in \mathbb{R}^k$. Applying the triangle inequality over an $H$-step imagined trajectory yields a linear compounding bound, $\|\mathbb{E}[\hat{s}_H] - s_0\| \le H\sqrt{k}\|\mathbf{W}\|$. Therefore, by controlling $\|\mathbb{E}[\hat{s}_H]\|$, we can upper bound the compounding error: $\|\mathbb{E}[\hat{s}_H] - s_H\| \le \|\mathbb{E}[\hat{s}_H]\| + \|s_H\|$.

## 4.2 WHY AND HOW DO WE PERFORM LONG-HORIZON PLANNING?

**Where to start planning?** From the Bayesian objective (Eq. 4), it is natural to initiate planning rollouts by sampling initial states $s_0 \sim \rho_{\mathcal{D}}$, where $\rho_{\mathcal{D}}$ is the empirical initial-state distribution. But this underrepresents states appearing *later* in real trajectories, as compounding errors make them harder to reach (Lambert et al., 2022; Lin et al., 2025). While layer normalization substantially mitigates this issue, the errors at later time steps can still grow exponentially (e.g., Fig. 5). To maintain stability, we sample starting states $s_t \sim \mathcal{D}$ from any timestep $t$, following MBPO-style branched rollouts (Janner et al., 2019). Since the environment appears as an epistemic POMDP to the *agent*, we also provide the corresponding history $h_t = (s_{0:t}, a_{0:t-1}, r_{1:t}) \in \mathcal{D}$, which we refer to as the *initial history*.

**Why do we need long-horizon planning?** Ideally, under a correct posterior, the Bayesian principle calls for *full-horizon* planning. In practice, however, compounding errors make full-horizon rollouts unreliable. Even so, *long-horizon* rollouts remain valuable when they are informative for augmenting training data (Young et al., 2023). Beyond augmentation and exploration, we identify a new role for Bayesian offline RL: long-horizon rollouts actively help mitigate **value overestimation**.

To illustrate, starting from a real history $h_t \in \mathcal{D}$, we sample $m_\theta \sim \mathbf{m}_\theta$ and generate a trajectory of length $H$ where $\hat{a}_{t+j} = \pi(\hat{h}_{t+j})$ comes from a deterministic policy, $(\hat{r}_{t+j+1}, \hat{s}_{t+j+1}) \sim m_\theta(\hat{s}_{t+j}, \hat{a}_{t+j})$, and $\hat{h}_t = h_t$. Applying one-step Bellman backups on the Bayesian value function *along this rollout*, i.e., $Q^{\mathrm{Bayes}}(\hat{h}_{t+j}, \hat{a}_{t+j}) \leftarrow \hat{r}_{t+j+1} + \gamma Q^{\mathrm{Bayes}}(\hat{h}_{t+j+1}, \pi(\hat{h}_{t+j+1})), 0 \le j < H$, we obtain an approximate $H$-step backup on the real history:

$$Q^{\mathrm{Bayes}}(h_t, \hat{a}_t) \leftarrow \sum_{j=0}^{H-1} \gamma^j \underbrace{\hat{r}_{t+j+1}}_{\text{lower bias}} + \underbrace{\gamma^H}_{\text{discounted}} \underbrace{Q^{\mathrm{Bayes}}(\hat{h}_{t+H}, \pi(\hat{h}_{t+H}))}_{\text{higher bias}}. \qquad (6)$$

This decomposition highlights the bias trade-off: imagined rewards can be low-bias if the model generalizes, while the bootstrapped term – more susceptible to overestimation (Kumar et al., 2019; Sims et al., 2024) – is exponentially discounted with $H$. We formalize this effect in Sec. C.2, extending the analysis of Sims et al. (2024). While prior works in offline RL mitigate overestimation *through conservatism* (e.g., uncertainty penalties), which hurts generalization, NEUBAY instead absorbs the overestimation risk across long horizons *without conservatism*.

**How should we truncate rollouts?** Given that Bayesian RL favors long-horizon planning, a key question is not simply *when* to truncate, but *where*, since model errors depend on specific $(s, a)$ pairs. A natural criterion is the model's uncertainty estimate $U_\theta(s, a)$, which correlates with prediction error. Prior work has pursued this *conservatively*, combining uncertainty threshold with short horizon caps (Pan et al., 2020; Zhan et al., 2021; Zhang et al., 2023; Frauenknecht et al., 2024). In contrast, we remove restriction on the maximum horizon.

Epistemic uncertainty is highly *non-uniform* even within $\mathcal{D}$: frequently visited $(s, a)$ pairs yield low uncertainty, while rarely seen ones result in high uncertainty. This reflects that epistemic uncertainty concerns the true model $m^*$, *not* the empirical model $m_{\mathcal{D}}$, which can be formalized via standard concentration

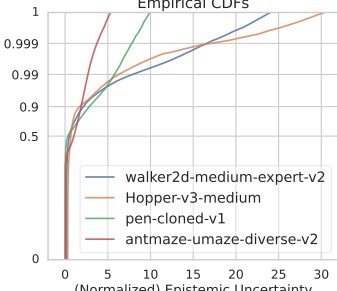

Figure 4: **Empirical CDFs** of epistemic uncertainty $U_\theta$ over $(s, a) \in \mathrm{supp}_{\mathcal{S} \times \mathcal{A}}(\mathcal{D})$, with logit-scaled y-axis. Uncertainties are normalized by the dataset mean, so 1 is the average value.

**Algorithm 1** NEUBAY: Full training loop

**Require:** Offline dataset $\mathcal{D}$, Online buffer $\mathcal{B} \leftarrow \emptyset$.
   World ensemble: $\mathbf{m}_{\boldsymbol{\theta}}$, Recurrent actor: $\pi_{\nu}(a_t \mid h_t)$, Recurrent critic: $Q_{\omega}(h_t, a_t)$.
1: Pretrain $\mathbf{m}_{\boldsymbol{\theta}}$ on $\mathcal{D}$ until convergence
2: **while** gradient steps $\leq$ max gradient steps **do**
3:     **for** $k = 1$ to $K$ rollouts **do**   ▷ Parallelized in practice
4:         Sample $m_{\theta} \sim \mathbf{m}_{\boldsymbol{\theta}}$
5:         Sample initial history $h_t \sim \mathcal{D}$
6:         Append ROLLOUT($h_t, m_{\theta}, \pi_{\nu}$) to $\mathcal{B}$
7:     **end for**
8:     **for** $g = 1$ to $G$ gradient steps **do**
9:         Optimize recurrent off-policy RL loss $L(Q_{\omega}, \pi_{\nu}; \tau, \kappa)$ with $\tau \sim \mathcal{B}$
10:     **end for**
11: **end while**

**Algorithm 2** NEUBAY: Rollout function

**Require:** Terminal function $f_{\text{term}}$, Max episode length $T$, Uncertainty threshold $\mathcal{U}(\zeta)$, ~~Rollout horizon $H$~~
1: **function** ROLLOUT($h_t, m_{\theta}, \pi_{\nu}$)
2:     Set $\hat{h}_t = h_t$ and done to False
3:     **while** done is False **do**
4:         $\hat{a}_t \sim \pi_{\nu}(\hat{h}_t), (\hat{r}_{t+1}, \hat{s}_{t+1}) \sim m_{\theta}(\hat{s}_t, \hat{a}_t)$
5:         $\hat{d}_{t+1} = f_{\text{term}}(\hat{s}_t, \hat{a}_t, \hat{s}_{t+1})$
6:         trunc $= (U_{\boldsymbol{\theta}}(\hat{s}_t, \hat{a}_t) > \mathcal{U}(\zeta))$
7:         done $= \hat{d}_{t+1} \lor$ trunc $\lor (t+1 \geq T)$
8:         $\hat{h}_{t+1} = (\hat{h}_t, \hat{a}_t, \hat{r}_{t+1}, \hat{s}_{t+1}), t \leftarrow t+1$
9:     **end while**
10:     **return** Trajectory $\tau$ (initial history $h_t$ plus imagined rollout)
11: **end function**

bounds in the frequentist sense.[2] As shown in Fig. 4, most empirical CDFs are long-tailed, with dataset-dependent skewness reflecting the underlying visitation distribution. This motivates our use of data-dependent truncation thresholds, described next.

**Design choice 3: uncertainty threshold $\mathcal{U}(\zeta)$ for rollout truncation.** Let $\zeta \in [0, 1]$, we define

$$\mathcal{U}(\zeta) = \mathcal{U}_{\boldsymbol{\theta}, \mathcal{D}}(\zeta) := F_Y^{-1}(\zeta), \quad Y := U_{\boldsymbol{\theta}}(s, a), \quad (s, a) \sim \mathcal{D},$$

where $F_Y^{-1}$ is the quantile function of $Y$. Quantile-based thresholds adapt naturally to different datasets and uncertainty scales. When $\zeta = 1.0$, the threshold equals the maximum uncertainty in $\mathcal{D}$, encouraging the longest possible rollouts. We avoid thresholds beyond the dataset maximum, since this implies uncertainty beyond all real data, a proxy for out-of-distribution.

### 4.3 STABLE RECURRENT OFF-POLICY RL TRAINING

The last core component of our algorithm is RL training. We follow the MBPO paradigm (Janner et al., 2019), using model-free RL on a mixture of real and model-generated data. Since the real data is off-policy and $\hat{\mathbb{P}}_{\mathcal{D}}$ induces a POMDP for the agent, we adopt recurrent off-policy RL (Ni et al., 2022) rather than on-policy RL (Fellows et al., 2025). We use a recurrent actor $\pi_{\nu}(a_t \mid h_t)$ and a recurrent critic $Q_{\omega}(h_t, a_t)$, each with a separate RNN encoder ($\nu_{\phi}(h_t)$ and $\omega_{\phi}(h_t)$, respectively) for stability (Ni et al., 2022). Long rollout horizons and the need for test-time generalization require memory spanning entire episodes (up to 1000 steps in our tasks), which exceeds the capacity of *vanilla* RNNs (Ni et al., 2023). To address this, we extend memoroid (Morad et al., 2024) to actor-critic architectures using linear recurrent units (LRUs) (Orvieto et al., 2023) as the RNN encoders, enabling both long-term memory and parallel optimization. While a memoroid-based architecture is powerful, it was originally designed for online POMDPs. Bridging the gap to epistemic POMDPs in offline RL introduces new challenges, motivating several design choices that we detail below.

**Design choice 4: balancing real and imagined data with $\kappa \in (0, 1)$.** Following MBPO, we introduce a mixing ratio $\kappa$ to weight real versus imagined transitions in the off-policy RL loss (see Sec. E). Prior work shows this ratio is sensitive in both offline and online RL (Lai et al., 2021). Intuitively, when real data is of higher quality, a larger $\kappa$ is preferable (Zhang et al., 2023).

**Design choice 5: small context encoder learning rate $\eta_{\phi}$ for training stability.** The learning rate is a critical hyperparameter in deep RL, and recurrent off-policy RL is especially sensitive to it. In online POMDPs, RESeL (Luo et al., 2024a) shows that recurrent encoders require much smaller (around $30\times$) learning rates than those of actor or critic heads, since history representations can diverge exponentially with history length even under tiny parameter changes.

### 4.4 OVERALL ALGORITHM

We now present the full method in Algorithm 1, integrating all design choices with the core rollout subroutine (Algorithm 2). NEUBAY remains conceptually simple, using single-step models and no

---

[2]With probability $1 - \delta$, for $(s, a) \in \text{supp}_{\mathcal{S} \times \mathcal{A}}(\mathcal{D})$, $\text{TV}(m_{\mathcal{D}}(s, a), m^*(s, a)) \leq c_{m^*, \delta}/\sqrt{n_{\mathcal{D}}(s, a)}$, where $n_{\mathcal{D}}(s, a)$ is the visitation count in $\mathcal{D}$ and $c_{m^*, \delta}$ is a constant (Kumar et al., 2020, Section D.3).

Table 1: Comparison of offline RL methods on the **D4RL locomotion** benchmark. We report mean normalized scores for all baselines, with ±std for competitive baselines. The **best mean score** is bolded, and marked methods are statistically similar under a $t$-test. Our results use 6 seeds, each evaluated at the final step with 20 episodes.

| Dataset | Model-free | | Conservative model-based | | | | | | | | | Bayesian-inspired | | | | Ours |
|---|---|---|---|---|---|---|---|---|---|---|---|---|---|---|---|---|
| | CQL | EDAC | COMBO | RAMBO | MOPO | LEQ | MOBILE | MoMo | ADMPO | SUMO | VIPO | APE-V | MAPLE | MoDAP | CBOP | NEUBAY |
| hc-random | 31.3 | 28.4 | 38.8 | 40.0 | 38.5 | 30.8±3.3 | 39.3±3.0 | 39.6±3.7 | **45.4**±2.8 | 34.9±2.1 | 42.5±0.2 | 29.9 | 41.5 | 36.5±1.8 | 32.8±0.4 | 37.0±3.3 |
| hp-random | 5.3 | 25.3 | 17.9 | 21.6 | 31.7 | 32.4±0.3 | 31.9±0.6 | 18.3±2.8 | 32.7±0.2 | 30.8±0.9 | **33.4**±1.9 | 31.3 | 10.7 | 8.9±1.1 | 31.4±0.0 | 24.5±28.5 |
| wk-random | 5.4 | 16.6 | 7.0 | 11.5 | 7.4 | 21.5±0.1 | 17.9±6.6 | 26.8±3.3 | 22.2±0.2 | 27.9±2.0 | 20.0±0.1 | 15.5 | 22.1 | 23.1±1.6 | 17.8±0.4 | **34.1**±6.8 |
| hc-med-rep | 45.3 | 61.3 | 55.1 | **77.6** | 72.1 | 65.5±1.1 | 71.7±1.2 | 72.9±1.8 | 67.6±3.4 | 76.2±1.3 | 77.2±0.4 | 64.6 | 69.5 | 67.3±3.4 | 66.4±0.3 | 72.1±2.4 |
| hp-med-rep | 86.3 | 101.0 | 89.5 | 92.8 | 103.5 | 103.9±1.3 | 103.9±1.0 | 104.0±1.8 | 104.4±0.4 | 109.9±1.4 | 109.6±0.9 | 98.5 | 85.0 | 94.2±4.8 | 104.3±0.4 | **110.6**±0.7 |
| wk-med-rep | 76.8 | 87.1 | 56.0 | 86.9 | 85.6 | 98.7±6.0 | 89.9±1.5 | 90.4±7.7 | 95.6±2.1 | 78.2±1.5 | 98.4±0.3 | 82.9 | 75.4 | 88.4±4.2 | 92.7±0.9 | **99.3**±19.3 |
| hc-medium | 46.9 | 65.9 | 54.2 | 68.9 | 73.0 | 71.7±4.4 | 74.6±1.2 | 77.1±0.9 | 72.2±0.6 | **84.3**±2.4 | 80.0±0.4 | 69.1 | 48.5 | 77.3±1.1 | 74.3±0.2 | 78.6±1.6 |
| hp-medium | 61.9 | 101.6 | 97.2 | 96.6 | 62.8 | 103.4±0.3 | 106.6±0.6 | **110.8**±2.3 | 107.4±0.6 | 104.8±2.1 | 107.7±1.0 | — | 44.1 | 106.6±1.9 | 102.6±0.1 | 54.2±7.2 |
| wk-medium | 79.5 | **92.5** | 81.9 | 85.0 | 84.1 | 74.9±26.9 | 87.7±1.1 | 95.0±1.4 | 93.2±1.1 | 94.1±2.5 | 93.1±1.8 | 90.3 | 81.3 | 81.1±6.5 | 95.5±0.4 | **106.4**±23.0 |
| hc-med-exp | 95.0 | 106.3 | 90.0 | 93.7 | 90.8 | 102.8±0.4 | 108.2±2.5 | 107.9±1.9 | 103.7±0.2 | 106.6±2.4 | **110.0**±0.4 | 101.4 | 55.4 | 103.4±4.3 | 105.4±1.6 | 109.5±8.7 |
| hp-med-exp | 96.9 | 110.7 | 111.1 | 83.3 | 81.6 | 109.4±1.8 | 112.6±0.2 | 109.1±0.4 | 112.7±0.3 | 107.8±0.7 | 113.2±0.1 | 105.7 | 95.3 | 94.5±7.8 | 111.6±0.2 | **114.8**±0.5 |
| wk-med-exp | 109.1 | 114.7 | 103.3 | 68.3 | 112.9 | 108.2±1.3 | 115.2±0.7 | 118.4±0.9 | 114.9±0.3 | **122.8**±0.4 | 117.7±1.0 | 110.0 | 107.0 | 112.2±2.8 | 117.2±0.5 | 120.7±1.3 |
| AVG | 61.6 | 76.0 | 66.8 | 68.9 | 70.3 | 76.9 | 80.0 | 80.9 | 81.0 | 81.5 | **83.6** | — | 61.3 | 74.5 | 79.3 | 80.1 |

uncertainty penalty, while carefully addressing the challenges of Bayesian offline RL. We highlight the lines where NEUBAY differs from prior work. For completeness, Sec. E provides details on RL loss and stopping criteria, although they are not needed to follow the experiments below.

## 5 EXPERIMENTS

**Benchmarks.** We evaluate on two widely used offline RL suites for continuous control: D4RL (Fu et al., 2020) and NeoRL (Qin et al., 2022). From **D4RL**, we use the **locomotion** benchmark (**12** datasets) spanning three MuJoCo environments: halfcheetah (hc), hopper (hp), and walker2d (wk). Each one provides four data regimes: random, medium-replay, medium, and medium-expert. From **NeoRL**, we use its **locomotion** benchmark (**9** datasets) built on the same environments, but with data collected from more deterministic policies, resulting in narrower coverage and different data regimes: Low, Medium, High. We also include the **Adroit** benchmark (**6** datasets) from D4RL, where a 28-DoF robotic arm manipulates objects to solve tasks (pen, door, hammer) with data collected from human demonstrations (human) and behavior cloning (cloned). The difficulty of Adroit lies in its high-dimensional control, small data size, and low-data quality. Lastly, we perform experiments on **AntMaze** (**6** datasets), where an 8-DoF ant robot must reach a goal position in a maze. Tasks span maze layouts of different sizes (umaze, medium, large) and start distributions (play, diverse), and are challenging for model-based RL due to navigation under sparse rewards. See more details in Sec. F.

**Baselines.** We evaluate against a broad set of offline RL baselines based on each benchmark. We group them into three categories, with a focus on model-based and Bayesian-inspired methods:

- **Conservative model-free RL:** behavior cloning (BC), CQL (Kumar et al., 2020), IQL (Kostrikov et al., 2022), EDAC (An et al., 2021), ReBRAC (Tarasov et al., 2023a).
- **Conservative model-based RL:** MOPO (Yu et al., 2020), COMBO (Yu et al., 2021), RAMBO (Rigter et al., 2022), ARMOR (Bhardwaj et al., 2023), MOBILE (Sun et al., 2023), LEQ (Park & Lee, 2025), MoMo (Srinivasan & Knottenbelt, 2024), ADMPO (Lin et al., 2025), SUMO (Qiao et al., 2025), VIPO (Chen et al., 2025), ScorePen (Liu et al., 2025).
- **Bayesian-inspired methods:** APE-V (Ghosh et al., 2022), MAPLE (Chen et al., 2021c), CBOP (Jeong et al., 2023), MoDAP (Choi et al., 2024).

**Algorithm setup.** In our main experiments (Sec. 5.1), we adopt the following **default** design choices: ensemble size $N = 100$, layer normalization in the world models, uncertainty threshold $\mathcal{U}(\zeta)$ of $\zeta = 1.0$. The learning rates of the actor and critic MLP heads are fixed to $1 \times 10^{-4}$, while their RNN encoders share a tied learning rate $\eta_\phi$ swept over $[3 \times 10^{-7}, 1 \times 10^{-4}]$, the exact range being benchmark-dependent. Similarly, the real data ratio $\kappa$ is swept within $[0.05, 0.95]$, also depending on the benchmark. We report the best results for each dataset. Note that this yields *two* per-dataset hyperparameters, consistent with common practice in conservative model-based RL (Yu et al., 2020; Rigter et al., 2022; Sun et al., 2023). Implementation details are provided in Sec. G. In Sec. 5.2, we conduct sensitivity and ablation studies on design choices.

### 5.1 BENCHMARKING RESULTS

We present results for D4RL locomotion in Tab. 1, NeoRL locomotion in Tab. 10, D4RL Adroit in Tab. 11, and D4RL AntMaze in Tab. 12. Overall, NEUBAY achieves competitive performance relative

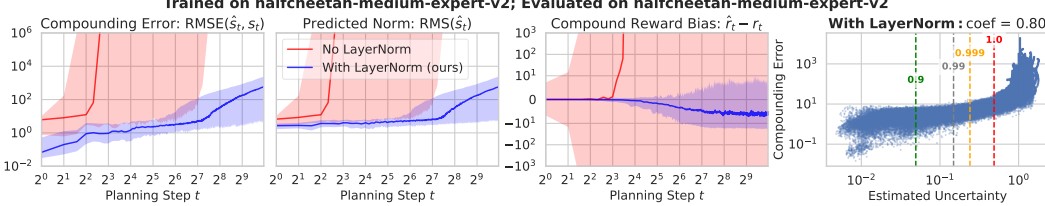

Figure 5: **Effect of LayerNorm in world models** trained and evaluated on halfcheetah-medium-expert-v2. We collect 200 rollouts and truncate only on `float32` overflow, without using an uncertainty threshold. For each metric, we plot the **median** (solid line) together with the **5-95% percentile band** across rollouts. The rightmost scatter plot show the Spearman's rank coefficient in the with-LayerNorm setting; vertical lines mark uncertainty thresholds $\zeta \in \{0.9, 0.99, 0.999, 1.0\}$. **Full results** and plotting setup are shown in Sec. H.2.

to prior conservative algorithms. The average normalized scores are 80.1 vs. 83.6 for the best baseline (VIPO) in D4RL locomotion, 64.7 vs. 73.3 for the best baseline (VIPO) in NeoRL locomotion, 21.1 vs. 28.1 for the best model-based baseline (MoMo) in D4RL Adroit, and 28.8 vs. 64.9 for the best model-based baseline (LEQ) in D4RL AntMaze. NEUBAY establishes **new state-of-the-art** (SOTA) mean performance on 7 of the 33 datasets, with 3 gains statistically significant. In terms of magnitude, it advances walker-random-v2 ($27.9 \to 34.1$), walker-medium-v2 ($95.5 \to 106.4$), pen-cloned-v1 ($74.1 \to 91.3$), and hammer-cloned-v1 ($5.0 \to 14.4$). Furthermore, NEUBAY surpasses prior Bayesian-inspired methods on average performance and most per-task scores, despite their reliance on conservatism.[3]

**When is NEUBAY better than conservative methods?** We group the 33 datasets into 3 categories:

- **Low-quality datasets (11 tasks):** includes 3 D4RL random, 3 NeoRL Low, 2 Adroit door, 2 Adroit hammer, and AntMaze umaze-diverse. Consistent with the bandit insight (Sec. 3), NEUBAY ranks among the best methods in 5 of 11 tasks and achieves reasonable performance in 10 of 11.
- **Medium-quality, moderate coverage (7 tasks):** includes 3 D4RL medium-replay, 3 D4RL medium-expert[4], and Adroit pen-cloned. NEUBAY ranks among the best methods in 5 of 7 tasks, without any failures.
- **Medium-quality, narrow coverage (15 tasks):** includes 3 D4RL medium, 3 NeoRL Medium, 3 NeoRL High, Adroit pen-human, 5 AntMaze tasks. NEUBAY is weaker, ranking among the best methods in only 3 of 15 tasks, while falling short in 10. This gap from the bandit insight stems from the difficulty of posterior modeling on narrow datasets, where severe model error in high-dimensional spaces can mislead NEUBAY's RL agent. This becomes critical in the AntMaze domain due to the contact-rich dynamics and sparse reward signals (see Sec. H.5 for analysis). Future advances in world modeling and planning may help close this gap.

## 5.2 WHAT MATTERS IN NEUBAY?

To understand which components are critical to the success of NEUBAY, we conduct sensitivity studies on the design choices introduced in Sec. 4 and ablation study on the uncertainty penalty and Markov agent, using the best hyperparameters identified in Sec. 5.1.

**LayerNorm in the world model and uncertainty-based truncation jointly enables long-horizon rollouts.** Fig. 5 (see Sec. H.2 for full results) shows that without LN, predicted state norms diverge quickly (around 10 steps here), driving exploding compounding errors. With LN, state norms remain bounded, which suppresses state error growth and stabilizes reward predictions. This matches our intuition based on Eq. 5: by normalizing features at each step, LN constrains prediction magnitudes and thus compounding error. Moreover, the rightmost scatter plot shows what uncertainty-based truncation *would* accomplish: although we display full rollouts to reveal their divergence, applying thresholds would cut them off before entering high-error regions (here, around $\geq 10^1$). Thus, *LayerNorm prevents error explosion, while uncertainty cutoff provides a complementary safeguard.*

**Adaptive long-horizon planning is the *decisive* factor for success.** We study the role of adaptive horizons by varying $\zeta \in \{0.9, 0.99, 0.999, 1.0\}$. Almost on all datasets, we find that the most aggressive choice, $\zeta = 1.0$, delivers the best performance. Fig. 1 shows a typical case, with full plots in Sec. H.3, a summary in Tab. 2, and complete results in Tab. 13.

---

[3]An exception is MoDAP, which avoids conservatism by continuing model training during policy learning.
[4]D4RL expert and NeoRL High are treated as medium-quality, as their behavior policies are not fully optimal.

Table 2: **Sensitivity and ablation results averaged across benchmarks.** The highlighted setting ($N$=100, $\lambda$=0.0, $\zeta$=1.0, using the entire history as agent input) is the main result. Ablations vary one hyperparameter at a time, except for the Markov agent, where we sweep the real data ratio $\kappa$ for a fair comparison. Shading shows degradation level: light (3–10), medium (10–30), dark (>30). Full per-dataset results appear in Tab. 13.

| Benchmark | Ensemble size $N$ | | | Unc. penalty coef. $\lambda$ | | | | | Truncation threshold $\zeta$ | | | | Agent input | |
|---|---|---|---|---|---|---|---|---|---|---|---|---|---|---|
| | 100 | 20 | 5 | 0.0 | 0.04 | 0.2 | 1.0 | 5.0 | 1.0 | 0.999 | 0.99 | 0.9 | Hist. | Mark. |
| D4RL Locomotion (12 tasks) | 80.1 | 71.6 | 65.9 | 80.1 | 79.8 | 80.4 | 73.1 | 57.4 | 80.1 | 69.6 | 45.1 | 22.5 | 80.1 | 75.8 |
| NeoRL Locomotion (9 tasks) | 64.7 | 60.3 | 59.8 | 64.7 | 61.5 | 59.8 | 58.3 | 42.5 | 64.7 | 58.8 | 36.3 | 16.7 | 64.7 | 66.8 |
| D4RL Adroit (3 non-zero tasks) | 42.2 | 30.0 | 32.6 | 42.2 | 33.7 | 34.2 | 38.1 | 34.1 | 42.2 | 17.8 | 16.0 | 1.9 | 42.2 | 36.4 |
| D4RL AntMaze (4 non-zero tasks) | 43.2 | 33.2 | 28.8 | 43.2 | 5.0 | 3.2 | 4.2 | 12.0 | 43.2 | 35.8 | 38.4 | 32.7 | 43.2 | 1.3 |

Notably, using $\zeta = 1.0$ yields **64-512 steps** for 75th-percentile horizon and **256-1000 steps** for maximum horizon, **in 21 out of 23 tasks with $T = 1000$**. These horizons are surprisingly long, running counter to the conventional wisdom in model-based RL that long rollouts hurt performance (Janner et al., 2019). That belief is justified when horizons are *fixed*, worst-case compounding error grows exponentially and makes long rollouts impractical (e.g., Fig. 5). Our results show that with *adaptive* horizons, compounding errors are kept under control, allowing rollouts to extend far beyond what fixed horizons permit. In addition, the large ensemble of models together with the *neutral* Bayesian objective prevents the agent from overcommitting to any single erroneous prediction, yielding robust long-horizon planning.

On the other hand, performance often collapses to *near zero* (i.e., scores $\leq 5$) with smaller thresholds such as 0.9 (16 datasets) or 0.99 (6 datasets). The consistent failure mode behind these cases is **severe value overestimation**, where estimated Q-values ($\mathbb{E}_{(h_t, a_t) \sim \mathcal{D}}[Q_\omega(h_t, a_t)]$) are much higher (2nd column in Fig. 1) compared to the actual performances. These results support our intuition in Eq. 6: without conservatism, Bayesian RL mitigates overestimation by relying more on imagined rewards (lower-biased as shown in Fig. 5 reward panel) using long horizons than on bootstrapped values. Crucially, adaptive horizons allow the algorithm to "trust" the model only within its in-distribution confidence region. As indicated by Fig. 4, the region at $\zeta = 1.0$ is much larger than at $\zeta = 0.9$, yielding substantially longer horizons that provide useful data for both augmentation and mitigating overestimation.

**Introducing conservatism to NEUBAY helps some tasks, but not on average.** To study conservatism in NEUBAY, we penalize the rewards with $\lambda \frac{U_\theta(\hat{s}, \hat{a})}{\mathbb{E}_{(s,a) \sim \mathcal{D}}[U_\theta(s,a)]}$, normalized by the dataset average so that $\lambda$ is more comparable across datasets. As summarized in Tab. 2, a strong uncertainty penalty ($\lambda = 1.0$ or $5.0$) generally hurts performance, while a small penalty ($\lambda = 0.04$) performs comparably to NEUBAY ($\lambda = 0$). The impact remains dataset-dependent, as detailed in Tab. 13.

Consistent with the bandit intuition, heavy penalties significantly worsen performance on 6 of 8 low-quality datasets, while leaving $\star$-medium-expert datasets largely unaffected. In contrast, some tasks benefit substantially: hopper-random-v2 ($24.5 \rightarrow 48.2$, a new SOTA), hopper-medium-v2 ($54.2 \rightarrow 105.8$), and pen-human-v1 ($20.8 \rightarrow 35.9$), but each requires a different $\lambda$, highlighting the need for tuning. However, penalties are not a universal remedy for narrow data: Walker2d-v3-Medium and halfcheetah-medium-v2 still fall short of the best baselines, even after tuning. Finally, in the AntMaze domain, penalties severely harm performance, reflecting the already poor quality of these datasets.

**Context encoder learning rate and real data ratio have to be tuned per dataset.** The best values of $\eta_\phi$ and $\kappa$ are reported in Tab. 4–Tab. 6, with selective learning curves shown in Fig. 13–Fig. 14. We find the optimal encoder learning rates are generally smaller than those used in online POMDPs (Luo et al., 2024a). For example, $1 \times 10^{-6}$ and $3 \times 10^{-7}$ yield the best results on 7 datasets. Our intuition is that, in Bayesian offline RL, smaller learning rates help curb overestimation by slowing down learning. For the real data ratio, we find $\kappa = 0.05$, widely used in prior conservative algorithms (Yu et al., 2020), often yields poor performance. In Bayesian offline RL, large $\kappa$ acts as a *softened* uncertainty penalty, limiting overtrust in the model while avoiding explicit penalties.

**Larger ensemble size improves performance.** As summarized in Tab. 2 and detailed in Tab. 13, reducing the ensemble size $N$ to 20 or 5 clearly degrades performance, though not drastically on most tasks. This indicates that $N = 100$ is close to the practical limit of what ensembling can offer for these tasks, and future work may explore more scalable ways to approximate the posterior.

**Using a Markov agent in NEUBAY works well on most locomotion tasks, but fails in AntMaze.** Popular offline RL methods employ Markov agents, while NEUBAY uses history-dependent agents due to its epistemic POMDP formulation. To assess the practical importance of memory, we replace

the history-dependent agent in NEUBAY with a Markov one, sweep over the same real-data-ratio range, and report the best results in Tab. 2 and Tab. 13. The Markov version performs similarly to the history-dependent one on most *locomotion* tasks, especially on the NeoRL benchmark ($64.7 \to 66.8$). However, it suffers severe degradation on hopper-medium-replay-v2 ($110.6 \to 47.2$) and on all AntMaze tasks ($43.2 \to 1.3$). This pattern suggests that in locomotion domains, epistemic uncertainty is relatively mild, so using a single observation may often infer the model index, making memory less critical. As a result, NEUBAY with a Markov agent remains a strong baseline in these settings. In contrast, AntMaze has high epistemic uncertainty, especially about the maze layout which is crucial to navigation, so memory is needed to accumulate information over time.

## 5.3 COMPUTATION COSTS

The total training cost of NEUBAY, similar to other model-based offline RL methods, consists of two phases: *world-model training* and *agent training*. In practice, the world model is trained once per seed, and its ensemble checkpoint is reused when tuning agents on top of it. Thus, world-model training is a **one-time cost**, while agent training dominates the overall computation. All experiments were run with three seeds in parallel on a single NVIDIA L40S GPU (48 GB). The implementation is in JAX, leveraging `vmap` for vectorized ensemble and `associative_scan` for efficient LRU optimization.

Consider halfcheetah-medium-expert-v2 (2M gradient steps) with an ensemble size of $N = 100$.

- World-model training (one-time): 6 hours per seed (18 hours for 3 seeds) and 10.7 GB memory.
- Recurrent agent training (main cost): **4.4 hours** per seed (13.2 hours for 3 seeds) and **2.6 GB** memory (1.5 GB for world model, 1.1 GB for agent).
- Markov agent training (ablation): 2.6 hours per seed (7.7 hours for 3 seeds) and 1.5 GB memory.

Notably, our agent training time is already faster than MOPO's PyTorch implementation reported by Jackson et al. (2025) (5.6 hours for 2M steps); this is to contextualize compute cost, not as a JAX–PyTorch comparison. Finally, as detailed in Sec. G.3, the rollout inference cost is minimal, and increases in ensemble size $N$ or rollout size $K$ have only a minor effect on runtime.

## 6 CONCLUSION AND FUTURE WORK

**Tuning guidelines for NEUBAY.** Drawing on our benchmarking and sensitivity studies in Sec. 5, we suggest the following guidelines when applying NEUBAY to new tasks or after modifying a key module (e.g., world model, uncertainty quantifier, or memory encoder):

- *Check dataset quality.* NEUBAY prefers low-quality or moderate-coverage datasets, but can underperform on narrow, medium-quality ones possibly due to current limits in posterior modeling.
- *Defaults first.* Start with a large ensemble size, layer normalization in the world model, and uncertainty threshold $\zeta = 1.0$.
- *Context encoder rate.* Tune $\eta_\phi$ within a wide range, typically $3$–$300\times$ smaller than the MLP head's learning rate.
- *Real data ratio.* Adjust $\kappa$ with $[0.5, 0.8]$ as a robust starting range.
- *Overestimation control.* Monitor estimated values; if severe overestimation occurs, mitigate it by reducing the discount factor $\gamma$ or lowering $\eta_\phi$, as suggested by our analysis in Eq. 6.

**Conclusion.** In this paper, we revisit conservatism as the dominant principle in offline RL and show that it is not universally optimal. Instead, we advance a Bayesian perspective that models epistemic uncertainty and trains history-dependent agents to maximize expected rewards over a posterior of world models. Building on this principle, we propose NEUBAY, a practical algorithm that mitigates compounding error with layer normalization, reduces value overestimation with adaptive long-horizon planning, and stabilizes recurrent training. Experiments across diverse benchmarks show that NEUBAY generally matches or surpasses strong conservative baselines, excels particularly on low-quality and moderate-coverage datasets, and challenges the belief that short-horizon planning is necessary in model-based RL.

**Future work.** We see several promising directions for future work. Improving world models, through multi-step prediction or generative models, can help push the limits of Bayesian offline RL. Better uncertainty quantification remains key for planning, suggesting deeper connections to Bayesian inference are worth exploring. Finally, reducing sensitivity to hyperparameters and dataset characteristics, as well as advancing off-policy evaluation for tuning, would make NEUBAY more robust and practical.

## REPRODUCIBILITY STATEMENT

We release our full codebase and pretrained world ensemble checkpoints in the supplementary material. We further provide detailed computation-cost analyses in Sec. 5.3. We provide dataset details in Sec. F and implementation details in Sec. G. These resources, together with the tuning guidelines in Sec. 6, are intended to ensure reproducibility and to make agent training on top of the released checkpoints straightforward.

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

APPENDIX

## A RELATED WORK

### A.1 OFFLINE CONSERVATIVE MODEL-BASED METHODS

Model-based methods can be broadly grouped into background planning and decision-time planning (Sutton & Barto, 2018, Chapter 8.8). **Background planning**, such as Dyna-style methods (Sutton, 1990; Janner et al., 2019), uses model-generated rollouts to learn a global policy or value function, which is then queried for action selection. **Decision-time planning**, such as model predictive control (MPC), performs online look-ahead at test time to select actions for the current state. Our work belongs to background planning. Below, we review these paradigms within the offline setting, focusing on how they incorporate the conservatism.

**Conservative background planning algorithms.** In this category of work, most methods enforce conservatism by constructing a *pessimistic MDP* that penalizes imagined state-action pairs, which we refer to as **uncertainty-penalized pessimism** in Sec. B. The earliest examples are MOReL (Kidambi et al., 2020) and MOPO (Yu et al., 2020), both approximating the uncertainty set with world ensembles. MOReL adopts strong pessimism by mapping any state-action pair with ensemble disagreement above a threshold to an absorbing state with a large penalty. This aggressive design disables value bootstrapping on uncertain regions and explains why MOReL supports long-horizon planning (up to 500 steps) even with severe compounding errors. In contrast, MOPO applies a milder penalty

to imagined rewards based on the aleatoric uncertainty, retaining bootstrapping and thus limiting rollouts to very short horizons (typically 1-5 steps).

Building on MOPO, several follow-up works redesign the uncertainty quantifier: MOBILE (Sun et al., 2023) uses inconsistencies in Bellman operators across ensemble members; Kim & Oh (2023) uses the inverse frequencies of state-action pairs; MoMo (Srinivasan & Knottenbelt, 2024) adopts energy-based models; SUMO (Qiao et al., 2025) employs k-nearest neighbors. Other works introduce alternative conservative mechanisms: COMBO (Yu et al., 2021) uses CQL regularizer; LEQ (Park & Lee, 2025) uses lower expectile regression. From the model side, several works improve the model learning or rollout sampling: Luo et al. (2024b) trains discriminators on $(s, a, s')$ to resample model rollouts by fidelity; VIPO (Chen et al., 2025) augments the standard MLE loss with a value-consistency objective that aligns behavior-policy values under model and true dynamics.

A different line of work jointly trains adversarial world models and policies, rather than freezing the models after pretraining, which we refer to as **adversarial model-based pessimism** in Sec. B. Several adversarial objectives have been explored: RAMBO (Rigter et al., 2022) minimizes value estimates; Yang et al. (2022) minimizes the divergence between real and imagined state-action distributions; ARMOR (Bhardwaj et al., 2023) minimizes value differences between the current and a reference policy; Rigter et al. (2023) extends the uncertainty set to incorporate aleatoric uncertainty.

Our work differs from these prior lines of research in principle, core components, and algorithmic design. We replace conservatism with Bayesianism as the guiding principle, substituting the standard Markov actor-critic with a history-dependent one required by Bayesian principle, and derive the NEUBAY algorithm with adaptive long-horizon planning, fundamentally distinct from prior methods. Moreover, prior works typically rely on tuning two critical hyperparameters: the *conservatism coefficient* $\lambda$ and *rollout horizon H*, to achieve strong performance. In contrast, NEUBAY eliminates the need for $\lambda$ by design and replaces $H$ with an uncertainty quantile $\zeta$, which remains fixed at 1.0 in main experiments.

**Conservative decision-time planning algorithms.** In this line of work, a planner is used at test time to select actions from the current state. The planner is composed of a world model, a conservative policy obtained by behavior cloning (Argenson & Dulac-Arnold, 2020), and a conservative value function learned by fitted Q evaluation (Zhan et al., 2021) or other pessimistic estimators (Janner et al., 2021). Given the conservative policy as a prior, the planner samples exploratory trajectories from world models and chooses the action with the highest estimated value. Decision-time planning can discover better actions at test time by performing targeted exploration from a specific state, particularly useful in unseen states. It is distinct from classical RL because it does not learn a global policy to maximize returns; instead, it directly searches for actions.

### A.2 OFFLINE BAYESIAN-INSPIRED AND NON-CONSERVATIVE RL

**Bayesian-inspired algorithms.** Several offline RL works draw inspiration from Bayesian ideas, such as using model posteriors and connections to Bayes-adaptive MDPs (Duff, 2002). However, their formulations or algorithms typically differ from optimizing the epistemic POMDP in Eq. 4, which is the focus of our work. For example, although Ghosh et al. (2022) introduces the Bayesian model-based formulation in Eq. 4, their proposed APE-V algorithm adopts a *conservative model-free* approximation. Rather than maintaining a posterior over dynamics models, APE-V uses an ensemble of belief-state Q-functions as a surrogate posterior over values, trained purely via TD errors. Each Q-function is trained using SAC-N (An et al., 2021), which enforces conservatism by minimizing over Q-ensemble predictions.

Similar to our algorithm, **MAPLE** (Chen et al., 2021c) and **MoDAP** (Choi et al., 2024) learn an ensemble of models for recurrent policy training. Unlike our approach, however, they store the hidden states of recurrent policies in the replay buffer and reuse them to initialize the policy's context for rollouts and updates. This design is known to induce context staleness in recurrent RL (Kapturowski et al., 2018), whereas our algorithm avoids this issue by storing and sampling full histories directly. MAPLE also reintroduces an uncertainty penalty and terminates rollouts based on a predefined state boundary, making it a conservative method. MoDAP remains penalty-free but fine-tunes the world models during policy learning to maintain ensemble diversity; therefore, its objective departs from the epistemic POMDP, which requires freezing the posterior during policy optimization.

Other works are Bayesian-inspired in different ways. CBOP (Jeong et al., 2023) uses a model posterior to weight multi-step TD targets in an MVE-style update (Feinberg et al., 2018) and applies lower-confidence penalties. Chen et al. (2024) proposes a Bayesian Monte Carlo planning method with an uncertainty penalty, used as a policy-improvement operator.

In contrast, our work aims to stay as close as possible to the epistemic POMDP: we avoid conservatism and do not fine-tune models during policy learning. The approximation in the algorithm level arises in the rollout distribution: we begin rollouts from intermediate histories and truncate them using an uncertainty threshold to mitigate compounding errors. As discussed in Sec. 4.2, using long rollouts keeps this deviation small in practice.

**Offline RL algorithms without conservatism.** Although conservatism dominates modern offline RL, classic offline (batch) RL was originally developed without any conservatism (Lagoudakis & Parr, 2003; Ernst et al., 2005; Riedmiller, 2005). These methods are based on *fitted Q-iteration*, which directly applies Markov Q-learning to offline data. While effective on small-scale problems with sufficient data coverage (Riedmiller, 2005), such algorithms are known to fail in high-dimensional settings (Fujimoto et al., 2019). More recently, Agarwal et al. (2020b) provides an *optimistic* perspective, showing that standard off-policy RL trained on the 50M transitions from DQN's replay buffer can outperform the behavior policy. Likewise, Yarats et al. (2022) demonstrates that off-policy RL can surpass conservative methods on diverse-coverage datasets collected by unsupervised agents.

In the model-based setting, MBPO (Janner et al., 2019), using short-horizon rollouts without conservatism, is known to underperform conservative counterparts such as MOPO in offline RL benchmarks (Yu et al., 2020). However, MuZero Unplugged (Schrittwieser et al., 2021) shows that MuZero, without conservatism, can achieve strong performance on the RL Unplugged suite (Gulcehre et al., 2020), which contains 200M Atari transitions and DMC control datasets using replay buffer from near-optimal RL agents. Zhai et al. (2024) observes that making MOPO optimistic instead of pessimistic can achieve strong performance on halfcheetah-random-v2, but worse on other D4RL tasks.

In summary, prior work on offline RL without conservatism largely relies on standard off-policy RL, which treats offline RL as optimizing a single MDP and thus follows an *optimistic* principle (Agarwal et al., 2020b). Such optimism can work well when the dataset has broad state-action coverage but typically fails under limited coverage. Our approach takes a different non-conservative path: instead of being optimistic, it follows a neutral Bayesian principle that lies **between optimism and pessimism**. This allows NEUBAY to avoid the failure modes of optimistic methods and, for the first time to our knowledge, extends the effectiveness of non-conservative offline RL to *low-quality* and *moderate-coverage* datasets, while still benefiting from diverse coverage when available.

### A.3 MODEL-GENERATED ROLLOUTS IN RL

**Effect of rollout length on value estimation.** Sims et al. (2024) identifies the *edge-of-reach* problem in offline conservative model-based RL (MBRL): when short-horizon rollouts are used, bootstrapping often occurs on states whose Q-values are never directly trained, leading to substantial overestimation once uncertainty penalties vanish (e.g., under true dynamics). Closely related to our work, they emphasize that such overestimation induced by short rollouts constitutes a distinct failure mode in offline MBRL, separate from classic compounding errors. To mitigate this, they replace dynamics-based uncertainty penalties with value-based ones via pessimistic Q-ensembles (An et al., 2021), while still relying on short-horizon rollouts.

Our work extends this line of reasoning in two ways. (1) We show that the same overestimation mechanism arises under Bayesian Bellman backups and extend Sims et al. (2024, Proposition 1) to Bayesian setting in Sec. C.2. (2) Instead of conservative short-horizon rollouts, we use adaptive long-horizon rollouts that exploit the vanishing factor $\gamma^H$ to naturally reduce the bootstrapped error and remove the need for conservatism.

**Reducing compounding errors and the scale of rollout horizon.** Model-generated rollouts suffer from compounding prediction errors (Talvitie, 2014; Lambert et al., 2022), which can grow quickly with the rollout horizon. These errors cause the performance gap between the policy evaluated under the learned model and under the true MDP, as formalized by theory such as the simulation lemma (see Uehara & Sun (2022, Lemma 9) and our Lemma 4). To prevent large compounding errors from harming policy learning, most online and offline MBRL methods (Janner et al., 2019;

Yu et al., 2020; Lu et al., 2021; Hafner et al., 2023; Hansen et al., 2024) restrict rollouts to very short horizons (typically **1-20 steps**). These approaches often share a minimalist setup: standard MLP world models without layer normalization, pure MLE training, and rollout procedures without uncertainty awareness.

Prior works attempt to reduce compounding errors and thus enable longer rollouts along three ways: *model architecture*, *training objective*, and *inference strategy*. On the architectural side, improving model smoothness can enhance generalization on unseen states (Asadi et al., 2018). On the training-objective side, multi-step architecture predicts future states given an initial state and an action sequence (Asadi et al., 2019), enabling horizons up to 500 steps with Transformers in online RL (Ma et al., 2024) and up to 50 steps with RNNs in offline RL (Lin et al., 2025). For inference strategies, MOREC (Luo et al., 2024b) resamples model rollouts using a fidelity estimator and scales horizons to 100 steps.

NEUBAY keeps the training objective fixed to standard one-step MLE, while contributing simple and effective components along the other two dimensions. Architecturally, we apply layer normalization (Ba et al., 2016) to stabilize prediction magnitudes (Sec. 4.1), increasing model smoothness (Asadi et al., 2018) and thus generalization; at inference time, we truncate rollouts using an uncertainty threshold as a proxy for compounding error (Sec. 4.2). Together, these components enable NEUBAY to successfully use horizons of **64-512** steps, which is rare in MBRL literature.

**Mechanism of adaptive horizon.** As discussed in Sec. 4.2, several prior works have used uncertainty thresholds to adaptively truncate rollouts. In the online setting, Pan et al. (2020) truncates rollouts for states whose uncertainty ranks in the top 25% of the current-step batch and caps the horizon at 10. Similarly, Frauenknecht et al. (2024) computes a 95% uncertainty quantile from the batch of first-step predictions in current rollouts, also with a maximum horizon of 10. In the offline setting, Zhan et al. (2021) adopts an 85% uncertainty quantile from the offline dataset to filter rollouts and imposes a maximum horizon of 16. Zhang et al. (2023) uses truncation thresholds proportional to the maximum uncertainty in the offline dataset, with a coefficient of 0.5 and a maximum horizon of 5. Overall, prior work employs adaptive horizons **in a conservative manner**, combining small thresholds with strict horizon caps. In contrast, our approach removes any maximum-horizon constraint and uses a large quantile threshold.

### A.4 BAYESIAN AND PARTIALLY OBSERVABLE RL

**Bayesian RL.** Bayesian RL (Vlassis et al., 2012; Ghavamzadeh et al., 2015) models epistemic uncertainty (also known as ambiguity (Ellsberg, 1961)) for purposes such as exploration (Osband et al., 2016), robustness (Rajeswaran et al., 2017; Derman et al., 2020; Rigter et al., 2021), and generalization (Ghosh et al., 2021; Jiang et al., 2023). Uncertainty can be incorporated in model-free methods (e.g., Bayesian Q-learning (Dearden et al., 1998)) or in model-based methods (e.g., BAMDPs (Duff, 2002)). Depending on the design preference, Bayesian RL can be ambiguity-seeking, ambiguity-neutral, or ambiguity-averse. Our work is grounded in the epistemic POMDP formulation (Ghosh et al., 2021; 2022), an ambiguity-neutral, model-based framework for generalization. This formulation stems from BAMDPs, which optimally balance the exploration-exploitation tradeoff but incur test-time exploratory costs, as illustrated in Sec. 3 and also known as the *cost of exploration* (Vuorio et al., 2024).

Epistemic POMDPs are also related to *meta-RL* (Beck et al., 2023), which likewise seeks to maximize expected performance over a distribution of MDPs. Each MDP is called a task in meta-RL. Bayesian meta-RL (Zintgraf et al., 2020; Dorfman et al., 2021) explicitly models this MDP posterior. The key difference is that in meta-RL the true environment is itself a distribution over MDPs (a particular POMDP), so the task uncertainty is *aleatoric*. In contrast, the epistemic POMDP assumes a single underlying MDP, and its task uncertainty is purely epistemic. As a result, meta-RL methods cannot be applied to solve the epistemic POMDP without modification.

**Recurrent model-free RL for online POMDPs.** Model-free RL offers a simple and effective way to tackle online POMDPs without explicitly learning belief-state representations (Ni et al., 2022; Yang & Nguyen, 2021). Memory is typically implemented with recurrent neural networks (RNNs) such as LSTMs (Hochreiter & Schmidhuber, 1997) or GRUs (Cho et al., 2014), but recent work shows that these architectures struggle with long-term memory (Parisotto et al., 2020; Ni et al., 2023). State-space models (SSMs) with linear recurrence (Orvieto et al., 2023; Gu & Dao, 2023) have emerged as

a compute-efficient alternative to Transformers (Vaswani et al., 2017), balancing long-term memory with parallel optimization (Blelloch, 1990). Recent applications of SSMs to recurrent RL (Lu et al., 2023; 2024; Morad et al., 2024; Luo et al., 2024a; Luis et al., 2024) demonstrate strong performance on online POMDP benchmarks (Morad et al., 2023; Zintgraf et al., 2020; Ni et al., 2022).

# B   FORMAL CONNECTIONS BETWEEN CONSERVATISM AND ROBUSTNESS

In this section, we place prior conservative algorithms in the (soft) robust MDP framework. For classic model-free and adversarial model-based pessimism, we make explicit how their updates correspond to particular uncertainty sets in *robust* MDPs (Wiesemann et al., 2013). For uncertainty-penalized pessimism, we connect it to *soft robust* MDP framework (Zhang et al., 2024b).[5] We follow the notation introduced in Sec. 2.

**Classic model-free pessimism.** Many model-free methods enforce an in-support (Fujimoto et al., 2019; Kumar et al., 2019) or in-sample (Kostrikov et al., 2022; Xu et al., 2023) constraint on the Bellman backup. This amounts to updating a pessimistic value function $Q^{\text{MF}}$ such that, for a transition tuple $(s, a, r, d, s') \in \mathcal{D}$,

$$Q^{\text{MF}}(s, a) \leftarrow r + \gamma(1 - d) \max_{a' \in \mathcal{A}, \text{ s.t. } (s', a') \in \mathcal{D}} Q^{\text{MF}}(s', a'). \tag{7}$$

This update is equivalent to assigning all out-of-dataset state–action pairs the minimal reward $-r_{\max}$, thereby enforcing a worst-case behavior. The corresponding uncertainty set $\mathfrak{M}_{\mathcal{D}}$ in the robust MDP framework (Eq. 3) can be written explicitly:

**Proposition 1.** *If the pessimistic update in Eq. 7 converges to a fixed point, then the induced uncertainty set for the corresponding robust MDP is*

$$\mathfrak{M}_{\mathcal{D}}^{\text{MF}} = \left\{ m \; \middle| \; \begin{array}{ll} m(r, s' \mid s, a) = m_{\mathcal{D}}(r, s' \mid s, a), & \forall (s, a) \in \mathcal{D} \\ m(r, s' \mid s, a) = p(r) \, \mathbb{1}(s' = s_{absorb}), & \forall p \in \Delta([-r_{\max}, r_{\max}]), \quad \forall (s, a) \notin \mathcal{D} \end{array} \right\}, \tag{8}$$

*where $m_{\mathcal{D}}$ is the empirical model (Eq. 2) and $s_{absorb} \notin \mathcal{D}$ is an artificial absorbing state.*

*Proof.* First, let $m \in \mathfrak{M}_{\mathcal{D}}^{\text{MF}}$ and decompose $m(r, s' \mid s, a) = R(r \mid s, a)P(s' \mid s, a)$, and denote the empirical model as $m_{\mathcal{D}}(r, s' \mid s, a) = R_{\mathcal{D}}(r \mid s, a)P_{\mathcal{D}}(s' \mid s, a)$. By construction, $\mathfrak{M}_{\mathcal{D}}^{\text{MF}}$ places no uncertainty on transitions: for $(s, a) \in \mathcal{D}$, $P(s, a)$ equals the empirical transition $P_{\mathcal{D}}(s, a)$, while for $(s, a) \notin \mathcal{D}$, $P(s, a)$ deterministically transitions to $s_{\text{absorb}}$. Thus, all epistemic uncertainty is in the reward function $R$.

Applying the robust MDP framework (Wiesemann et al., 2013), the optimal value function $Q^*$ is Markov and satisfies the robust Bellman optimality equation:

$$Q^*(s, a) = \min_{R(s,a) \in \mathfrak{M}_{\mathcal{D}}^{\text{MF}}(s,a)} \mathbb{E}_{r \sim R(s,a)}[r] + \gamma \mathbb{E}_{s' \sim P(s,a)} \left[ \max_{a' \in \mathcal{A}} Q^*(s', a') \right], \forall (s, a) \in \mathcal{S} \times \mathcal{A}, \tag{9}$$

$$Q^*(s, a) - \gamma \mathbb{E}_{s' \sim P(s,a)} \left[ \max_{a' \in \mathcal{A}} Q^*(s', a') \right] = \min_{R(s,a) \in \mathfrak{M}_{\mathcal{D}}^{\text{MF}}} \mathbb{E}_{r \sim R(s,a)}[r] = \begin{cases} \mathbb{E}_{r \sim R_{\mathcal{D}}(s,a)}[r] & (s, a) \in \mathcal{D}, \\ -r_{\max} & (s, a) \notin \mathcal{D}, \end{cases} \tag{10}$$

where the last line uses the fact that $R(\cdot \mid s, a)$ may be any distribution on $[-r_{\max}, r_{\max}]$, so the worst case is attained by a Dirac mass at $-r_{\max}$.

Therefore, we can simplify Eq. 10 by cases. (1) Absorbing state:

$$\forall a \in \mathcal{A}, \quad Q^*(s_{\text{absorb}}, a) - \gamma \max_{a' \in \mathcal{A}} Q^*(s_{\text{absorb}}, a') = -r_{\max}. \tag{11}$$

Since $Q^*(s_{\text{absorb}}, a)$ is a constant w.r.t. $a$, it follows that $Q^*(s_{\text{absorb}}, a) = -\frac{r_{\max}}{1-\gamma}, \forall a$, reaches the minimal return. (2) Unseen state-action pairs,

$$\forall (s, a) \notin \mathcal{D}, \quad Q^*(s, a) - \gamma \max_{a' \in \mathcal{A}} Q^*(s_{\text{absorb}}, a') = -r_{\max}. \tag{12}$$

---

[5]The term "soft robustness" is used differently in prior work: Derman et al. (2018) use it for a Bayesian formalism, while Zhang et al. (2024b) define it as a risk-sensitive MDP relaxing strict worst-case robustness. In this paper, we adopt both usages but cite them accordingly.

This implies $Q^*(s, a) = -\frac{r_{\max}}{1-\gamma}, \forall (s, a) \notin \mathcal{D}$. (2) Seen state-action pairs,

$$\forall (s, a) \in \mathcal{D}, \quad Q^*(s, a) = \mathbb{E}_{r \sim R_{\mathcal{D}}(s,a)}[r] + \gamma \mathbb{E}_{s' \sim P_{\mathcal{D}}(s,a)}\left[\max_{a' \in \mathcal{A}} Q^*(s', a')\right] \tag{13}$$

$$= \mathbb{E}_{r \sim R_{\mathcal{D}}(s,a)}[r] + \gamma \mathbb{E}_{s' \sim P_{\mathcal{D}}(s,a)}\left[\max_{a' \in \mathcal{A}, \text{s.t. } (s',a') \in \mathcal{D}} Q^*(s', a')\right] \tag{14}$$

The last line follows that $Q^*(s, a_{\text{out}}) = -\frac{r_{\max}}{1-\gamma} \leq Q^*(s, a_{\text{in}}), \forall (s, a_{\text{in}}) \in \mathcal{D}, \forall (s, a_{\text{out}}) \notin \mathcal{D}$. Finally, Eq. 14 recovers the pessimism principle underlying Eq. 7. This includes many model-free offline RL algorithms, such as BCQ (Fujimoto et al., 2019, Equation 10), BEAR (Kumar et al., 2019, Definition 4.1), EMaQ (Ghasemipour et al., 2021, Theorem 3.3), IQL (Kostrikov et al., 2022, Corollary 2.1). □

**Adversarial model-based pessimism.** One class of model-based methods constructs an explicit uncertainty set around the empirical model (Uehara & Sun, 2022; Rigter et al., 2022):

$$\mathfrak{M}_{\mathcal{D}}^{\text{MB}} = \{m \mid \mathbb{E}_{(s,a) \sim \mathcal{D}}[\text{div}(m(s,a), m_{\mathcal{D}}(s,a))] \leq \epsilon\}, \tag{15}$$

where popular choices of the divergence $\text{div}(\cdot, \cdot)$ include total variation (TV) distance and KL divergence.

**Uncertainty-penalized pessimism: soft robust MDP.** Another line of model-based methods incorporates explicit uncertainty penalties into value updates (Yu et al., 2020; Kidambi et al., 2020; Jeong et al., 2023; Sun et al., 2023). For imagined transitions $(\hat{s}, \hat{a}, \hat{r}, \hat{d}, \hat{s}')$, the pessimistic update takes the form:

$$Q^{\text{MB}}(\hat{s}, \hat{a}) \leftarrow \hat{r} - \lambda U(\hat{s}, \hat{a}) + \gamma(1 - \hat{d}) \max_{\hat{a}' \in \mathcal{A}} Q^{\text{MB}}(\hat{s}', \hat{a}'), \tag{16}$$

where $U : \mathcal{S} \times \mathcal{A} \to \mathbb{R}^+$ is an uncertainty measure based on dataset $\mathcal{D}$ and learned world models, and $\lambda > 0$ controls the degree of pessimism. Similar uncertainty penalties have also been incorporated into model-free value functions (Bai et al., 2022; An et al., 2021).

We now provide a connection between Eq. 16 and the *soft robust MDP* framework of (Zhang et al., 2024b, Section 5). While our derivation follows a similar line to theirs, we include it here to be self-contained. Consider a robust MDP with a *policy-dependent* uncertainty set:

$$\max_{\pi} \min_{m} J(\pi, m) \quad \text{s.t.} \quad \mathbb{E}_{(s,a) \sim (m,\pi)}[\text{div}(m(s,a), m^*(s,a))] \leq \epsilon, \tag{17}$$

where the divergence constraint describes the uncertainty set, taken in expectation under occupancy measure induced by $m$ and $\pi$. We use Lagrangian relaxation with a coefficient $\alpha \geq 0$ to transform the problem into a soft robust MDP:

$$\max_{\pi} \min_{m} J(\pi, m) + \alpha \mathbb{E}_{(s,a) \sim (m,\pi)}[\text{div}(m(s,a), m^*(s,a))]. \tag{18}$$

In dynamic programming form, for a given $(s, a)$ pair, the inner optimization becomes

$$\min_{m(\cdot|s,a)} \mathbb{E}_{s' \sim m(\cdot|s,a)}\left[V^{\text{MB}}(s')\right] + \alpha \, \text{div}(m(\cdot \mid s,a), m^*(\cdot \mid s,a)), \tag{19}$$

where $V^{\text{MB}}$ is the policy's state-value function. This inner problem can be transformed by duality, depending on the choice of divergence. For the KL divergence, one can apply Donsker and Varadhan's formula (Donsker & Varadhan, 1975)[6] to state that Eq. 19 is equivalent to

$$-\alpha \log\left(\mathbb{E}_{s' \sim m^*(\cdot|s,a)}\left[\exp\left(-\frac{V^{\text{MB}}(s')}{\alpha}\right)\right]\right) = \mathbb{E}_{s' \sim m^*}[V^{\text{MB}}(s')] - \frac{1}{2\alpha}\text{Var}_{m^*}[V^{\text{MB}}(s')] + O\left(\frac{1}{\alpha^3}\right), \tag{20}$$

where we use cumulant expansion.[7]

Therefore, the corresponding soft-robust Bellman optimality equation (Zhang et al., 2024b, Equation 15), ignoring higher-order terms, becomes

$$Q^{\text{MB}}(s, a) = R^*(s, a) - \frac{\gamma}{2\alpha}\text{Var}_{s' \sim P^*}[\max_{a'} Q^{\text{MB}}(s', a')] + \gamma \mathbb{E}_{s' \sim P^*}\left[\max_{a'} Q^{\text{MB}}(s', a')\right] \tag{21}$$

---

[6]For any probability distributions $x, x^* \in \Delta^k$ (the $k$-dimensional simplex), any vector $y \in \mathbb{R}^k$, and $\alpha > 0$, the duality formula is $-\alpha \log(\langle x^*, \exp(-y/\alpha)\rangle) = \min_x \langle x, y \rangle + \alpha \, \text{KL}(x \mid\mid x^*)$.

[7]For a random variable $X$, $\log(\mathbb{E}[\exp(tX)]) = t\mathbb{E}[X] + \frac{t^2}{2}\text{Var}[X] + O(t^3)$. We substitute $t = -1/\alpha$.

In practice, model-based RL methods approximate $m^* = (R^*, P^*)$ with an ensemble of learned models trained on $\mathcal{D}$. While the variance term in Eq. 21 reflects the aleatoric uncertainty of the true dynamics, ensemble-based variance also incorporates epistemic uncertainty by the law of total variance, thus blending both. Prior work has interpreted the penalty as aleatoric (Yu et al., 2020), epistemic (Sun et al., 2023), or both (Rigter et al., 2023); here we focus on its epistemic interpretation. Accordingly, the ensemble variance provides a practical surrogate for the penalty in Eq. 16.

## C UNDERSTANDING ON BAYESIAN OFFLINE RL

### C.1 CONNECTION BETWEEN BAYESIANISM AND PARTIAL OBSERVABILITY

**Epistemic POMDP as a special class of POMDP.** As noted by Ghosh et al. (2022, Appendix A), the Bayesian objective (Eq. 4) can be cast as a POMDP (Cassandra et al., 1994). We adapt their proof here. The POMDP's state space is $\mathcal{S}^+ = \mathcal{S} \times \mathfrak{M}_{\mathcal{D}}$ with the same action space $\mathcal{A}$, where $\mathfrak{M}_{\mathcal{D}} = \mathrm{supp}(\mathbb{P}_{\mathcal{D}})$. The joint reward–transition function in the POMDP is

$$\mathbb{P}(r_{t+1}, s_{t+1}^+ \mid s_t^+, a_t) = \mathbb{1}(m_{t+1} = m_t) m_t(r_{t+1}, s_{t+1} \mid s_t, a_t),$$

where the initial state is $s_0^+ := (s_0, m_0)$ with $s_0 \sim \rho$ and $m_0 \sim \mathbb{P}_{\mathcal{D}}$. It is partially observable because the agent only observes $s_t \in s_t^+$, while the model $m_t \equiv m_0 \in \mathfrak{M}_{\mathcal{D}}$ remains hidden but fixed throughout each episode.

### C.2 DIMINISHING BOOTSTRAPPED ERROR IN LONG-HORIZON ROLLOUTS

We formalize the insight in Eq. 6 using a result adapted from Sims et al. (2024, Proposition 1).

Consider a rollout generated by a deterministic world model $m$ and a deterministic policy $\pi$,

$$\tau = (h_t, \hat{a}_t, \hat{r}_{t+1}, \hat{s}_{t+1}, \hat{a}_{t+1}, \dots, \hat{s}_{t+H}),$$

where the initial history $h_t$ is drawn from the dataset $\mathcal{D}$, $\hat{a}_{t+j} = \pi(\hat{h}_{t+j})$, $(\hat{r}_{t+j+1}, \hat{s}_{t+j+1}) = m(\hat{s}_{t+j}, \hat{a}_{t+j})$ and $\hat{h}_t = h_t$.

Assume a tabular policy evaluation setting and let $Q^\pi$ denote the exact value function of policy $\pi$ under the MDP $m$. Define the Bellman evaluation operator $\mathcal{T}$: for a value function $Q$,

$$\mathcal{T}Q(\hat{h}_{t+j}, \hat{a}_{t+j}) := \hat{r}_{t+j+1} + \gamma Q(\hat{h}_{t+j+1}, \pi(\hat{h}_{t+j+1})).$$

We perform the value update along the imagined rollout backward in time. Let $Q_0$ be the initial value function prior to value update and $Q_j$ be the value function at $j$-th iteration. For $j = 1, \dots, H$, define $Q_j(\hat{h}_{t+H-j}, \hat{a}_{t+H-j}) \approx \mathcal{T}Q_{j-1}(\hat{h}_{t+H-j}, \hat{a}_{t+H-j})$, with a per-step **TD error** bounded by:

$$|Q_j(\hat{h}_{t+H-j}, \hat{a}_{t+H-j}) - \mathcal{T}Q_{j-1}(\hat{h}_{t+H-j}, \hat{a}_{t+H-j})| \le \delta_j.$$

Following Sims et al. (2024), assume that $\hat{h}_{t+H}$ is an **edge-of-reach** history, i.e., a history used as Bellman targets but never itself updated. The **bootstrapped error** at this truncated point is

$$\epsilon := |Q_0(\hat{h}_{t+H}, \pi(\hat{h}_{t+H})) - Q^\pi(\hat{h}_{t+H}, \pi(\hat{h}_{t+H}))|.$$

We can decompose the total error for $Q_j$: for $j = 1, \dots, H$,

$$
\begin{aligned}
\xi_j &:= |Q_j(\hat{h}_{t+H-j}, \hat{a}_{t+H-j}) - Q^\pi(\hat{h}_{t+H-j}, \hat{a}_{t+H-j})| \\
&\le |Q_j(\hat{h}_{t+H-j}, \hat{a}_{t+H-j}) - \mathcal{T}Q_{j-1}(\hat{h}_{t+H-j}, \hat{a}_{t+H-j})| \\
&\quad + |\mathcal{T}Q_{j-1}(\hat{h}_{t+H-j}, \hat{a}_{t+H-j}) - Q^\pi(\hat{h}_{t+H-j}, \hat{a}_{t+H-j})| \\
&\le \delta_j + |\gamma Q_{j-1}(\hat{h}_{t+H-j+1}, \hat{a}_{t+H-j+1}) - \gamma Q^\pi(\hat{h}_{t+H-j+1}, \hat{a}_{t+H-j+1})| \\
&= \delta_j + \gamma \xi_{j-1}.
\end{aligned}
\tag{22}
$$

Unrolling the recursion and using the fact that $\xi_0 = \epsilon$, the value at the final iteration is bounded by:

$$|Q_H(h_t, \hat{a}_t) - Q^\pi(h_t, \hat{a}_t)| = \xi_H \le \sum_{j=0}^{H-1} \gamma^j \delta_j + \gamma^H \epsilon. \tag{23}$$

Thus, as long as the bootstrapped error $\epsilon$ dominates the intermediate TD errors, increasing rollout horizon $H$ exponentially suppresses the dominant source of error. In the model-based setting, these intermediate errors also include model prediction errors on rewards. Therefore, this formalizes our insight in Eq. 6.

# D   FORMAL EXISTENCE PROOF OF THE ADVANTAGE OF BAYESIANISM

In this section, we compare the conservative and Bayesian principles in offline RL and highlight conditions under which the Bayesian approach yields provable advantages. After restating the conservative objective and its reliance on data coverage, we introduce a relaxed notion of coverage from a Bayesian view. This allows us to lower-bound the performance gap between Bayesian and robust solutions. We begin by defining the optimal policies in their respective objectives:

$$\pi^*(m^*) \in \operatorname*{argmax}_\pi J(\pi, m^*), \qquad\qquad \text{(ideal policy)}$$

$$\pi^*(\mathfrak{M}_\mathcal{D}) \in \operatorname*{argmax}_\pi J(\pi; \mathfrak{M}_\mathcal{D}), \qquad\qquad \text{(robust-optimal policy)}$$

$$\pi^*(\mathbb{P}_\mathcal{D}) \in \operatorname*{argmax}_\pi \mathbb{E}_{m\sim\mathbb{P}_\mathcal{D}}[J(\pi, m)]. \qquad\qquad \text{(Bayes-optimal policy)}$$

We then introduce the notion of *robust sub-optimality gap*, measuring the gap of the robust-optimal policy relative to the ideal one:

$$S_{\mathfrak{M}_\mathcal{D}}(m^*) := J(\pi^*(m^*), m^*) - J(\pi^*(\mathfrak{M}_\mathcal{D}), m^*). \qquad (24)$$

To assess when $\pi^*(\mathfrak{M}_\mathcal{D})$ is competitive, one seeks an upper bound on this gap. Its tightness depends on dataset quality: the closer the data coverage is to $\pi^*(m^*)$, the smaller the gap. Prior work formalizes this dependence via coverage assumptions (Uehara & Sun, 2022; Li et al., 2024). We unify these notions through the following definition.

## D.1   COMPARISON ON CONCENTRABILITY COEFFICIENTS

In this subsection, we define the robust and Bayesian concentrability coefficients and derive the numerical relationship between the two.

**Definition 1** (Model-dependent concentrability). *Given an MDP model $m$ and a policy $\pi$, let $d_m^\pi$ be the state-action occupancy measure of $\pi$ on $m$, and let $\beta \in \Delta(\mathcal{S} \times \mathcal{A})$ be the offline distribution induced by the dataset $\mathcal{D}$ on $m^*$. The concentrability coefficient of $\pi$ in MDP $m$ under offline distribution $\beta$ is defined as:*

$$\mathcal{C}(\pi, m) := \frac{\mathbb{E}_{(s,a)\sim d_m^\pi}\big[\mathrm{TV}(m(s,a), m^*(s,a))^2\big]}{\mathbb{E}_{(s,a)\sim\beta}\big[\mathrm{TV}(m(s,a), m^*(s,a))^2\big]}, \qquad (25)$$

*where $\mathrm{TV}$ denotes the total variation distance between distributions.*

Intuitively, $\mathcal{C}(\pi, m)$ quantifies the mismatch between $m$ and $m^*$ under the policy distribution versus the dataset distribution.

We then extend the *model-based concentrability* of Uehara & Sun (2022, Definition 1), originally defined only for transition dynamics, to also incorporate the reward function. We refer to this generalization as robust concentrability.

**Definition 2** (**Robust concentrability** (Uehara & Sun, 2022)). *Let $\mathfrak{M}_\mathcal{D}$ be a realizable hypothesis class of models consistently built from the dataset $\mathcal{D}$,*

$$\mathcal{C}(\pi) := \sup_{m\in\mathfrak{M}_\mathcal{D}} \mathcal{C}(\pi, m). \qquad (26)$$

If $m^* \in \mathfrak{M}_\mathcal{D}$, the robust concentrability is upper bounded by the classic density-ratio-based concentrability, as shown in Uehara & Sun (2022, Section 4):

$$\mathcal{C}(\pi) \le \sup_{(s,a)\in\mathcal{S}\times\mathcal{A}} \frac{d_{m^*}^\pi(s, a)}{\beta(s, a)}.$$

We now consider an even *weaker* notion of coverage by extending model-dependent concentrability with a Bayesian posterior over models.

**Definition 3** (**Bayesian concentrability**).

$$\mathcal{C}_{Bayes}(\pi) := \frac{\mathbb{E}_{m \sim \mathbb{P}_{\mathcal{D}}}\left[\mathbb{E}_{(s,a) \sim d_m^\pi}\left[\mathrm{TV}(m(s,a), m^*(s,a))^2\right]\right]}{\mathbb{E}_{m \sim \mathbb{P}_{\mathcal{D}}}\left[\mathbb{E}_{(s,a) \sim \beta}[\mathrm{TV}(m(s,a), m^*(s,a))^2]\right]}. \tag{27}$$

We now show that Bayesian concentrability is always upper bounded by its robust counterpart.

**Proposition 2** (Bayesian concentrability is upper-bounded by robust concentrability). *Assume that* $\mathbb{E}_{m \sim \mathbb{P}_{\mathcal{D}}}[g(m)] > 0$, *then*

$$C_{Bayes}(\pi) \leq \sup_{m \in \mathrm{supp}(\mathbb{P}_{\mathcal{D}})} \mathcal{C}(\pi, m).$$

*Proof.* Denote

$$f(m) := \mathbb{E}_{(s,a) \sim d_m^\pi}\left[\mathrm{TV}(m(s,a), m^*(s,a))^2\right],$$
$$g(m) := \mathbb{E}_{(s,a) \sim \beta}\left[\mathrm{TV}(m(s,a), m^*(s,a))^2\right],$$

so that

$$\mathcal{C}(\pi, m) = \frac{f(m)}{g(m)}, \quad \mathcal{C}_{\mathrm{Bayes}}(\pi) := \frac{\mathbb{E}_{m \sim \mathbb{P}_{\mathcal{D}}}[f(m)]}{\mathbb{E}_{m \sim \mathbb{P}_{\mathcal{D}}}[g(m)]}.$$

$$\mathrm{LHS} = \frac{\mathbb{E}_{m \sim \mathbb{P}_{\mathcal{D}}}\left[g(m)\frac{f(m)}{g(m)}\right]}{\mathbb{E}_{m \sim \mathbb{P}_{\mathcal{D}}}[g(m)]} \leq \sup_{m \in \mathrm{supp}(\mathbb{P}_{\mathcal{D}})} \frac{f(m)}{g(m)} = \mathrm{RHS},$$

by the elementary inequality $\sum_i w_i x_i \leq \sum_i w_i \max_j x_j = (\sup_j x_j) \sum_i w_i$ for $w \geq 0$. $\square$

The next example sharpens this bound to a strict inequality, making the Bayesian coverage assumption $\mathcal{C}_{\mathrm{Bayes}}(\pi) < \infty$ strictly weaker than its robust analogue $\mathcal{C}(\pi) < \infty$.

**Example 1** (Strictness of Bayesian concentrability bound). *Let $\beta$ have incomplete coverage over $\mathcal{S} \times \mathcal{A}$. Following the notations from the proof of Proposition 2, consider two models:*

1. *Model $m_1$ is a small perturbation of $m^*$ on all of $\mathcal{S} \times \mathcal{A}$. Then, the numerator $f(m_1) < \infty$, the denominator $g(m_1) > 0$, making $\mathcal{C}(\pi, m_1) < \infty$.*

2. *Model $m_2$ is equivalent to $m^*$ on $\mathrm{supp}(\beta)$, i.e., $m_2(s,a) = m^*(s,a), \forall (s,a) \in \mathrm{supp}(\beta)$, but there exists an off-support pair $(s^\dagger, a^\dagger) \notin \mathrm{supp}(\beta)$ with $d_{m_2}^\pi(s^\dagger, a^\dagger) > 0$ and $\mathrm{TV}(m_2(s^\dagger, a^\dagger), m^*(s^\dagger, a^\dagger)) > 0$. In that case, $g(m_2) = 0, f(m_2) > 0$, so $\mathcal{C}(\pi, m_2) = \infty$.*

*If the posterior $\mathbb{P}_{\mathcal{D}}$ assigns weights $\mathbb{P}_{\mathcal{D}}(m_1) = 1 - \varepsilon$ and $\mathbb{P}_{\mathcal{D}}(m_2) = \varepsilon$ with any $\varepsilon \in (0, 1)$, then*

$$\mathbb{E}_{m \sim \mathbb{P}_{\mathcal{D}}}[f(m)] = (1 - \varepsilon)f(m_1) + \varepsilon f(m_2) < \infty, \quad \mathbb{E}_{m \sim \mathbb{P}_{\mathcal{D}}}[g(m)] = (1 - \varepsilon)g(m_1) > 0,$$

*and $\mathcal{C}_{Bayes}(\pi) < \infty$, while $\sup_{m \in \mathrm{supp}(\mathbb{P}_{\mathcal{D}})} \mathcal{C}(\pi, m) = \infty$.*

## D.2 THEOREM 1 AND PROOF

To formalize the benefit of taking a Bayesian approach over a conservative one, we define the *Bayesian sub-optimality gap* as:

$$S_{\mathbb{P}_{\mathcal{D}}}(m^*) = J(\pi^*(m^*), m^*) - J(\pi^*(\mathbb{P}_{\mathcal{D}}), m^*), \tag{28}$$

which we aim to compare with the robust gap via their difference:

$$\Delta_{\mathbb{P}_{\mathcal{D}}, \mathfrak{M}_{\mathcal{D}}}(m^*) := S_{\mathfrak{M}_{\mathcal{D}}}(m^*) - S_{\mathbb{P}_{\mathcal{D}}}(m^*) = J(\pi^*(\mathbb{P}_{\mathcal{D}}), m^*) - J(\pi^*(\mathfrak{M}_{\mathcal{D}}), m^*). \tag{29}$$

The theorem below establishes a lower bound on this difference.

**Theorem 1.** *Assume that $\gamma > 1/2$. Then, we can construct a dataset $\mathcal{D}$, a set of MDPs $\mathfrak{M}_{\mathcal{D}}$, and a posterior $\mathbb{P}_{\mathcal{D}}$ such that $m^* \in \mathrm{supp}(\mathbb{P}_{\mathcal{D}}) \subseteq \mathfrak{M}_{\mathcal{D}}$, and for some $\delta_0 > 0$, it holds with probability at least $\delta_0$ that,*

$$\Delta_{\mathbb{P}_{\mathcal{D}}, \mathfrak{M}_{\mathcal{D}}}(m^*) > \frac{8r_{\max}}{(1-\gamma)^2} \sqrt{\frac{\ln(|\mathfrak{M}_{\mathcal{D}}|/\delta_0)}{|\mathcal{D}|}} \left(\sqrt{\mathcal{C}(\pi^*(\mathfrak{M}_{\mathcal{D}}))} - \sqrt{\max(\mathcal{C}_{Bayes}(\pi^*(m^*)), \mathcal{C}_{Bayes}(\pi^*(\mathbb{P}_{\mathcal{D}})))}\right).$$

*Proof sketch.* The proof proceeds in two steps, each being established in Lemma 1 and Lemma 2 respectively. (1) We upper-bound the Bayesian gap $S_{\mathbb{P}_\mathcal{D}}(m^*)$ (Lemma 1). This bound makes explicit the "price of Bayesianism": coverage adequate for the ideal policy $\pi^*(m^*)$ alone does not suffice. We also need reasonable coverage for the Bayes-optimal policy, i.e., $\mathcal{C}_{\text{Bayes}}(\pi^*(\mathbb{P}_\mathcal{D})) < \infty$, even though posterior averaging makes $\mathcal{C}_{\text{Bayes}}(\cdot)$ more likely to be finite. (2) We lower-bound the robust gap $S_{\mathfrak{M}_\mathcal{D}}(m^*)$ (Lemma 2), by constructing an MDP with inadequate frequentist coverage relative to the behavior policy. $\square$

*Remark* 1. This theorem shows that when $\mathcal{C}(\pi^*(\mathfrak{M}_\mathcal{D})) > \max(\mathcal{C}_{\text{Bayes}}(\pi^*(m^*)), \mathcal{C}_{\text{Bayes}}(\pi^*(\mathbb{P}_\mathcal{D})))$, the Bayes-optimal policy performs better than the robust-optimal policy in the true MDP. By construction, the lower bound in Lemma 2 holds when $\epsilon$ – the parametric gap between two well-chosen MDPs – is smaller than $c \cdot \mathcal{C}(\pi^*(\mathfrak{M}_\mathcal{D}))/|\mathcal{D}|$. This is more likely to occur when the dataset is small or when the concentrability $\mathcal{C}(\pi^*(\mathfrak{M}_\mathcal{D}))$ is large. In other words, adopting a frequentist (or conservative) approach rather than Bayesianism implies that the dataset does not provide sufficient statistical information when: (1) it is small, or (2) its coverage differs from that of the ground truth optimal policy. Those are the two features measuring the "quality" of an offline dataset. The Bayesian approach potentially yields better performance than conservatism because, for a given amount of data, the Bayesian concentrability is smaller than the robust one, making the lower bound irrelevant for the Bayesian setting.

**Lemma 1** (**Upper bound on Bayesian sub-optimality gap**). *Assume that $m^* \in \text{supp}(\mathbb{P}_\mathcal{D}) \subseteq \mathfrak{M}_\mathcal{D}$. Then, for any $\delta \in (0,1)$, with probability at least $1 - \delta$,*

$$S_{\mathbb{P}_\mathcal{D}}(m^*) \le \frac{8 r_{\max}}{(1-\gamma)^2} \sqrt{\frac{\ln(|\mathfrak{M}_\mathcal{D}|/\delta)}{|\mathcal{D}|}} \sqrt{\max(\mathcal{C}_{Bayes}(\pi^*(m^*)), \mathcal{C}_{Bayes}(\pi^*(\mathbb{P}_\mathcal{D})))}.$$

*Proof of Lemma 1.* By definition of the Bayesian sub-optimality gap (Eq. 28),

$$S_{\mathbb{P}_\mathcal{D}}(m^*)$$
$$= J(\pi^*(m^*), m^*) - J(\pi^*(\mathbb{P}_\mathcal{D}), m^*)$$
$$= J(\pi^*(m^*), m^*) - \mathbb{E}_{m \sim \mathbb{P}_\mathcal{D}}[J(\pi^*(m^*), m)] + \mathbb{E}_{m \sim \mathbb{P}_\mathcal{D}}[J(\pi^*(m^*), m)]$$
$$\quad - \mathbb{E}_{m \sim \mathbb{P}_\mathcal{D}}[J(\pi^*(\mathbb{P}_\mathcal{D}), m)] + \mathbb{E}_{m \sim \mathbb{P}_\mathcal{D}}[J(\pi^*(\mathbb{P}_\mathcal{D}), m)] - J(\pi^*(\mathbb{P}_\mathcal{D}), m^*)$$
$$\le (J(\pi^*(m^*), m^*) - \mathbb{E}_{m \sim \mathbb{P}_\mathcal{D}}[J(\pi^*(m^*), m)]) + (\mathbb{E}_{m \sim \mathbb{P}_\mathcal{D}}[J(\pi^*(\mathbb{P}_\mathcal{D}), m)] - J(\pi^*(\mathbb{P}_\mathcal{D}), m^*)),$$

where the last inequality holds by definition of $\pi^*(\mathbb{P}_\mathcal{D})$ being a solution of $\max_\pi \mathbb{E}_{m \sim \mathbb{P}_\mathcal{D}}[J(\pi, m)]$. The first term measures how the performance of the ideal policy $\pi^*(m^*)$ under the true MDP deviates from its Bayesian average. The second term measures the analogous deviation for the Bayes-optimal policy $\pi^*(\mathbb{P}_\mathcal{D})$. We aim to upper-bound each of these regret terms.

**Step 1: Upper bound** $J(\pi^*(m^*), m^*) - \mathbb{E}_{m \sim \mathbb{P}_\mathcal{D}}[J(\pi^*(m^*), m)]$**.**

$$J(\pi^*(m^*), m^*) - \mathbb{E}_{m \sim \mathbb{P}_\mathcal{D}}[J(\pi^*(m^*), m)] = \mathbb{E}_{m \sim \mathbb{P}_\mathcal{D}}[J(\pi^*(m^*), m^*) - J(\pi^*(m^*), m)]$$
$$\le \frac{2 r_{\max}}{(1-\gamma)^2} \mathbb{E}_{m \sim \mathbb{P}_\mathcal{D}}\left[ \mathbb{E}_{(s,a) \sim d_m^{\pi^*(m^*)}}[\text{TV}(m(s,a), m^*(s,a))] \right],$$

according to Lemma 4. Therefore, by Jensen's inequality,

$$J(\pi^*(m^*), m^*) - \mathbb{E}_{m \sim \mathbb{P}_\mathcal{D}}[J(\pi^*(m^*), m)] \le \frac{2 r_{\max}}{(1-\gamma)^2} \mathbb{E}_{m \sim \mathbb{P}_\mathcal{D}}\left[ \sqrt{\mathbb{E}_{(s,a) \sim d_m^{\pi^*(m^*)}}[\text{TV}(m(s,a), m^*(s,a))^2]} \right]$$
$$\le \frac{2 r_{\max}}{(1-\gamma)^2} \sqrt{\mathbb{E}_{m \sim \mathbb{P}_\mathcal{D}}\left[ \mathbb{E}_{(s,a) \sim d_m^{\pi^*(m^*)}}[\text{TV}(m(s,a), m^*(s,a))^2] \right]},$$

and by construction of the Bayesian concentrability coefficient (Eq. 27):

$$J(\pi^*(m^*), m^*) - \mathbb{E}_{m \sim \mathbb{P}_\mathcal{D}}[J(\pi^*(m^*), m)]$$
$$\le \frac{2 r_{\max}}{(1-\gamma)^2} \sqrt{\mathcal{C}_{\text{Bayes}}(\pi^*(m^*)) \mathbb{E}_{m \sim \mathbb{P}_\mathcal{D}}\left[ \mathbb{E}_{(s,a) \sim \beta}[\text{TV}(m(s,a), m^*(s,a))^2] \right]}.$$

Finally, since by construction $\mathbb{P}_\mathcal{D}$ is supported only on models that achieve MLE under $\mathcal{D}$, every $m \in \text{supp}(\mathbb{P}_\mathcal{D})$ satisfies the PAC bound of Lemma 3, yielding

$$J(\pi^*(m^*), m^*) - \mathbb{E}_{m \sim \mathbb{P}_\mathcal{D}}[J(\pi^*(m^*), m)] \le \frac{4 r_{\max}}{(1-\gamma)^2} \sqrt{\mathcal{C}_{\text{Bayes}}(\pi^*(m^*))} \sqrt{\frac{\ln(|\mathfrak{M}_\mathcal{D}|/\delta)}{|\mathcal{D}|}}.$$

**Step 2: Upper bound** $\mathbb{E}_{m\sim\mathbb{P}}[J(\pi^*(\mathbb{P}_\mathcal{D}),m)] - J(\pi^*(\mathbb{P}_\mathcal{D}),m^*)$**.** We use the same reasoning as above, but applied this time to the Bayes-optimal policy, and deduce the following bound:

$$\mathbb{E}_{m\sim\mathbb{P}}[J(\pi^*(\mathbb{P}_\mathcal{D}),m)] - J(\pi^*(\mathbb{P}_\mathcal{D}),m^*) \leq \frac{4r_{\max}}{(1-\gamma)^2}\sqrt{\mathcal{C}_{\text{Bayes}}(\pi^*(\mathbb{P}_\mathcal{D}))}\sqrt{\frac{\ln(|\mathfrak{M}_\mathcal{D}|/\delta)}{|\mathcal{D}|}}.$$

We finally obtain:

$$S_{\mathbb{P}_\mathcal{D}}(m^*) \leq \frac{4r_{\max}}{(1-\gamma)^2}\sqrt{\frac{\ln(|\mathfrak{M}_\mathcal{D}|/\delta)}{|\mathcal{D}|}}\left(\sqrt{\mathcal{C}_{\text{Bayes}}(\pi^*(m^*))} + \sqrt{\mathcal{C}_{\text{Bayes}}(\pi^*(\mathbb{P}_\mathcal{D}))}\right)$$

$$\leq \frac{8r_{\max}}{(1-\gamma)^2}\sqrt{\frac{\ln(|\mathfrak{M}_\mathcal{D}|/\delta)}{|\mathcal{D}|}}\sqrt{\max(\mathcal{C}_{\text{Bayes}}(\pi^*(m^*)), \mathcal{C}_{\text{Bayes}}(\pi^*(\mathbb{P}_\mathcal{D})))}.$$

$\square$

**Lemma 2** (**Existence of a lower bound on robust sub-optimality gap**). *Assume that $\gamma > 1/2$. We can construct an MDP instance $m^* = m_1$ inducing an offline dataset $\mathcal{D}$ and a set of models $\mathfrak{M} = \{m_{-1}, m_1\}$, such that the optimal robust policy $\pi^*(\mathfrak{M})$ satisfies w.p. at least $\delta_0$:*

$$J(\pi^*(m^*),m^*) - J(\pi^*(\mathfrak{M}),m^*) > \frac{8r_{\max}}{(1-\gamma)^2}\sqrt{\frac{\ln(|\mathfrak{M}|/\delta_0)}{|\mathcal{D}|}}\sqrt{\mathcal{C}(\pi^*(\mathfrak{M}))}.$$

*Proof of Lemma 2.* Similarly as Li et al. (2024), we build two MDPs (here, two sequential two-armed bandits) and a behavior policy such that the conservative value estimated from the resulting dataset is far from the optimal return.

**The example.** Consider two sequential bandits $\mathfrak{M} := \{m_{-1}, m_1\}$ such that $m^* = m_1$ is the ground-truth model. We parameterize both by $\theta \in \{-1, 1\}$. They share the same action space $\mathcal{A} := \{-1, 1\}$, the same reward for the negative action, but different reward for the positive action, namely, $R_\theta(-1) \sim \mathcal{B}(1/2)$ while $R_\theta(1) \sim \mathcal{B}(1/2 + \theta\epsilon)$ with $\epsilon > 0$ that will be determined later. Clearly, the ideal policy for each MDP is $\pi_\theta^* = \theta$. With notational abuse, let the behavior policy be $\beta(1) = \beta = 1 - \beta(-1)$ from which we collect $|\mathcal{D}|$ i.i.d. samples. As will be justified in the following, we set $\beta = \frac{(1-\gamma)^4}{64r_{\max}^2\mathcal{C}(\pi^*(\mathfrak{M}))} \in [0, 1]$.

**Relation to the bandit example in Sec. 3.** Here, action $-1$ plays the role of the sampling action with Bernoulli parameter $1/2$, and action $1$, the "uncovered action" in the skewed bandit dataset of Sec. 3. We deliberately avoid setting $\beta(1) = 0$, since that would violate the data coverage assumption and make $\mathcal{C}(\pi^*(\mathfrak{M}))$ infinite.[8] Yet, the dataset is still skewed because it can be generated from $m_1$ under a suboptimal policy (but close-to-optimal policy for $m_{-1}$). In that case, it is statistically hard to identify the correct model *when $\epsilon$ is too small*, which is how we choose it for the lower bound to hold.

**Proof.** With a slight abuse of notation, denote a policy by $\pi := \pi(1) = 1 - \pi(-1)$. The discounted value may be expressed according to the underlying MDP as:

$$J(\pi, \theta) = \sum_{t=0}^{\infty}\gamma^t\left(\pi\cdot(1/2 + \theta\epsilon) + (1-\pi)\cdot 1/2\right) = \sum_{t=0}^{\infty}\gamma^t\left(\frac{1}{2} + \theta\epsilon\pi\right) = \frac{1 + 2\theta\epsilon\pi}{2(1-\gamma)}.$$

For each model $\theta \in \{-1, 1\}$, the suboptimality gap of a policy $\pi$ is thus:

$$\delta^\pi(1) = \frac{1 + 2\epsilon}{2(1-\gamma)} - \frac{1 + 2\epsilon\pi}{2(1-\gamma)} = \frac{2\epsilon(1-\pi)}{2(1-\gamma)} = \frac{\epsilon(1-\pi)}{1-\gamma},$$

$$\delta^\pi(-1) = \frac{1}{2(1-\gamma)} - \frac{1 - 2\epsilon\pi}{2(1-\gamma)} = \frac{\epsilon\pi}{1-\gamma}.$$

More synthetically, and since $\gamma \in [0, 1)$, we get:

$$\delta^\pi(\theta) = \frac{\epsilon}{1-\gamma}(1 - \pi(\theta)). \tag{30}$$

---

[8]In the limit $\mathcal{C}(\pi^*(\mathfrak{M})) \to \infty$, we recover the example in Sec. 3.

By contradiction, suppose that we can find a policy estimate from the dataset such that:

$$\mathbb{P}_\theta(J(\pi^*(\theta), \theta) - J(\hat{\pi}, \theta) \leq \epsilon) = \mathbb{P}_\theta(\delta^{\hat{\pi}}(\theta) \leq \epsilon) \geq \frac{7}{8}.$$

Then, from Eq. 30, we should have with probability greater than $7/8$ that:

$$\frac{\epsilon}{1-\gamma}(1 - \hat{\pi}(\theta)) \leq \epsilon \iff \hat{\pi}(\theta)\frac{\epsilon}{1-\gamma} \geq \frac{\epsilon}{1-\gamma} - \epsilon \iff \hat{\pi}(\theta) \geq 1 - (1-\gamma) = \gamma.$$

By assumption, $\gamma > 1/2$. If the above statement were true, then we could construct the following estimator $\hat{\theta}$ of $\theta$ based on $\hat{\pi}$:

$$\hat{\theta} = \operatorname*{argmax}_{a \in \{-1,1\}} \hat{\pi}(a) \tag{31}$$

and thus, $\mathbb{P}_\theta(\hat{\theta} = \theta) = \mathbb{P}_\theta(\hat{\pi}(\theta) > 1/2) \geq \mathbb{P}_\theta(\hat{\pi}(\theta) \geq \gamma) \geq 7/8$.

We analyze the hypothesis testing of identifying the true MDP given the generated data. Formally, let the test $\phi(\mathcal{D}) = 0$ mean "decide $m^* = m_{-1}$" and $\phi(\mathcal{D}) = 1$ mean "decide $m^* = m_1$". Consider the minimax probability of error $p_e$ between $m_{-1}$ and $m_1$:

$$p_e = \inf_\phi \max(\mathbb{P}_{-1}(\phi(\mathcal{D}) \neq 0), \ \mathbb{P}_1(\phi(\mathcal{D}) \neq 1)),$$

where $\mathbb{P}_\theta(\mathcal{D})$ denotes the sampling distribution of $\mathcal{D}$ under model $\theta$. By (Bickel et al., 2009)[Thm. 2.2], the following lower bound holds:

$$p_e \geq \frac{1}{4} \exp(-\text{KL}(\mathbb{P}_{-1}(\mathcal{D}) \,||\, \mathbb{P}_1(\mathcal{D}))). \tag{32}$$

Each data sample $\mathcal{D}_i$ is generated according to a mixture of two Bernoullis $\mathcal{D}_i \sim \beta \mathcal{B}(1/2 + \theta\epsilon) + (1 - \beta)\mathcal{B}(1/2)$. Let $n_w$ be the number of 1-reward samples, i.e., successful events $\mathcal{D}_i = 1$ (respectively, $n_l$ the number of 0-reward samples). Then:

$$\mathbb{P}(\mathcal{D}_i = 1) = \beta\left(\frac{1}{2} + \theta\epsilon\right) + (1-\beta)\frac{1}{2} = \frac{1}{2} + \beta\theta\epsilon,$$

$$\mathbb{P}(\mathcal{D}_i = 0) = \beta\left(1 - \left(\frac{1}{2} + \theta\epsilon\right)\right) + (1-\beta)\left(1 - \frac{1}{2}\right) = \beta\left(\frac{1}{2} - \theta\epsilon\right) + \frac{1}{2} - \frac{\beta}{2} = \frac{1}{2} - \beta\theta\epsilon,$$

and the likelihood for the whole dataset is:

$$\mathbb{P}_\theta(\mathcal{D}) = \left(\frac{1}{2} + \beta\theta\epsilon\right)^{n_w}\left(\frac{1}{2} - \beta\theta\epsilon\right)^{n_l}. \tag{33}$$

Based on Eq. (33), we can compute the divergence:

$$\text{KL}(\mathbb{P}_1(\mathcal{D}) \,||\, \mathbb{P}_{-1}(\mathcal{D})) = n_w\text{KL}\left(\mathcal{B}\left(\frac{1}{2} + \beta\epsilon\right), \mathcal{B}\left(\frac{1}{2} - \beta\epsilon\right)\right) + n_l\text{KL}\left(\mathcal{B}\left(\frac{1}{2} - \beta\epsilon\right), \mathcal{B}\left(\frac{1}{2} + \beta\epsilon\right)\right),$$

by the additive property of KL between independent samples. Remarking that

$$\text{KL}\left(\mathcal{B}\left(\frac{1}{2} + \epsilon\beta\right), \mathcal{B}\left(\frac{1}{2} - \epsilon\beta\right)\right)$$

$$= \left(\frac{1}{2} + \epsilon\beta\right)\log\left(\frac{\frac{1}{2} + \epsilon\beta}{\frac{1}{2} - \epsilon\beta}\right) + \left(1 - \frac{1}{2} - \epsilon\beta\right)\log\left(\frac{1 - \frac{1}{2} - \epsilon\beta}{1 - \frac{1}{2} + \epsilon\beta}\right)$$

$$= \left(\frac{1}{2} + \epsilon\beta\right)\log\left(\frac{\frac{1}{2} + \epsilon\beta}{\frac{1}{2} - \epsilon\beta}\right) + \left(\frac{1}{2} - \epsilon\beta\right)\log\left(\frac{\frac{1}{2} - \epsilon\beta}{\frac{1}{2} + \epsilon\beta}\right)$$

$$= \text{KL}\left(\mathcal{B}\left(\frac{1}{2} - \epsilon\beta\right), \mathcal{B}\left(\frac{1}{2} + \epsilon\beta\right)\right),$$

it results that

$$\text{KL}(\mathbb{P}_{-1}(\mathcal{D}) \,||\, \mathbb{P}_1(\mathcal{D})) = (n_w + n_l)\text{KL}\left(\mathcal{B}\left(\frac{1}{2} + \epsilon\beta\right), \mathcal{B}\left(\frac{1}{2} - \epsilon\beta\right)\right)$$

$$= |\mathcal{D}| \left( \frac{1}{2} + \epsilon\beta \right) \log \left( \frac{\frac{1}{2} + \epsilon\beta}{\frac{1}{2} - \epsilon\beta} \right) + |\mathcal{D}| \left( \frac{1}{2} - \epsilon\beta \right) \log \left( \frac{\frac{1}{2} - \epsilon\beta}{\frac{1}{2} + \epsilon\beta} \right)$$

$$= |\mathcal{D}| \left( \frac{1}{2} + \epsilon\beta - \frac{1}{2} + \epsilon\beta \right) \log \left( \frac{\frac{1}{2} + \epsilon\beta}{\frac{1}{2} - \epsilon\beta} \right)$$

$$= 2|\mathcal{D}|\epsilon\beta \log \left( \frac{1 + 2\epsilon\beta}{1 - 2\epsilon\beta} \right).$$

For a small enough $\epsilon$, the series expansion of the logarithm term yields:

$$\log \left( \frac{1 + 2\epsilon\beta}{1 - 2\epsilon\beta} \right) = 2 \cdot 2\epsilon\beta + o((2\epsilon\beta)^2) = 4\epsilon\beta + o((\epsilon\beta)^2)$$

so that

$$\text{KL}(\mathbb{P}_{-1}(\mathcal{D}) \, \| \, \mathbb{P}_1(\mathcal{D})) = 2|\mathcal{D}|\epsilon\beta(4\epsilon\beta + o((\epsilon\beta)^2)) = 8|\mathcal{D}|(\epsilon\beta)^2 + o((\epsilon\beta)^3).$$

We can eventually deduce that for a small enough $\epsilon$:

$$\text{KL}(\mathbb{P}_{-1}(\mathcal{D}) \, \| \, \mathbb{P}_1(\mathcal{D})) \leq c|\mathcal{D}|\epsilon^2\beta. \tag{34}$$

The binary testing lower bound (32) together with Eq. 34 establishes that the misidentification probability $p_e$ is at least $1/8$ as long as

$$\exp\left( -c|\mathcal{D}|\epsilon^2\beta \right) \geq 1/2, \tag{35}$$

which is equivalent to $\epsilon \leq \sqrt{\frac{\ln(2)}{c|\mathcal{D}|\beta}}$.

To conclude, we have assumed that there is a policy estimate $\hat{\pi}$ such that $\mathbb{P}_\theta(\delta^{\hat{\pi}}(\theta) \leq \epsilon) \geq 7/8$, namely, $\mathbb{P}_{-1}(\delta^{\hat{\pi}}(-1) > \epsilon) < 1/8$ and $\mathbb{P}_1(\delta^{\hat{\pi}}(1) > \epsilon) < 1/8$. Then, in view of our previous arguments, the estimator $\hat{\theta}$ as defined in Eq. 31 must satisfy $\mathbb{P}_{-1}(\hat{\theta} \neq \theta) < 1/8$ and $\mathbb{P}_{-1}(\hat{\theta} \neq \theta) < 1/8$, which contradicts the misidentification lower-bound $p_e \geq 1/8$ when $\epsilon \leq \sqrt{\frac{\ln(2)}{c|\mathcal{D}|\beta}}$. Remarking that in our example, $|\mathfrak{M}| = 2$ and $\delta_0 = 1/8$, we set $c = \ln(2)/\ln(16)$ to conclude that any policy estimate $\hat{\pi}$ inevitably satisfies $\mathbb{P}_\theta(J(\pi^*(\theta), \theta) - J(\hat{\pi}, \theta) \geq \epsilon) \geq \frac{7}{8}$[9]. In particular, the statement holds for the optimal robust policy $\pi^*(\mathfrak{M})$. $\qquad\square$

### D.3 AUXILIARY LEMMAS FOR THEOREM 1

We first recall a standard PAC bound for maximum likelihood estimation (MLE), adapted from Agarwal et al. (2020a, Theorem 21).

**Lemma 3 (MLE PAC-bound).** *Let $\beta \in \Delta(\mathcal{S} \times \mathcal{A})$ be the offline distribution induced by $\mathcal{D}$, and*

$$\hat{m} = \underset{m \in \mathfrak{M}}{\arg\max} \, \mathbb{E}_{(s,a,r,s') \sim \mathcal{D}}[\log m(r, s' \mid s, a)]$$

*be the MLE model within a finite uncertainty set $\mathfrak{M}$. Suppose $m^* \in \mathfrak{M}$. Then, for any $\delta \in (0, 1)$, with probability at least $1 - \delta$:*

$$\mathbb{E}_{(s,a) \sim \beta}\left[ \text{TV}(\hat{m}(s,a), m^*(s,a))^2 \right] \leq \frac{2 \log(|\mathfrak{M}|/\delta)}{|\mathcal{D}|}.$$

We next establish a *general* simulation lemma below. Unlike the classic simulation lemma (Uehara & Sun, 2022, Lemma 9), which applies only to stationary policies and considers transition errors alone, our result (i) extends to history-dependent policies via general Bellman recursions, and (ii) incorporates discrepancies in both transition and reward functions.

---

[9]For that specific value $c = \ln(2)/\ln(16)$, any $\epsilon < \min\{\frac{1}{\beta}\sqrt{1/4 - 8\beta}, 1/(2\beta)\}$ would make the inequality (35) valid, as long as $\beta < 1/32$. Since $\mathcal{C}(\pi^*(\mathfrak{M})) \geq 1$ and $r_{\max} = 1$, condition $\beta < 1/32$ is automatically fulfilled. Additionally, for $\beta < 1/32$, it holds that $\sqrt{\frac{\ln(2)}{c|\mathcal{D}|\beta}} \leq \frac{1}{\beta}\sqrt{1/4 - 8\beta}$, so $\epsilon \leq \sqrt{\frac{\ln(2)}{c|\mathcal{D}|\beta}}$ is a more restrictive bound on $\epsilon$.

**Lemma 4** (**General simulation lemma**). *For any history-dependent policy $\pi : \mathcal{H}_t \to \Delta(\mathcal{A})$ and any two MDP models $m = (P, R), \hat{m} = (\hat{P}, \hat{R})$, it holds that:*

$$|J(\pi, m) - J(\pi, \hat{m})| \leq \frac{2r_{\max}}{(1-\gamma)^2} \mathbb{E}_{(s,a) \sim d_{\hat{m}}^{\pi}}[\mathrm{TV}(m(s,a), \hat{m}(s,a))].$$

*Proof.* Given an MDP $m$ and a policy $\pi$, define the value function starting from a history $h_t \in \mathcal{H}_t$:

$$V_m^{\pi}(h_t) := \mathbb{E}_{\pi,m}\left[\sum_{k=t}^{\infty} \gamma^{k-t} r_{k+1} | h_t\right].$$

It satisfies the history-based Bellman recursion, where $h_{t+1} = (h_t, a_t, r_{t+1}, s_{t+1})$,

$$V_m^{\pi}(h_t) = \mathbb{E}_{a_t \sim \pi(h_t)}\left[\mathbb{E}_{r_{t+1} \sim R(s_t,a_t), s_{t+1} \sim P(s_t,a_t)}[r_{t+1} + \gamma V_m^{\pi}(h_{t+1})]\right]$$
$$= \mathbb{E}_{a_t \sim \pi(h_t)}\left[\mathbb{E}_{(r_{t+1},s_{t+1}) \sim m(s_t,a_t)}[r_{t+1} + \gamma V_m^{\pi}(h_{t+1})]\right]. \tag{36}$$

Define the value difference between models $m$ and $\hat{m}$ given a history $h_t \in \mathcal{H}_t$:

$$\Delta^{\pi}(h_t) := V_m^{\pi}(h_t) - V_{\hat{m}}^{\pi}(h_t).$$

Thus the return gap is

$$|J(\pi, m) - J(\pi, \hat{m})| = |\mathbb{E}_{s_0 \sim \rho}[\Delta^{\pi}(s_0)]|.$$

Applying recursion (36) to both models $m$ and $\hat{m}$, we get:

$$\Delta^{\pi}(h_t) = \mathbb{E}_{a_t \sim \pi(h_t)}\left[\mathbb{E}_{m(s_t,a_t)}[r_{t+1} + \gamma V_m^{\pi}(h_{t+1})] - \mathbb{E}_{\hat{m}(s_t,a_t)}[r_{t+1} + \gamma V_{\hat{m}}^{\pi}(h_{t+1})]\right].$$

Decompose this into two parts:

$$\Delta_1^{\pi}(h_t, a_t) := \mathbb{E}_{m(s_t,a_t)}[r_{t+1} + \gamma V_m^{\pi}(h_{t+1})] - \mathbb{E}_{\hat{m}(s_t,a_t)}[r_{t+1} + \gamma V_m^{\pi}(h_{t+1})],$$
$$\Delta_2^{\pi}(h_t, a_t) := \mathbb{E}_{\hat{m}(s_t,a_t)}[r_{t+1} + \gamma V_m^{\pi}(h_{t+1})] - \mathbb{E}_{\hat{m}(s_t,a_t)}[r_{t+1} + \gamma V_{\hat{m}}^{\pi}(h_{t+1})],$$
$$\Delta^{\pi}(h_t) = \mathbb{E}_{a_t \sim \pi(h_t)}[\Delta_1^{\pi}(h_t, a_t) + \Delta_2^{\pi}(h_t, a_t)]. \tag{37}$$

Applying the fundamental property of TV distance (Levin & Peres, 2017), the first term:

$$\Delta_1^{\pi}(h_t, a_t) \leq 2\mathrm{TV}(m(s_t,a_t), \hat{m}(s_t,a_t)) \cdot \|r_{t+1} + \gamma V_m^{\pi}(h_{t+1})\|_{\infty}$$
$$\leq 2\mathrm{TV}(m(s_t,a_t), \hat{m}(s_t,a_t))\frac{r_{\max}}{1-\gamma}, \tag{38}$$

where the second inequality stems from the fact that:

$$\|r_{t+1} + \gamma V_m^{\pi}(h_{t+1})\|_{\infty} \leq r_{\max} + \gamma \frac{r_{\max}}{1-\gamma} = \frac{r_{\max}}{1-\gamma}.$$

The second term can be recognized as:

$$\Delta_2(h_t, a_t) = \gamma \mathbb{E}_{(r_{t+1}, s_{t+1}) \sim \hat{m}(s_t,a_t)}[\Delta^{\pi}(h_{t+1})]. \tag{39}$$

Combining Eq. 37 with Equations (38) and (39) yields:

$$\Delta^{\pi}(h_t) \leq \mathbb{E}_{a_t \sim \pi(h_t)}\left[\frac{2r_{\max}}{1-\gamma}\mathrm{TV}(m(s_t,a_t), \hat{m}(s_t,a_t))\right] + \gamma \mathbb{E}_{a_t \sim \pi(h_t)}\left[\mathbb{E}_{\hat{m}(s_t,a_t)}[\Delta^{\pi}(h_{t+1})]\right].$$

For convenience, denote the one-step error difference at time $t$ by:

$$\epsilon(s_t, a_t) := \frac{2r_{\max}}{1-\gamma}\mathrm{TV}(m(s_t,a_t), \hat{m}(s_t,a_t)),$$

so that

$$\Delta^{\pi}(h_t) \leq \mathbb{E}_{a_t \sim \pi(h_t)}[\epsilon(s_t, a_t)] + \gamma \mathbb{E}_{a_t \sim \pi(h_t)}\left[\mathbb{E}_{(r_{t+1}, s_{t+1}) \sim \hat{m}(s_t,a_t)}[\Delta^{\pi}(h_{t+1})]\right].$$

Iterating from $t = 0$ and taking $\mathbb{E}_{s_0 \sim \rho}$,

$$\mathbb{E}_{s_0 \sim \rho}[\Delta^{\pi}(s_0)]$$
$$\leq \mathbb{E}_{s_0 \sim \rho, a_0 \sim \pi(s_0)}[\epsilon(s_0, a_0)] + \gamma \mathbb{E}_{s_0 \sim \rho, a_0 \sim \pi(s_0), (r_1, s_1) \sim \hat{m}(s_0, a_0)}[\Delta^{\pi}(h_1)]$$

$$= \mathbb{E}_{h_0 \sim P^\pi_{\hat{m},0}, a_0 \sim \pi(h_0)}[\epsilon(s_0, a_0)] + \gamma \mathbb{E}_{h_1 \sim P^\pi_{\hat{m},1}}[\Delta^\pi(h_1)]$$

$$\leq \mathbb{E}_{h_0 \sim P^\pi_{\hat{m},0}, a_0 \sim \pi(h_0)}[\epsilon(s_0, a_0)] + \gamma \left( \mathbb{E}_{h_1 \sim P^\pi_{\hat{m},1}, a_1 \sim \pi(h_1)}[\epsilon(s_1, a_1)] + \gamma \mathbb{E}_{h_2 \sim P^\pi_{\hat{m},2}}[\Delta^\pi(h_2)] \right)$$

$$= \mathbb{E}_{h_0 \sim P^\pi_{\hat{m},0}, a_0 \sim \pi(h_0)}[\epsilon(s_0, a_0)] + \gamma \mathbb{E}_{h_1 \sim P^\pi_{\hat{m},1}, a_1 \sim \pi(h_1)}[\epsilon(s_1, a_1)] + \gamma^2 \mathbb{E}_{h_2 \sim P^\pi_{\hat{m},2}}[\Delta^\pi(h_2)]$$

$$\vdots$$

$$\leq \sum_{t=0}^\infty \gamma^t \mathbb{E}_{h_t \sim P^\pi_{\hat{m},t}, a_t \sim \pi(h_t)}[\epsilon(s_t, a_t)]$$

$$= \frac{1}{1-\gamma} \mathbb{E}_{(s,a) \sim d^\pi_{\hat{m}}}[\epsilon(s, a)],$$

where in the last line, we use the discounted occupancy $d^\pi_{\hat{m}}$ under $\hat{m}$:

$$d^\pi_{\hat{m}}(s, a) = (1 - \gamma) \sum_{t=0}^\infty \gamma^t \mathbb{P}(s_t = s, a_t = a \mid \pi, \hat{m}).$$

Finally, by symmetry (the same argument applied to $-\mathbb{E}_{s_0 \sim \rho}[\Delta^\pi(s_0)]$),

$$|\mathbb{E}_{s_0 \sim \rho}[\Delta^\pi(s_0)]| \leq \left| \frac{1}{1-\gamma} \mathbb{E}_{(s,a) \sim d^\pi_{\hat{m}}}[\epsilon(s, a)] \right|$$

$$= \frac{1}{1-\gamma} \mathbb{E}_{(s,a) \sim d^\pi_{\hat{m}}} \left[ \frac{2 r_{\max}}{1-\gamma} \mathrm{TV}(m(s, a), \hat{m}(s, a)) \right]$$

$$= \frac{2 r_{\max}}{(1-\gamma)^2} \mathbb{E}_{(s,a) \sim d^\pi_{\hat{m}}}[\mathrm{TV}(m(s, a), \hat{m}(s, a))].$$

$\square$

## D.4 RELATED WORK ON OFFLINE RL THEORY

Theoretical works on offline RL aim to establish performance guarantees without online exploration. The key difficulty is limited data coverage: when the dataset sufficiently covers all state-action pairs, standard PAC-style guarantees can be obtained without requiring conservatism (Munos & Szepesvári, 2008). Under partial coverage, however, conservative algorithms that penalize policies deviating from well-supported regions are essential to ensure robust learning. Many prior results establish minimax optimality with information-theoretic bounds under *worst-case* assumptions over possible MDPs (Jin et al., 2021; Rashidinejad et al., 2021; Li et al., 2024). The lower-bound part in Lemma 2 draws on the same information-theoretic principle as Li et al. (2024), but we provide a simpler counterexample tailored to our sequential bandit problem in Sec. 3.

Complementary to this line of work, our work adopts a Bayesian viewpoint that focuses on the *average case* over possible MDPs, an aspect underexplored in the theoretical literature. An exception is the Bayesian offline RL setting studied in Uehara & Sun (2022, Section 8) under a Markov policy, optimized via mirror descent with posterior sampling. Their analysis derives a Bayesian sub-optimality gap and interprets the resulting soft robustness as "implicit pessimism". However, because their analysis is restricted to Markov policies, whereas the Bayes-optimal policy is generally history-dependent, their regret bound is looser than ours in Lemma 1.

Recent work (Fellows et al., 2025) addresses the same Bayesian offline RL problem with history-dependent policies. Their main result (Theorem 1) provides a Bayesian sub-optimality bound expressed by an expected KL divergence, called the posterior information loss, and takes the worst case over all policies. Their derivation starts from a TV distance, which they upper bound by a KL divergence to leverage the product rule of logarithms. In that respect, our Lemma 1 gives a tighter upper bound as it is based on TV distance rather than KL, which requires extending the simulation lemma to history-dependent policies (see Lemma 4). Additionally, the regret bound provided in Fellows et al. (2025, Theorem 1) involves a supremum over all policies. Although we believe their bound can be established directly on the Bayes-optimal policy rather than the worst case, building on their current result would require a full data-coverage assumption. In contrast, our lower bound in Lemma 1 only requires partial data coverage. Finally, Fellows et al. (2025) focuses on bounding the

Bayes regret. At the same time, we additionally provide a lower bound on the robust sub-optimality gap in Lemma 2, showing when the Bayesian approach can provably outperform conservative ones.

# E   NEUBAY ALGORITHM DETAILS

**RL loss function.** In recurrent off-policy RL, optimization is typically performed on full trajectories (rather than i.i.d. transition tuples) for compute efficiency. In our setting, each training trajectory is a *concatenation* of a real prefix and an imagined rollout: starting from an initial history $h_t \in \mathcal{D}$, a world model and the policy generate future steps until truncation at $t' \leq T$. Formally, let

$$\tau = (s_{0:t} \oplus s_{t+1:t'}, a_{0:t-1} \oplus a_{t:t'-1}, r_{1:t} \oplus \hat{r}_{t+1:t'}, d_{1:t} \oplus \hat{d}_{t+1:t'}),$$

where $\oplus$ denotes *concatenation*. Our RL loss on $\tau$ balances contributions from real and imagined segments:

$$L(Q_\omega, \pi_\nu; \tau, \kappa) := \frac{\kappa}{t} \sum_{j=0}^{t-1} l(Q_\omega, \pi_\nu; h_j, a_j, r_{j+1}, d_{j+1}, s_{j+1}) + \frac{1-\kappa}{t'-t} \sum_{j=t}^{t'-1} l(Q_\omega, \pi_\nu; \hat{h}_j, \hat{a}_j, \hat{r}_{j+1}, \hat{d}_{j+1}, \hat{s}_{j+1}).$$

(40)

Here, $\hat{h}_j$ denotes the imagined history (with $\hat{h}_t = h_t$) and $\kappa \in (0,1)$ is the real data ratio. The per-step loss $l(Q_\omega, \pi_\nu; h_j, a_j, r_{j+1}, d_{j+1}, s_{j+1})$ is standard off-policy loss without conservatism, such as DQN (Mnih et al., 2013) for discrete control and SAC (Haarnoja et al., 2018a) for continuous control.

**Planning stopping criteria.** In our rollout subroutine (Algorithm 2), a rollout finishes when any of three conditions hold:

$$\text{done}_{t+1} := (U_{\boldsymbol{\theta}}(\hat{s}_t, \hat{a}_t) > \mathcal{U}(\zeta)) \vee (t+1 \geq T) \vee f_{\text{term}}(\hat{s}_t, \hat{a}_t, \hat{s}_{t+1}).$$

(41)

1. Uncertainty truncation: as described in Sec. 4.2, instead of enforcing a fixed horizon $H$, we truncate rollouts adaptively using an uncertainty threshold calibrated on the real dataset. This allows planning to extend as long as the model remains confident.
2. Timout truncation: to remain consistent with test-time evaluation, we impose a hard cap at the environment's maximum episode length $T$, regardless of rollout length.
3. Ground-truth termination: we retain the environment's rule-based terminal function to provide true terminal signals $\hat{d}_{t+1}$, following prior model-based RL methods (Yu et al., 2020). Including this prior knowledge makes our algorithm directly comparable to model-based baselines, which are our main focus.

Importantly, only the terminal signal disables bootstrapping in RL, while both uncertainty- and timeout-based truncations preserve bootstrapping[10], which aligns with the Bayesian objective.

# F   DATASET DETAILS

**The bandit dataset.** As introduced in Sec. 3, we construct a skewed two-armed bandit dataset. We collect 10 trajectories of length $T = 100$, yielding $|\mathcal{D}| = 1000$ action–reward pairs. We use one-hot encoding on actions as inputs for reward models and agents. The dataset is split into training and validation sets with a 4:1 ratio. Since $\mathcal{D}$ only covers arm 0 with $p_0^* = 0.5$, the true parameter of arm 1, $p_1^*$, is completely unseen.

At test time, we vary $p_1^* \in \{0.01, 0.3, 0.55, 0.7, 0.99\}$, where each choice defines a distinct bandit problem. Each problem is evaluated over 20 independent episodes, and all problems are assessed in parallel, yielding $5 \times 20 = 100$ evaluation runs. The normalized return during test time is computed by: $\frac{1}{T} \sum_{t=0}^{T-1} r_{t+1}$, where $r_{t+1} \sim \mathcal{B}(p_{a_t}^*)$.

**D4RL locomotion benchmark.** We evaluate on the standard D4RL locomotion benchmark (Fu et al., 2020), comprising 12 datasets formed by the Cartesian product of tasks (halfcheetah, hopper, walker2d) and dataset types (random-v2, medium-v2, medium-replay-v2, medium-expert-v2). This benchmark is the most widely used in offline RL research. The underlying environments are OpenAI

---

[10] https://gymnasium.farama.org/tutorials/gymnasium_basics/handling_time_limits/.

Gym tasks: HalfCheetah-v2 ($\mathcal{S} \subset \mathbb{R}^{17}, \mathcal{A} \subset \mathbb{R}^6$), Hopper-v2 ($\mathcal{S} \subset \mathbb{R}^{11}, \mathcal{A} \subset \mathbb{R}^3$), and Walker2d-v2 ($\mathcal{S} \subset \mathbb{R}^{17}, \mathcal{A} \subset \mathbb{R}^6$). The maximum episode step $T$ is 1000. Hopper-v2 and Walker2d-v2 have termination functions, while HalfCheetah-v2 does not.

Dataset sizes vary: 100k-200k transitions for medium-replay, 1M for random and medium, and 2M for medium-expert. These datasets also differ qualitatively. The random dataset is collected with a uniformly random policy; medium-replay corresponds to the replay buffer of an agent trained to a medium-level policy; medium itself is generated directly from a medium-level policy; and medium-expert is a mixture of trajectories from both a medium policy and an expert policy. Based on these properties, we categorize random as *low-quality*, medium-replay and medium-expert as *moderate coverage*, and medium as *narrow coverage*.

Performance is reported in terms of normalized scores, following the D4RL and NeoRL conventions:

$$\text{normalized score} = \frac{\text{score} - \text{random score}}{\text{expert score} - \text{random score}} \times 100.$$

It is worth noting that the expert policies in these locomotion tasks are not strictly optimal. As a result, it is possible for algorithms to achieve normalized scores greater than 100 (e.g., around 120). Therefore, we consistently categorize all medium-$\star$ datasets as medium-quality.

**NeoRL locomotion benchmark.** The locomotion benchmark in the NeoRL (Qin et al., 2022) has been widely used in recent model-based offline RL methods. The setup closely mirrors the D4RL locomotion benchmark, with nearly identical environments: HalfCheetah-v3 ($\mathcal{S} \subset \mathbb{R}^{18}, \mathcal{A} \subset \mathbb{R}^6$), Hopper-v3 ($\mathcal{S} \subset \mathbb{R}^{12}, \mathcal{A} \subset \mathbb{R}^3$), and Walker2d-v3 ($\mathcal{S} \subset \mathbb{R}^{18}, \mathcal{A} \subset \mathbb{R}^6$). Compared to D4RL, NeoRL increases the state dimensionality by one in each environment.

For each environment, NeoRL provides three datasets (Low, Medium, High), collected using policies of the corresponding performance levels. This leads to 9 datasets in total. Compared to their D4RL counterparts, these policies are more deterministic, leading to smaller data coverage, which we categorize as *narrow coverage* in this paper. Dataset sizes range from roughly 200k to 1M transitions. To avoid ambiguity, throughout the paper we denote NeoRL datasets with capitalized environment names and the v3 suffix (e.g., *HalfCheetah-v3-Medium*), while D4RL datasets are written in lowercase with the v2 suffix (e.g., *halfcheetah-medium-v2*).

**D4RL Adroit benchmark.** The Adroit benchmark is widely regarded as substantially more challenging than locomotion tasks. It involves controlling a 28-DoF robotic arm to perform high-dimensional manipulation tasks: Pen ($\mathcal{S} \subset \mathbb{R}^{45}, \mathcal{A} \subset \mathbb{R}^{24}, T = 100$), Door ($\mathcal{S} \subset \mathbb{R}^{39}, \mathcal{A} \subset \mathbb{R}^{28}, T = 200$), Hammer ($\mathcal{S} \subset \mathbb{R}^{46}, \mathcal{A} \subset \mathbb{R}^{26}, T = 200$). Among these, Pen includes a termination function, while Door and Hammer do not. Rewards combine a dense component based on distances with a sparse component that grants a large bonus once a distance threshold is satisfied.

In addition to the high dimensionality and sparsity of rewards, most Adroit datasets are either small or of limited quality. The human demonstration datasets (human-v1) contain only 5k–10k transitions, whereas the cloned datasets (cloned-v1), constructed by mixing behavior cloning with human demonstrations, contain 500k–1M transitions. This leads to 6 datasets in total. Accordingly, we categorize human datasets as *narrow coverage* and cloned datasets as *moderate coverage*.

Dataset performance levels vary significantly, as shown in the $\pi_\mathcal{D}$ column of Tab. 11. Pen-human-v1 and pen-cloned-v1 achieve normalized scores in the range of 70–90, whereas door-human-v1, door-cloned-v1, hammer-human-v1, and hammer-cloned-v1 yield scores below 10. Accordingly, we categorize pen-$\star$ datasets as *medium-quality*, and the remaining ones as *low-quality*. This disparity explains why most algorithms fail to achieve meaningful results on the door and hammer datasets. Finally, we exclude the relocate tasks from our experiments, as both prior baselines and our method consistently obtain near-zero scores in this setting.

**D4RL AntMaze benchmark.** AntMaze is a particularly challenging benchmark in D4RL due to its sparse reward structure. It involves controlling a MuJoCo Ant ($\mathcal{S} \subset \mathbb{R}^{29}, \mathcal{A} \subset \mathbb{R}^8$) to navigate in different maze layouts (umaze, medium, large). The agent receives reward 1 only when reaching a goal position in the X–Y plane within a threshold, after which the episode terminates, so the maximum return is 1. Goals are randomly sampled from a small region but remain unobserved to the agent, making evaluation noisy for RL algorithms. The maximum episode length is $T = 700$ for umaze and $T = 1000$ for medium and large mazes.

For each layout, two datasets (play and diverse) are provided, each containing 1M transitions. The *diverse* variants introduce a wider set of start positions compared to *play*. However, all AntMaze datasets are generated by a hierarchical controller: a high-level breadth-first search (BFS) planner selects waypoints, which are then executed by a low-level learned policy. As a result, trajectories are highly structured and largely restricted to narrow corridors that connect goals without collisions. This planner-driven generation induces narrow coverage, a property noted in prior work (Zhang et al., 2024a).

Performance in AntMaze is generally low relative to other D4RL domains. In particular, umaze-diverse has an average score of only 1.0, which we categorize as *low-quality*. The remaining AntMaze datasets achieve scores above 10.0, which we categorize as *medium-quality* (though they could also be viewed as low-quality when considered outside the AntMaze domain).

Following prior model-based RL work (LEQ (Park & Lee, 2025)[11], ADMPO (Lin et al., 2025)[12]), we adopt the same terminal functions in AntMaze. As in LEQ, we also shift the reward by $-1.0$. Since termination reveals information about the reward function (termination implies success), we do not compare against other model-based or model-free algorithms. Instead, our baselines in AntMaze are the methods reported in LEQ and ADMPO.

## G   IMPLEMENTATION DETAILS

Our implementation is built on JAX (Bradbury et al., 2018) and Equinox (Kidger & Garcia, 2021)[13].

### G.1   WORLD MODEL ENSEMBLE AND PLANNING

**World model ensemble.** Throughout our experiments, we train an ensemble of 128 MLPs on each dataset. Following MOPO (Yu et al., 2020), for continuous control tasks, we rank the models by MSE on a validation set consisting of 1000 held-out transitions from $\mathcal{D}$. For the bandit task, we rank by negative log-likelihood (NLL) that captures uncertainty, since the true reward follows Bernoulli distribution. The top $N$ models are then selected to form the world model ensemble $\mathbf{m}_{\boldsymbol{\theta}}$, which remains fixed during subsequent policy training. In our main experiments (Sec. 5.1), we use $N = 100$. For the sensitivity analysis (Sec. 5.2), we vary this number to 5 and 20.

Following MOBILE (Sun et al., 2023; Sun, 2023)[14], each MLP $m_{\theta}(s, a)$ has 4 hidden layers with a width of 200 for the locomotion benchmarks, and 5 layers with a width of 400 for Adroit benchmark. As described in Sec. 4.1, we apply LayerNorm after each hidden layer; this design choice is further ablated in Sec. 5.2. The basic block consists of (Linear $\rightarrow$ LayerNorm $\rightarrow$ leaky ReLU) when LayerNorm is enabled, and (Linear $\rightarrow$ leaky ReLU) when it is disabled. For the bandit task, we use a small reward-model ensemble that has 2 hidden layers with a width of 16. The weights of each MLP are initialized independently using the default `equinox.nn.Linear` scheme, i.e., $\mathrm{Unif}[-\frac{1}{\sqrt{\dim_{\mathrm{in}}}}, \frac{1}{\sqrt{\dim_{\mathrm{in}}}}]$, where $\dim_{\mathrm{in}}$ denotes the input feature dimension.

For continuous control tasks, during ensemble training, we sample transition tuples $(s, a, r, s')$ from $\mathcal{D}$ using a batch size of 256 to estimate the MLE loss. The inputs $(s, a)$ and outputs $(r, s')$ are standardized by subtracting the mean and dividing by the standard deviation computed over the dataset; during inference, predictions are inverse-transformed to restore the original scale. We use AdamW optimizer (Loshchilov & Hutter, 2019) with a weight decay coefficient of $5 \times 10^{-5}$, and a learning rate of $1 \times 10^{-3}$ for locomotion tasks and $3 \times 10^{-4}$ for Adroit tasks. Training is terminated early if the validation MSE fails to improve by more than 0.01 relative within five consecutive epochs, following the early stopping procedure in MOBILE. In the bandit task, the model learning rate is $1 \times 10^{-3}$, batch size is 128, and improvement threshold is 0.001 absolute.

**Planning.** At the start of planning, following Sec. 4.2, we sample a batch of initial histories $h_t \sim \mathcal{D}$, drawn uniformly across time steps. The batch size is fixed at 100, regardless of the ensemble size $N$. Since the values of $N$ used in our experiments $\{5, 20, 100\}$ are divisors of 100, we distribute the

---

[11]https://github.com/kwanyoungpark/LEQ.
[12]https://github.com/HxLyn3/ADMPO.
[13]https://github.com/patrick-kidger/equinox.
[14]https://github.com/yihaosun1124/mobile.

histories evenly across ensemble members. The recurrent policy $\pi_\nu$ initializes its hidden state $z_t$ with $h_t$ and then interacts with each world model to generate corresponding rollout until reaching the stopping criterion described in Sec. E. The uncertainty threshold $\mathcal{U}(\zeta)$ for truncation is fixed at $\zeta = 1.0$ in our main experiments and varied to $\{0.9, 0.99, 0.999\}$ in the sensitivity analysis. The entire planning process is parallelized on a GPU and thus introduces only negligible time costs.

For the bandit task, since there are no transition dynamics and hence no compounding error, we directly optimize the Bayesian objective: planning starts at $t = 0$ and truncates at $t = T$.

## G.2 RECURRENT OFF-POLICY RL

**Off-policy loss implementation.** For continuous control tasks, we follow a recent recurrent off-policy RL algorithm RESeL (Luo et al., 2024a) to use REDQ (Chen et al., 2021b) as the per-step loss for $l(\cdot)$, used in the overall RL loss defined by Eq. 40. REDQ builds on SAC (Haarnoja et al., 2018a;b)[15], maintaining an ensemble of 10 critic MLPs and sampling 2 of them to form bootstrapped targets in the critic loss, while the actor maximizes the average Q-value over all ensemble members. Although REDQ substantially reduces value overestimation, it does not deliberately *underestimate* values. Moreover, REDQ and RESeL were designed for *online* RL; in our framework, they are thus *not* considered conservative algorithms.

For the bandit task, we adopt dueling DQN (Wang et al., 2016) following memoroid (Morad et al., 2024) as the discrete control algorithm. Exploration is $\epsilon$-greedy, annealed from 1.0 to 0.1 over the first 10% of gradient steps.

**Agent architecture implementation.** As introduced in Sec. 4.3, our agent consists of a recurrent actor $\pi_\nu : \mathcal{H}_t \to \Delta(\mathcal{A})$ and a recurrent critic $Q_\omega : \mathcal{H}_t \times \mathcal{A} \to \mathbb{R}^{10}$. The critic outputs an ensemble of 10 Q-values, following the REDQ design adopted in RESeL. Both actor and critic maintain their own RNN encoders, $\nu_\phi : \mathcal{H}_t \to \mathcal{Z}$ and $\omega_\phi : \mathcal{H}_t \to \mathcal{Z}$, which share the same architecture but are optimized independently. As mentioned earlier, we adopt the *memoroid* framework (Morad et al., 2024)[16] to use the linear recurrent unit (LRU) (Orvieto et al., 2023) as the backbone encoder for both $\nu_\phi$ and $\omega_\phi$. The actor and critic MLP heads have 2 or 3 hidden layers with a hidden size of 256. For the bandit task, there is only a critic MLP head with 2 hidden layers.

The LRU begins with a nonlinear preprocessing layer that projects the raw input history $h_t$ into a 256-dimensional feature space (preserving the time dimension). This is followed by a stack of two LRU layers, each performing a linear recurrence update parameterized by a complex-valued diagonal matrix. Each layer maintains a hidden state of size 128, resulting in a recurrent representation $z_t \in \mathbb{C}^{2 \times 128}$. From this recurrent state, the model produces a real-valued output vector $\tilde{z}_t \in \mathbb{R}^{128}$ via a nonlinear projection. During training and inference, $\tilde{z}_t$ is fed into the actor or critic MLP heads, while the complex hidden state $z_t$ is preserved for recurrent updates during policy inference.

We fix the recurrent architecture, including hidden sizes, across all experiments. However, NEUBAY is compatible with any RNN encoder, and we leave a study of architectural variations to future work. For the ablation study with Markov agent, we remove the LRU encoder and only train the actor-critic MLP.

**Tape-based batching.** Classic recurrent RL relies on *segment-based batching* (Ni et al., 2022), where sequences are padded to a fixed length, forming 3D tensors of shape $(\text{batch size}, \text{sequence length}, \text{dim})$ with NaN masks for shorter sequences. This wastes memory and lowers sample efficiency. Memoroid (Morad et al., 2024) introduces *tape-based batching*, which exploits the monoid algebra of linear RNNs such as LRUs. Instead of padding, variable-length sequences are concatenated into a single 2D "tape", making the effective batch size equal to the sum of raw sequence lengths. Inline resets of hidden states prevent leakage across sequences within a tape (Lu et al., 2023), and computation over the tape is parallelized using associative scan in JAX (Bradbury et al., 2018). To enable JIT compilation, a fixed tape length is enforced by dropping any trailing timesteps that exceed this length. We refer readers to Morad et al. (2024) for full details.

---

[15]Our REDQ implementation is adapted from the SAC-N Equinox codebase: https://github.com/Howuhh/sac-n-jax.

[16]https://github.com/proroklab/memoroids.

In our implementation, we adopt tape-based batching and we set the tape length to be larger than the maximal episode length $T$ in each task. To reduce computation time, we choose a relatively small tape length, similar to prior work in online POMDPs (Morad et al., 2024; Luo et al., 2024a).

**Training hyperparameters.** Tab. 3 summarizes the modules and hyperparameters fixed in our experiments. Consistent with IQL (Kostrikov et al., 2022) and MOBILE (Sun et al., 2023), we apply cosine learning rate decay only to the actor network (both the RNN encoder and MLP head), but not to the critic, in order to promote stability during the later stages of training.

The REDQ (SAC) entropy coefficient $\alpha$ is auto-tuned with a target entropy of $-\dim(\mathcal{A})$ (Haarnoja et al., 2018b) for all datasets except for D4RL and NeoRL Hopper datasets and D4RL AntMaze domain. In the D4RL hopper domain, we find our algorithm is sensitive to $\alpha$, consistent with prior recurrent RL work (Luo et al., 2024a). To address this, we follow MOBILE and fix $\alpha = 0.2$ across all hopper datasets (four in D4RL and three in NeoRL), without further tuning.

In AntMaze, sparse-reward navigation makes SAC sensitive to the choice of $\alpha$, e.g., ADMPO (Lin et al., 2025) uses a fixed $\alpha = 0.05$ and disables the entropy term backup in critic loss. We follow their insights but instead tune the target entropy $\in \{-\dim(\mathcal{A}), -5\dim(\mathcal{A}), -10\dim(\mathcal{A})\}$. We find $-10\dim(\mathcal{A})$ works best overall and report it in our main results.

For the bandit task, we sweep the RNN encoder learning rate $\eta_\phi \in \{1 \times 10^{-6}, 3 \times 10^{-6}, 1 \times 10^{-5}, 3 \times 10^{-5}, 1 \times 10^{-4}\}$ in the critic network. Consistent with observations from ReSEL (Luo et al., 2024a) and our continuous control results, a small learning rate is crucial for stable training. We use $\eta_\phi = 3 \times 10^{-6}$ in our bandit experiments.

Table 3: Fixed hyperparameters used in our recurrent agents. The last block (actor and policy entropy) is only used in continuous control.

| Module or Hyperparameter | Value |
| --- | --- |
| Actor and critic RNN encoders | 2-layer LRUs (Orvieto et al., 2023) |
| RNN hidden state size | 256 |
| Actor and critic heads | 2-layer MLPs (3 layers in Adroit) |
| Basic block of MLP head | (Linear $\rightarrow$ LayerNorm $\rightarrow$ leaky ReLU) |
| MLP head hidden size | 256 |
| Batch size (i.e., tape length) | 2048 (1024 in Adroit, 1000 in bandit) |
| Update-to-data (UTD) ratio | 0.05 (0.02 in bandit) |
| Gradient steps | 2M (3M in AntMaze, 20k in bandit) |
| Replay buffer size | Full size, i.e., (60M in AntMaze, 1M in bandit, 40M otherwise) |
| Discount factor $\gamma$ | 0.99 |
| Critic head's learning rate | $1 \times 10^{-4}$ |
| Gradient norm clipping | 1000 (10000 in Adroit, 1 in bandit and AntMaze) |
| Actor head's learning rate | $1 \times 10^{-4}$ |
| Actor's learning rate decay | Cosine decay to 0.0 |
| Entropy coef. $\alpha$'s learning rate | $1 \times 10^{-4}$ |
| Entropy coef. $\alpha$ | Auto-tuned with target $-\dim(\mathcal{A})$ (AntMaze uses $-10\dim(\mathcal{A})$ while Hopper uses fixed $\alpha = 0.2$) |

Table 4: Best hyperparameters per dataset in the D4RL locomotion benchmark. We sweep $\eta_\phi \in \{3 \times 10^{-7}, 1 \times 10^{-6}, 3 \times 10^{-6}, 1 \times 10^{-5}, 3 \times 10^{-5}\}$ and $\kappa \in \{0.05, 0.5, 0.8\}$.

| Dataset | RNN encoder lr $\eta_\phi$ | Real data ratio $\kappa$ |
|---|---|---|
| halfcheetah-random-v2 | $3 \times 10^{-5}$ | 0.8 |
| hopper-random-v2 | $1 \times 10^{-5}$ | 0.5 |
| walker2d-random-v2 | $3 \times 10^{-5}$ | 0.5 |
| halfcheetah-medium-replay-v2 | $1 \times 10^{-5}$ | 0.05 |
| hopper-medium-replay-v2 | $3 \times 10^{-7}$ | 0.5 |
| walker2d-medium-replay-v2 | $1 \times 10^{-6}$ | 0.5 |
| halfcheetah-medium-v2 | $3 \times 10^{-5}$ | 0.8 |
| hopper-medium-v2 | $1 \times 10^{-6}$ | 0.8 |
| walker2d-medium-v2 | $3 \times 10^{-6}$ | 0.5 |
| halfcheetah-medium-expert-v2 | $3 \times 10^{-5}$ | 0.8 |
| hopper-medium-expert-v2 | $3 \times 10^{-6}$ | 0.5 |
| walker2d-medium-expert-v2 | $1 \times 10^{-6}$ | 0.8 |

Table 5: Best hyperparameters per dataset in the NeoRL locomotion benchmark. We sweep $\eta_\phi \in \{3 \times 10^{-7}, 1 \times 10^{-6}, 3 \times 10^{-6}, 1 \times 10^{-5}, 3 \times 10^{-5}\}$ and $\kappa \in \{0.5, 0.8\}$.

| Dataset | RNN encoder lr $\eta_\phi$ | Real data ratio $\kappa$ |
|---|---|---|
| HalfCheetah-v3-Low | $1 \times 10^{-5}$ | 0.8 |
| Hopper-v3-Low | $3 \times 10^{-6}$ | 0.8 |
| Walker2d-v3-Low | $3 \times 10^{-7}$ | 0.5 |
| HalfCheetah-v3-Medium | $3 \times 10^{-5}$ | 0.5 |
| Hopper-v3-Medium | $3 \times 10^{-6}$ | 0.5 |
| Walker2d-v3-Medium | $3 \times 10^{-7}$ | 0.8 |
| HalfCheetah-v3-High | $3 \times 10^{-5}$ | 0.5 |
| Hopper-v3-High | $3 \times 10^{-6}$ | 0.5 |
| Walker2d-v3-High | $3 \times 10^{-7}$ | 0.8 |

Table 6: Best hyperparameters per dataset in the D4RL Adroit benchmark. We sweep $\eta_\phi \in \{1 \times 10^{-6}, 3 \times 10^{-6}, 1 \times 10^{-5}, 3 \times 10^{-5}, 1 \times 10^{-4}\}$ and $\kappa \in \{0.5, 0.8\}$. For the remaining Adroit datasets, all hyperparameter settings yield near-zero performance.

| Dataset | RNN encoder lr $\eta_\phi$ | Real data ratio $\kappa$ |
|---|---|---|
| pen-human-v1 | $3 \times 10^{-5}$ | 0.5 |
| pen-cloned-v1 | $1 \times 10^{-5}$ | 0.8 |
| hammer-cloned-v1 | $1 \times 10^{-5}$ | 0.5 |

Table 7: Best hyperparameters per dataset in the D4RL AntMaze benchmark. We sweep $\eta_\phi \in \{1 \times 10^{-6}, 3 \times 10^{-6}, 1 \times 10^{-5}, 3 \times 10^{-5}\}$ and $\kappa \in \{0.5, 0.8, 0.95\}$. For the large maze datasets, all hyperparameter settings yield near-zero performance.

| Dataset | RNN encoder lr $\eta_\phi$ | Real data ratio $\kappa$ |
|---|---|---|
| antmaze-umaze-v2 | $3 \times 10^{-6}$ | 0.95 |
| antmaze-umaze-diverse-v2 | $3 \times 10^{-6}$ | 0.95 |
| antmaze-medium-play-v2 | $1 \times 10^{-5}$ | 0.95 |
| antmaze-medium-diverse-v2 | $1 \times 10^{-6}$ | 0.95 |

### G.3 COMPUTATION DETAILS

Table 8: **Time cost (seconds) of the Rollout function** (Algorithm 2) for different ensemble sizes $N$ and numbers of parallel rollouts $K$. In our main experiments, $N = K = 100$.

|           | $K = 5$ | $K = 20$ | $K = 100$ |
|-----------|---------|----------|-----------|
| $N = 5$   | 2.7s    | 3.0s     | 4.7s      |
| $N = 20$  | N/A     | 3.6s     | 5.0s      |
| $N = 100$ | N/A     | N/A      | 5.3s      |

**Rollout costs.** Tab. 8 reports the time cost of our Rollout implementation for halfcheetah-medium-expert-v2. We assign each ensemble member an equal number of rollouts, so we only benchmark configurations where $K$ is divisible by $N$. Thanks to full vectorization over ensemble members and rollouts using `jax.vmap`, rollout inference is very efficient: increasing $N$ or $K$ has only a minor effect on runtime. Consequently, rollout cost is negligible, and agent training time is dominated by gradient updates rather than rollout generation.

Table 9: **World model training time (per seed)** for different total ensemble size $N_{\text{total}}$. In practice, we use $N_{\text{total}} = 128$.

| $N_{\text{total}} = 8$ | $N_{\text{total}} = 32$ | $N_{\text{total}} = 128$ |
|------------------------|-------------------------|--------------------------|
| 1.2 hrs (1x)           | 1.7 hrs (1.42x)         | 6.0 hrs (5x)             |

**Model training time.** For completeness, Tab. 9 reports the training time for several choices of $N_{\text{total}}$ on halfcheetah-medium-expert-v2 to provide additional compute context. The training time is *roughly sublinear* in $N_{\text{total}}$. In practice, we train one ensemble of size $N_{\text{total}} = 128$ and select the top $N = 5, 20, 100$.

**Parameter size.** In the locomotion benchmarks, the world model ensemble with $N = 100$ contains fewer than 15M parameters in total. The recurrent encoder has fewer than 600k parameters, the actor MLP head fewer than 200k, and the critic MLP head (with an ensemble size of 10) fewer than 1.5M.

**Comparison with Unifloral's JAX implementation (Jackson et al., 2025).** Unifloral reports a fast JAX implementation of MOPO (28 minutes for 2M gradient steps). This speedup primarily stems from their use of `jax.lax.scan` to JIT-compile the entire MOPO gradient update loop, a technique known to provide roughly $5\times$ speedups in JAX.[17] Our NEUBAY implementation does not adopt this optimization because it is not directly compatible with our recurrent RL update, although it can be incorporated with additional engineering effort. Applying the same `scan`-based optimization to the Markov version of NEUBAY would reduce its training time from 2.6 hours to approximately 0.5 hours, comparable to Unifloral's numbers. Importantly, we report MOPO's PyTorch runtime *solely to contextualize compute context*, not as a comparison of JAX versus PyTorch efficiency, and the current training cost (4.4 hours) of NEUBAY is already within a practical and expected range for model-based RL.

---

[17]See `https://github.com/Howuhh/sac-n-jax`.

## H FURTHER RESULTS AND DISCUSSION

### H.1 FULL BENCHMARKING RESULTS

Table 10: Comparison of offline RL methods on the **NeoRL locomotion** benchmark. We report mean normalized scores for all baselines, with ±std for competitive baselines. The **best mean score** is bolded, and marked methods are statistically similar under a $t$-test. Our results use 6 seeds, each evaluated at the final step with 20 episodes.

| Dataset | Model-free | | Conservative model-based | | | | | | | Bayesian-inspired | | Ours |
|---|---|---|---|---|---|---|---|---|---|---|---|---|
| | EDAC | CQL | MOPO | COMBO | MOBILE | LEQ | ADMPO | ScorePen | VIPO | MAPLE | MoDAP | NEUBAY |
| hc-Low | 31.3 | 38.2 | 40.1 | 32.9 | $54.7_{\pm3.0}$ | $33.4_{\pm1.6}$ | $52.8_{\pm1.2}$ | $49.6_{\pm1.2}$ | $\mathbf{58.5}_{\pm0.1}$ | 33.4 | $53.9_{\pm1.1}$ | $53.1_{\pm1.1}$ |
| hp-Low | 18.3 | 16.0 | 6.2 | 17.9 | $17.4_{\pm3.9}$ | $24.2_{\pm2.3}$ | $22.3_{\pm0.1}$ | $21.1_{\pm2.3}$ | $\mathbf{30.7}_{\pm0.3}$ | 22.7 | $26.1_{\pm4.7}$ | $30.3_{\pm2.9}$ |
| wk-Low | 40.2 | 44.7 | 11.6 | 31.7 | $37.6_{\pm2.0}$ | $65.1_{\pm2.3}$ | $55.9_{\pm3.8}$ | $51.4_{\pm1.4}$ | $\mathbf{67.6}_{\pm0.7}$ | 33.9 | $51.3_{\pm7.8}$ | $43.9_{\pm5.9}$ |
| hc-Med | 54.9 | 54.6 | 62.3 | 50.8 | $77.8_{\pm1.4}$ | $59.2_{\pm3.9}$ | $69.3_{\pm1.7}$ | $77.4_{\pm1.0}$ | $80.9_{\pm0.2}$ | 69.5 | $81.0_{\pm2.3}$ | $\mathbf{81.1}_{\pm0.8}$ |
| hp-Med | 44.9 | 64.5 | 1.0 | 56.3 | $51.1_{\pm13.3}$ | $\mathbf{104.3}_{\pm5.2}$ | $51.5_{\pm5.0}$ | $90.9_{\pm1.3}$ | $66.3_{\pm0.2}$ | 27.7 | $44.2_{\pm15.3}$ | $95.7_{\pm11.5}$ |
| wk-Med | 57.6 | 57.3 | 39.9 | 53.8 | $62.2_{\pm1.6}$ | $45.2_{\pm19.4}$ | $70.1_{\pm2.4}$ | $65.8_{\pm1.6}$ | $\mathbf{76.8}_{\pm0.1}$ | 40.7 | $70.8_{\pm3.1}$ | $50.5_{\pm8.8}$ |
| hc-High | 81.4 | 77.4 | 65.9 | 62.2 | $83.0_{\pm4.6}$ | $71.8_{\pm8.0}$ | $84.0_{\pm0.8}$ | $81.4_{\pm1.0}$ | $\mathbf{89.4}_{\pm0.6}$ | — | $84.1_{\pm8.3}$ | $68.3_{\pm23.6}$ |
| hp-High | 52.5 | 76.6 | 11.5 | 63.2 | $87.8_{\pm26.0}$ | $95.5_{\pm13.9}$ | $87.6_{\pm4.9}$ | $86.3_{\pm1.3}$ | $\mathbf{107.7}_{\pm0.5}$ | — | $52.4_{\pm3.2}$ | $96.8_{\pm7.7}$ |
| wk-High | 75.5 | 75.3 | 18.0 | 71.8 | $74.9_{\pm3.4}$ | $73.7_{\pm1.1}$ | $\mathbf{82.2}_{\pm1.9}$ | $78.0_{\pm1.8}$ | $81.7_{\pm1.0}$ | — | $73.6_{\pm2.8}$ | $62.7_{\pm14.5}$ |
| AVG | 50.7 | 56.1 | 28.5 | 49.0 | 60.7 | 63.6 | 64.0 | 66.9 | $\mathbf{73.3}$ | — | 59.7 | 64.7 |

Table 11: Comparison of offline RL methods on the **D4RL Adroit** benchmark. We report mean normalized scores for all baselines, with ±std for competitive baselines. The **best mean score** is bolded, and marked methods are statistically similar under a $t$-test. Our results use 6 seeds, each evaluated at the final step with 20 episodes. In addition, we include the average dataset performance, denoted as $\pi_{\mathcal{D}}$, to provide a reference for **data quality**.

| Dataset | $\pi_{\mathcal{D}}$ | Model-free | | | | Conservative model-based | | | | | Ours |
|---|---|---|---|---|---|---|---|---|---|---|---|
| | | BC | EDAC | IQL | ReBRAC | MOPO | MOBILE | ARMOR | MoMo | VIPO | NEUBAY |
| pen-human | 88.7 | $71.0_{\pm6.3}$ | $52.1_{\pm8.6}$ | $78.5_{\pm8.2}$ | $\mathbf{103.2}_{\pm8.5}$ | 10.7 | $30.1_{\pm14.6}$ | $72.8_{\pm13.9}$ | 74.9 | $52.6_{\pm7.7}$ | $20.8_{\pm13.2}$ |
| pen-cloned | 68.7 | $51.9_{\pm15.2}$ | $68.2_{\pm7.3}$ | $83.4_{\pm8.2}$ | $\mathbf{102.8}_{\pm7.8}$ | 54.6 | $69.0_{\pm9.3}$ | $51.4_{\pm15.5}$ | 74.1 | $71.1_{\pm9.5}$ | $91.3_{\pm21.2}$ |
| door-human | 7.7 | $2.3_{\pm4.0}$ | $10.7_{\pm6.8}$ | $3.3_{\pm1.8}$ | $-0.1_{\pm0.0}$ | -0.2 | $-0.2_{\pm0.1}$ | $6.3_{\pm6.0}$ | $\mathbf{11.3}$ | $2.0_{\pm0.3}$ | $0.0_{\pm0.0}$ |
| door-cloned | 3.2 | $-0.1_{\pm0.0}$ | $9.6_{\pm8.3}$ | $3.1_{\pm1.8}$ | $0.1_{\pm0.1}$ | 15.3 | $\mathbf{24.0}_{\pm22.8}$ | $-0.1_{\pm0.0}$ | 5.8 | — | $0.0_{\pm0.0}$ |
| hammer-human | 1.5 | $\mathbf{3.0}_{\pm3.4}$ | $0.8_{\pm0.4}$ | $1.8_{\pm0.8}$ | $0.2_{\pm0.2}$ | 0.3 | $0.4_{\pm0.2}$ | $1.9_{\pm1.6}$ | 1.7 | $1.1_{\pm0.9}$ | $0.0_{\pm0.0}$ |
| hammer-cloned | 0.7 | $0.6_{\pm0.2}$ | $0.3_{\pm0.0}$ | $1.5_{\pm0.7}$ | $5.0_{\pm3.8}$ | 0.5 | $1.5_{\pm0.4}$ | $0.7_{\pm0.6}$ | 0.7 | $2.1_{\pm0.2}$ | $\mathbf{14.4}_{\pm9.7}$ |
| AVG | N/A | 21.5 | 23.6 | 28.6 | $\mathbf{35.2}$ | 13.5 | 20.8 | 22.2 | 28.1 | — | 21.1 |

Table 12: Comparison of offline model-based RL methods on the **D4RL AntMaze** benchmark. We report mean normalized scores for all baselines, with ±std for competitive baselines. The **best mean score** is bolded, and marked methods are statistically similar under a $t$-test. Our results use 6 seeds, each evaluated at the final step with 100 episodes. In addition, we include the average dataset performance, denoted as $\pi_{\mathcal{D}}$, to provide a reference for **data quality**.

| Dataset | $\pi_{\mathcal{D}}$ | CBOP | MOBILE | MOBILE† | ADMPO | LEQ | NEUBAY |
|---|---|---|---|---|---|---|---|
| umaze | 28.8 | $0.0_{\pm0.0}$ | $0.0_{\pm0.0}$ | 77.0 | $88.4_{\pm1.2}$ | $\mathbf{94.4}_{\pm6.3}$ | $66.1_{\pm20.1}$ |
| umaze-diverse | 1.0 | $0.0_{\pm0.0}$ | $0.0_{\pm0.0}$ | 20.4 | $\mathbf{81.7}_{\pm8.6}$ | $71.0_{\pm12.3}$ | $74.4_{\pm12.1}$ |
| medium-play | 19.6 | $0.0_{\pm0.0}$ | $0.0_{\pm0.0}$ | 64.6 | $23.9_{\pm6.3}$ | $\mathbf{58.8}_{\pm33.0}$ | $12.8_{\pm19.2}$ |
| medium-diverse | 11.3 | $0.0_{\pm0.0}$ | $0.0_{\pm0.0}$ | 1.6 | $24.1_{\pm5.7}$ | $\mathbf{46.2}_{\pm23.2}$ | $19.4_{\pm12.4}$ |
| large-play | 10.5 | $0.0_{\pm0.0}$ | $0.0_{\pm0.0}$ | 2.6 | $8.3_{\pm4.1}$ | $\mathbf{58.6}_{\pm9.1}$ | $0.0_{\pm0.0}$ |
| large-diverse | 10.6 | $0.0_{\pm0.0}$ | $0.0_{\pm0.0}$ | 7.2 | $0.0_{\pm0.0}$ | $\mathbf{60.2}_{\pm18.3}$ | $0.5_{\pm1.1}$ |
| AVG | N/A | 0.0 | 0.0 | 28.9 | 37.7 | $\mathbf{64.9}$ | 28.9 |

**Source of benchmarked baseline results.** For the D4RL locomotion benchmark in Tab. 1, we adopt the results of CQL, EDAC, and MOPO from the MOBILE paper (Sun et al., 2023, Table 1), where the MOPO results correspond to the tuned variant (denoted as MOPO* therein). Results for the remaining baselines are taken directly from their respective original publications. For MAPLE, we report the improved variant that employs an ensemble size of 142 (instead of the default 14) as described in (Chen et al., 2021c), to enable a fairer comparison with our method, which uses an ensemble size of 100. Finally, we note that prior works may differ in the total number of gradient steps used for training compared to ours (2M steps in this benchmark); for example, MOPO, MOBILE,

ADMPO, and VIPO report results after 3M steps, while LEQ and SUMO use 1M steps. We retain these numbers as reported, since we believe this setting reflects the most faithful comparison with previously published results.

For the NeoRL locomotion benchmark in Tab. 10, we use EDAC results from the MOBILE paper (Sun et al., 2023), CQL and MOPO from the NeoRL paper (Qin et al., 2022), and COMBO from the MoDAP paper (Choi et al., 2024). All other baselines are taken from their original publications. For MoDAP on Hopper-v3-High, we report the improved result obtained with an ensemble size of 40, as presented by the authors to demonstrate the benefit of larger ensembles.

For the D4RL Adroit benchmark in Tab. 11, we report BC, IQL, and ReBRAC results from the CORL paper (Tarasov et al., 2023b), and MOPO from the MOBILE paper (Sun et al., 2023). Results for other baselines are taken from their original publications. Standard deviations for MOPO and MoMo were not available, so we omit them.

For the D4RL AntMaze benchmark in Tab. 12, we report CBOP and MOBILE from the LEQ paper (Park & Lee, 2025), and a different tuning of MOBILE which we denote as MOBILE[†] from the ADMPO paper (Lin et al., 2025). Results for other baselines are taken from their original publications. We omit standard deviations for MOBILE[†] since they are not reported in the ADMPO paper.

**Statistical tests for benchmarking results.** To assess statistical significance, we conduct Welch's one-sided $t$-test ($p < 0.05$). In the reported tables, we highlight all methods whose average scores are not significantly different from the best-performing method.

## H.2 FULL RESULTS ON COMPOUNDING ERRORS

Fig. 6–Fig. 8 report the full results on compounding errors to complement Fig. 5 in the main paper.

**Plotting setup.** For each world ensemble trained given a dataset, we evaluate compounding errors using *three* datasets that share the same underlying MDP ($\star$-random-v2, $\star$-medium-replay-v2, and $\star$-medium-expert-v2). For each evaluation dataset, we collect 200 rollouts in total (two rollouts per ensemble member). This evaluation protocol allows us to span a broad range of exploratory behaviors, similar to Zhou et al. (2025).

We generate a synthetic rollout as follows: first we draw $m_\theta \sim \mathbf{m}_\theta$ and an entire real trajectory $(s_{0:T}, a_{0:T-1}, r_{1:T})$ from the evaluation dataset. We then let $\hat{s}_0 = s_0$, and $(\hat{r}_{t+1}, \hat{s}_{t+1}) \sim m_\theta(\hat{s}_t, a_t), \forall t < T$. Synthetic rollouts are truncated *only* when numerical overflow occurs (float32); we do not apply an uncertainty threshold and we ignore the terminal function. Because rollout lengths vary in Hopper, we apply forward filling (pandas.DataFrame.ffill) so that medians and percentile statistics remain well-defined. For the leftmost two columns, RMS denotes the root-mean-square, $\mathrm{RMS}(x) = \sqrt{\frac{1}{k}\sum_{i=1}^{k} x_i^2}$ for $x \in \mathbb{R}^k$, which normalizes the $\ell_2$ norm to be dimension-invariant. RMSE denotes the root-mean-square error, $\mathrm{RMSE}(x, y) = \mathrm{RMS}(x - y)$ for $x, y \in \mathbb{R}^k$. For the rightmost scatter plot, we aggregate all state-action pairs from these rollouts and show the relation between estimated uncertainty $U_\theta(\hat{s}_t, a_t)$ and the next-state error $\mathrm{RMSE}(\hat{s}_{t+1}, s_{t+1})$.

**LayerNorm significantly suppresses worst-case (95%) errors across all evaluation setups.** Across the $6 \times 3 = 18$ evaluation setups shown in Fig. 6–Fig. 8, LayerNorm consistently prevents compounding state-error and reward-bias explosion (1st and 3rd columns) by stabilizing the predicted state norm (2nd column). In contrast, without LayerNorm, **16 of the 18** setups exhibit clear error explosion. We further observe that models trained on medium and medium-expert datasets (with narrower coverage) tend to explode more rapidly than those trained on medium-replay datasets (with broader coverage). The only two non-exploding cases without LayerNorm occur when models trained on medium-replay are evaluated with random action sequences, likely because medium-replay offers the broadest coverage and includes random-action trajectories.

**LayerNorm also significantly suppresses medium-case errors.** LayerNorm effectively controls medium-case errors across all evaluation setups. In contrast, without LayerNorm, **12 of the 18** setups exhibit medium-case error explosion (with the remaining 6 showing comparable performance to the LayerNorm variant). Notably, although models trained on random datasets do not explode under random-action rollouts, they *do* explode under higher-quality action sequences when LayerNorm is removed.

When comparing the medium-case error of our baseline model (without LayerNorm) to prior work's baselines, we observe:

- In MOREC (Luo et al., 2024b, Figure 6, without transition filtering), their baseline exhibits compounding MAE of $10^5$ (vs. RMSE of $10^0$ in ours) on halfcheetah-random, $10^5$ (vs. $10^1$) on halfcheetah-medium, $10^3$ (vs. $10^0$) on halfcheetah-medium-replay, and $10^6$ (vs. $\gg 10^6$) on halfcheetah-medium-expert for 100 rollout steps. Thus, our baseline achieves lower errors in three datasets, with one dataset (medium-expert) showing higher RMSE.
- In PGD (Jackson et al., 2024, Figure 6, MOPO), their baseline reports a compounding MSE of 8.0 (vs. RMSE of $10^0$ in ours) for a model trained on halfcheetah-medium and evaluated on halfcheetah-random over 16 rollout steps. Thus, the results are similar.
- In ADMPO (Lin et al., 2025, Figure 2, Ensemble Dynamics Model), their baseline exhibits compounding error of $10^6$ on hopper-medium (vs. $10^0$ in ours) and $10^6$ on hopper-medium-replay (vs. $10^{-1}$) over 50 rollout steps. Thus, our baseline performs better in both settings.
- In D-MPC (Zhou et al., 2025, Figure 2, MLP), their baseline has a RMSE around $10^{-1}$ (vs. RMSE $> 10^1$ in ours) for a model trained on halfcheetah-medium over 256 rollout steps. Thus, our baseline is worse than theirs.

Overall, these results show **mixed baseline performance**: our baseline is generally stronger than those in Luo et al. (2024b); Lin et al. (2025), comparable to Jackson et al. (2024), and weaker than Zhou et al. (2025). Although all baselines use one-step MLP dynamics models, differences in architecture (e.g., D-MPC's 2-layer MLP vs. the 4-layer MOPO-style MLP), initialization, and evaluation protocols likely contribute to the discrepancy. To facilitate consistent future comparisons, we release our model checkpoints and evaluation code.

**Uncertainty threshold can safeguard against compounding error with LayerNorm.** For world ensembles with LayerNorm, we find that the uncertainty threshold $\zeta = 1.0$ (used in our main experiments) reliably separates severe error regions, since $\zeta = 1.0$ approximates the boundary of in-distribution data. While the Spearman's rank coefficient, often used to benchmark uncertainty estimation accuracy (Lu et al., 2021), is less than 0.6 in 5 out of the 18 setups, we argue that this metric is less critical for Bayesian RL. Unlike approaches that penalize with uncertainties at every step, we only use uncertainty as a binary cutoff for truncation. As a result, our method is inherently more robust to imperfect uncertainty ranking, requiring only that severe errors lie beyond the $\zeta = 1.0$ boundary.

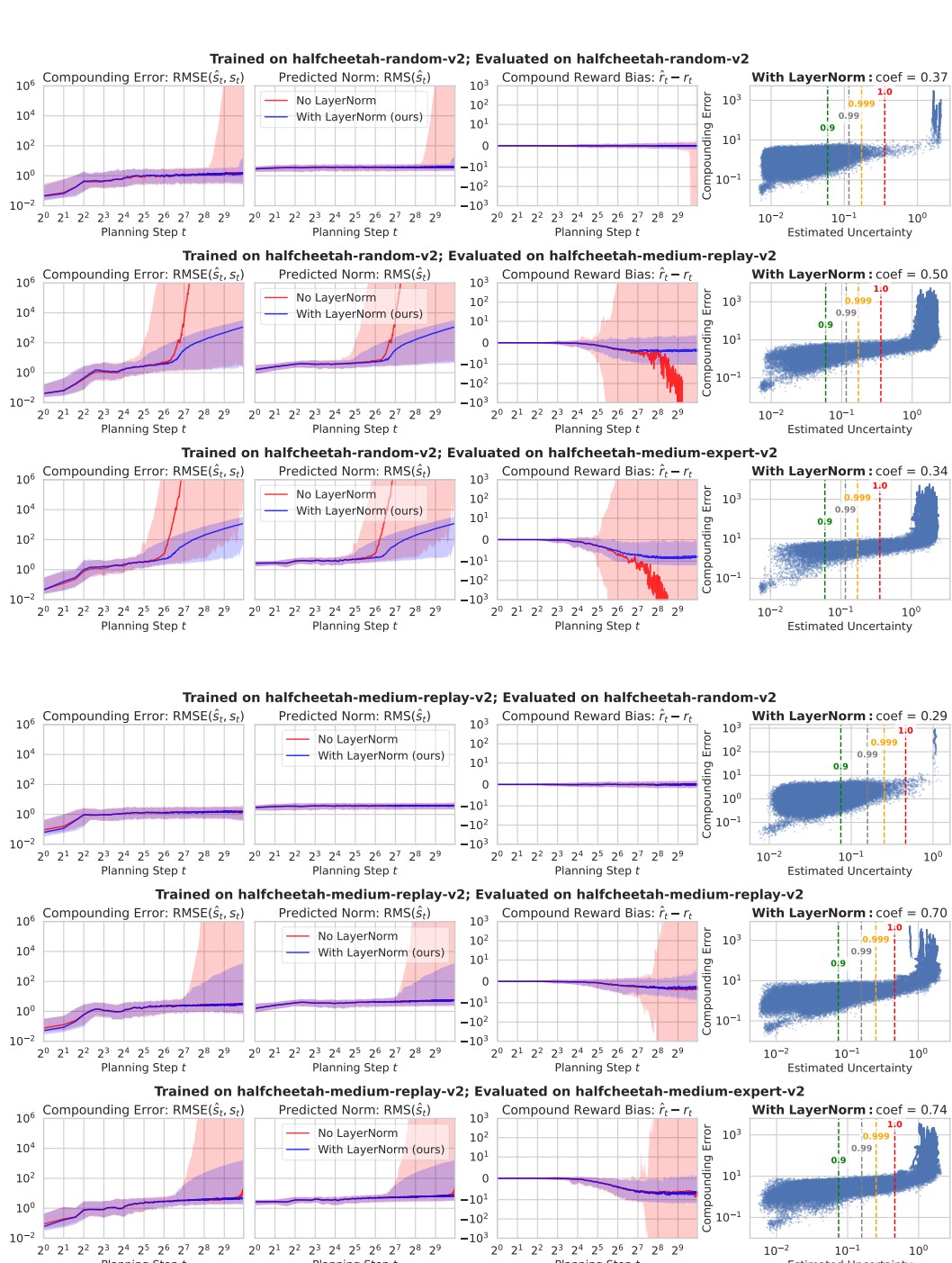

Figure 6: **Effect of LayerNorm in world models trained on halfcheetah-random-v2 and halfcheetah-medium-replay-v2.** For each metric, we plot the **median** (solid line) together with the **5-95% percentile band** across 200 rollouts. The rightmost scatter plots show the Spearman's rank coefficients in the with-LayerNorm setting; vertical lines mark uncertainty thresholds $\zeta \in \{0.9, 0.99, 0.999, 1.0\}$.

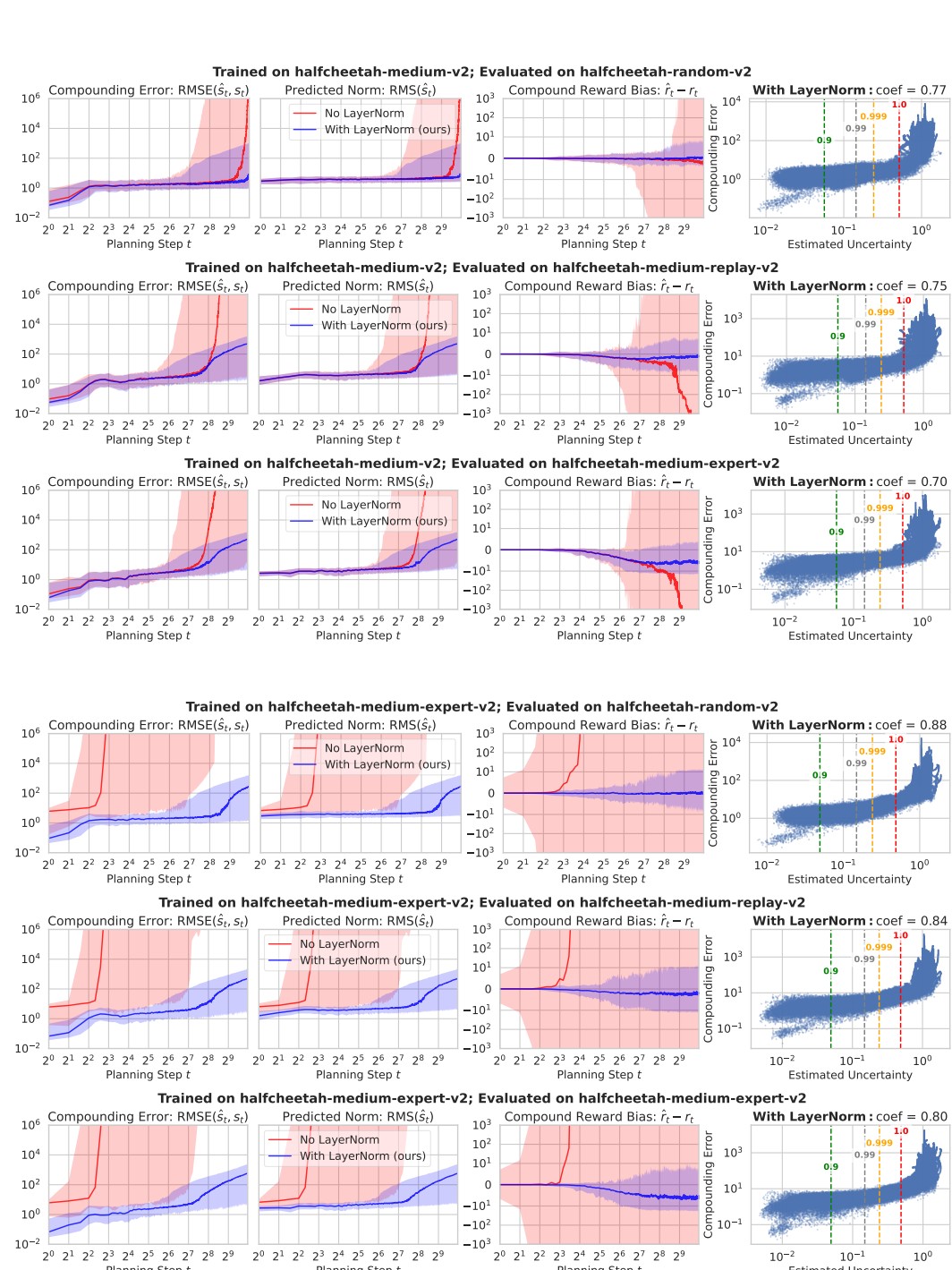

Figure 7: **Effect of LayerNorm in world models trained on halfcheetah-medium-v2 and halfcheetah-medium-expert-v2.**

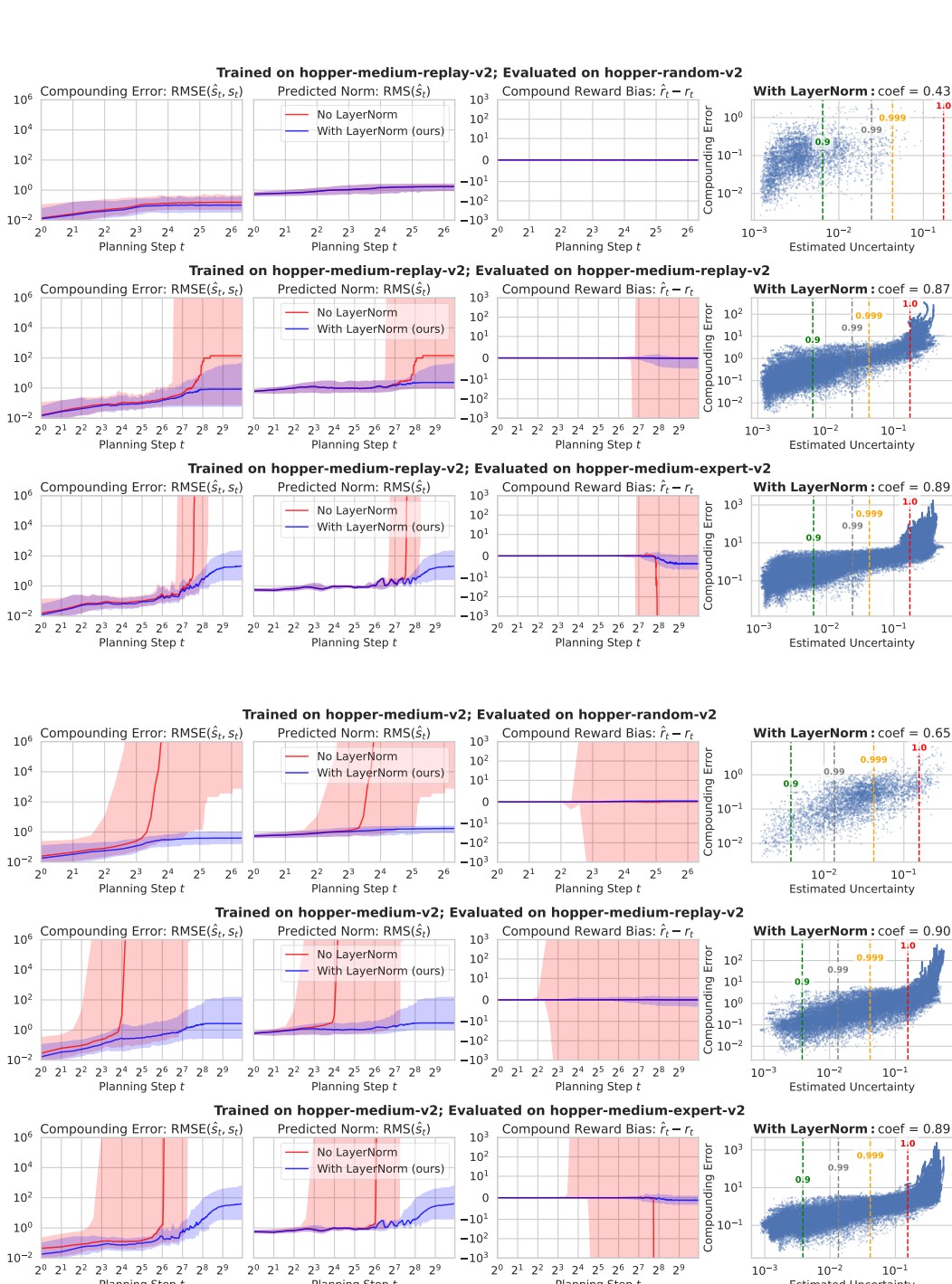

Figure 8: **Effect of LayerNorm in world models trained on hopper-medium-replay-v2 and hopper-medium-v2.**

### H.3 Full Results on Adaptive Long-Horizon Planning

Fig. 9–Fig. 12 report the full ablation results on truncation thresholds to complement Fig. 1, with the maximum rollout horizon for each batch of training rollouts shown in the last columns.

A key observation is that using a quantile threshold of $\zeta = 0.9$ corresponds *roughly* to a horizon cap of 1–10, which mirrors the short fixed horizons commonly adopted in prior work. This shows that such prior choices are not suitable for guiding the design of Bayesian RL, where adaptive long horizons are essential.

**Summary on the effect of $\zeta$.** We further count the number of complete failures (i.e., scores $\leq 5.0$) under different thresholds using Tab. 13. With $\zeta = 0.9$, 16 datasets fail; with $\zeta = 0.99$, 6 datasets fail; and with $\zeta = 0.999$, none fail. This suggests that a safe range for $\zeta$ lies between 0.999 and 1.0. Although $\zeta = 0.999$ often performs similarly to 1.0 (as the resulting adaptive horizons are close), we observe clear advantages of 1.0 on tasks such as D4RL walker2d-random-v2 and pen-cloned-v1. **Thus, we recommend using $\zeta = 1.0$ as a starting point for our algorithm.**

**Summary on horizon scales.** Although NEUBAY uses adaptive horizons, we report the empirical horizon scales (75th-percentile and maximum) to illustrate the typical horizon length required under our Bayesian formulation. In D4RL and NeoRL locomotion tasks ($T = 1000$), NEUBAY uses 75th-percentile horizons of $2^4$-$2^6$ in 4 tasks, $2^7$ in 5 tasks, $2^8$ in 9 tasks, and $2^9$ in 3 tasks; the corresponding maximum horizons of $2^6$-$2^8$ in 4 tasks, $2^9$ in 4 tasks, 1000 in 11 tasks. In Adroit ($T = 100$ or 200), the 75th-percentile horizon is $2^6$-$2^7$, and the maximum horizon reaches the episode length $T$. In AntMaze ($T = 700$ or 1000), the 75th-percentile horizon is $2^4$-$2^8$; the maximum horizon reaches $2^8$-$2^9$. Overall, these statistics show that NEUBAY selects 75th-percentile rollout horizons of **64-512 steps** and maximum horizons of **256-1000 steps**, **in 21 out of 23 tasks with $T = 1000$.**

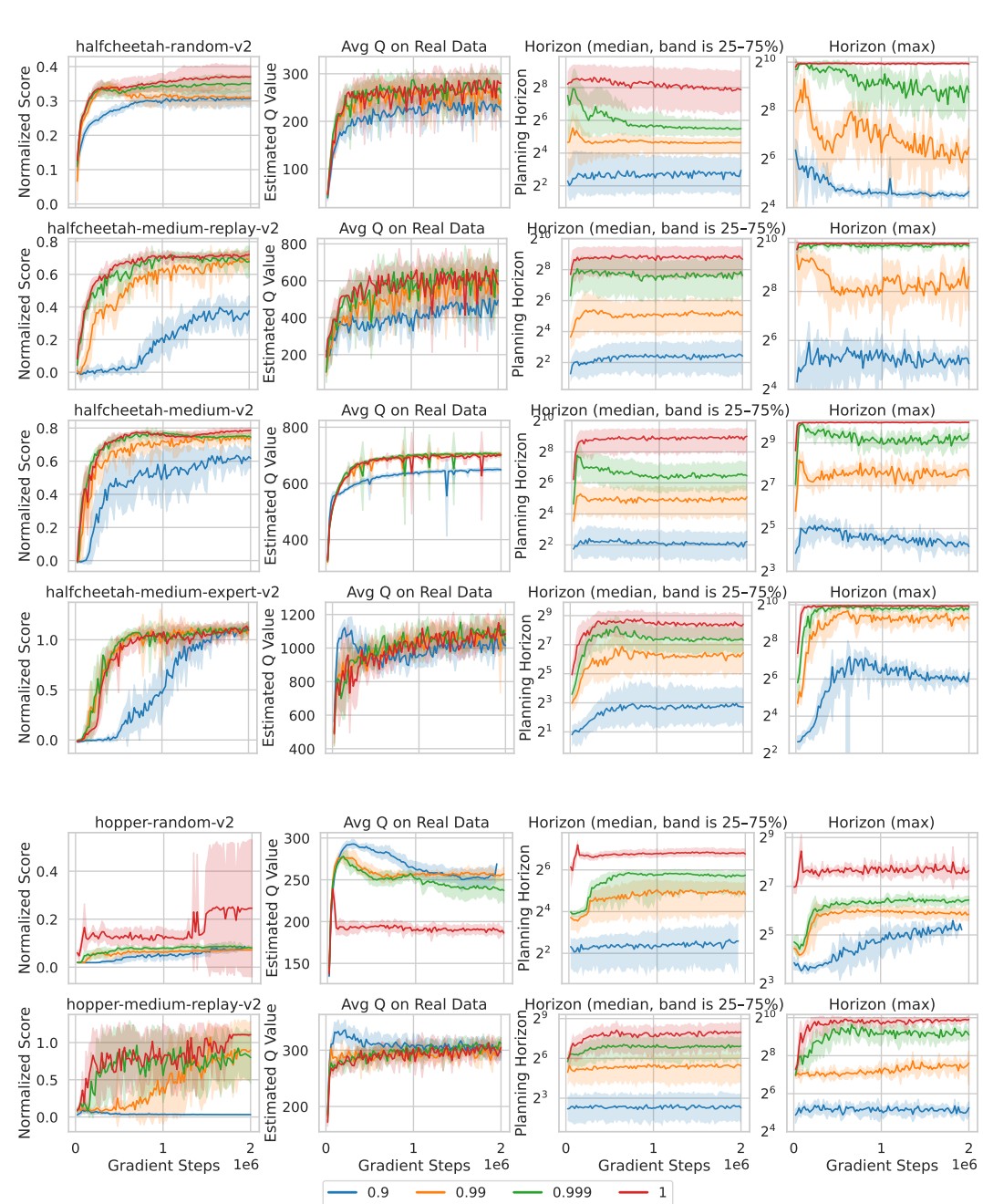

Figure 9: Ablation on the uncertainty quantile $\zeta$ for rollout truncation in D4RL locomotion datasets. Results for the remaining datasets are shown in the next figure.

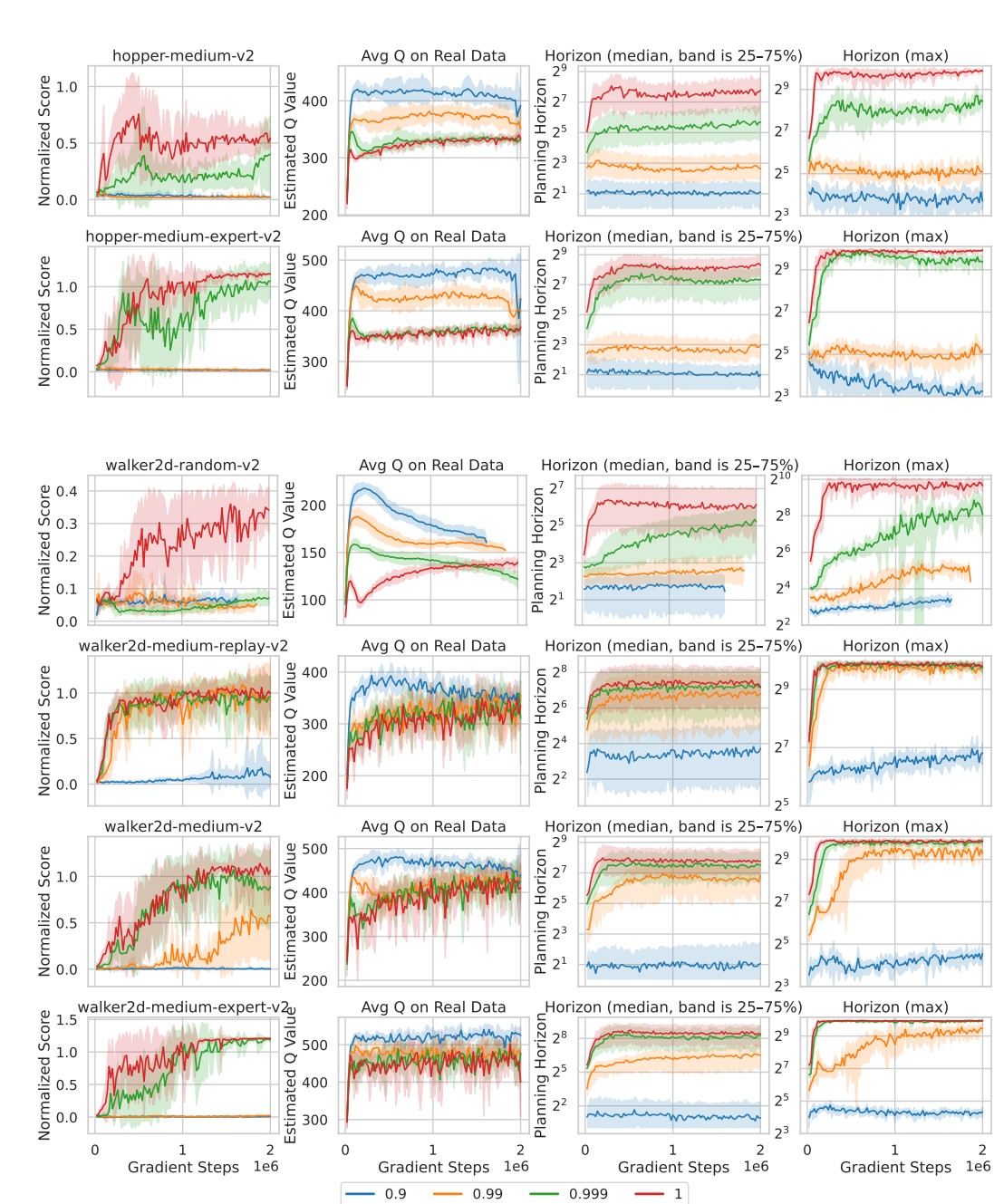

Figure 10: Ablation on the uncertainty quantile $\zeta$ for rollout truncation in D4RL locomotion datasets.

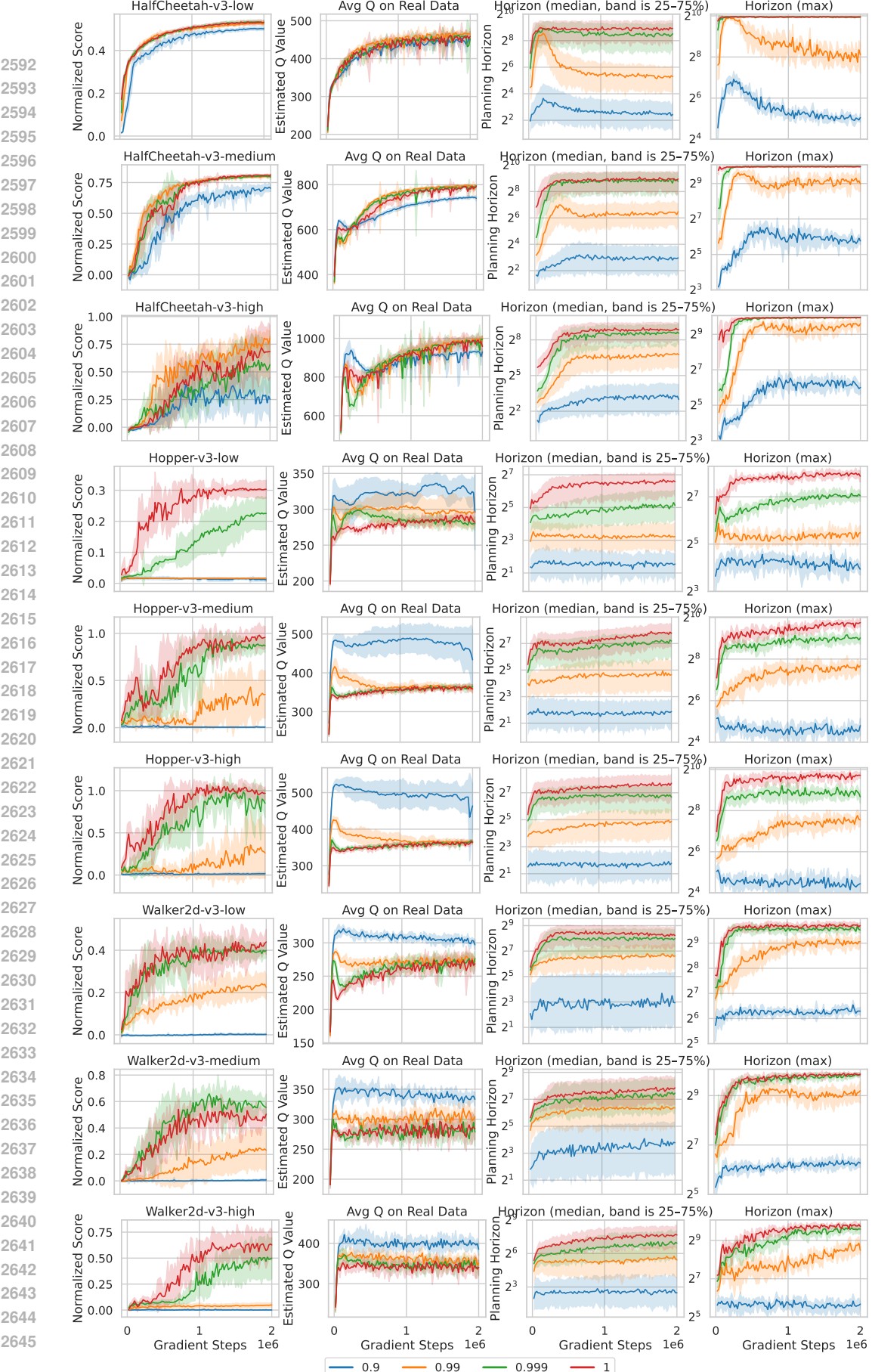

Figure 11: Ablation on the uncertainty quantile $\zeta$ for rollout truncation in NeoRL datasets.

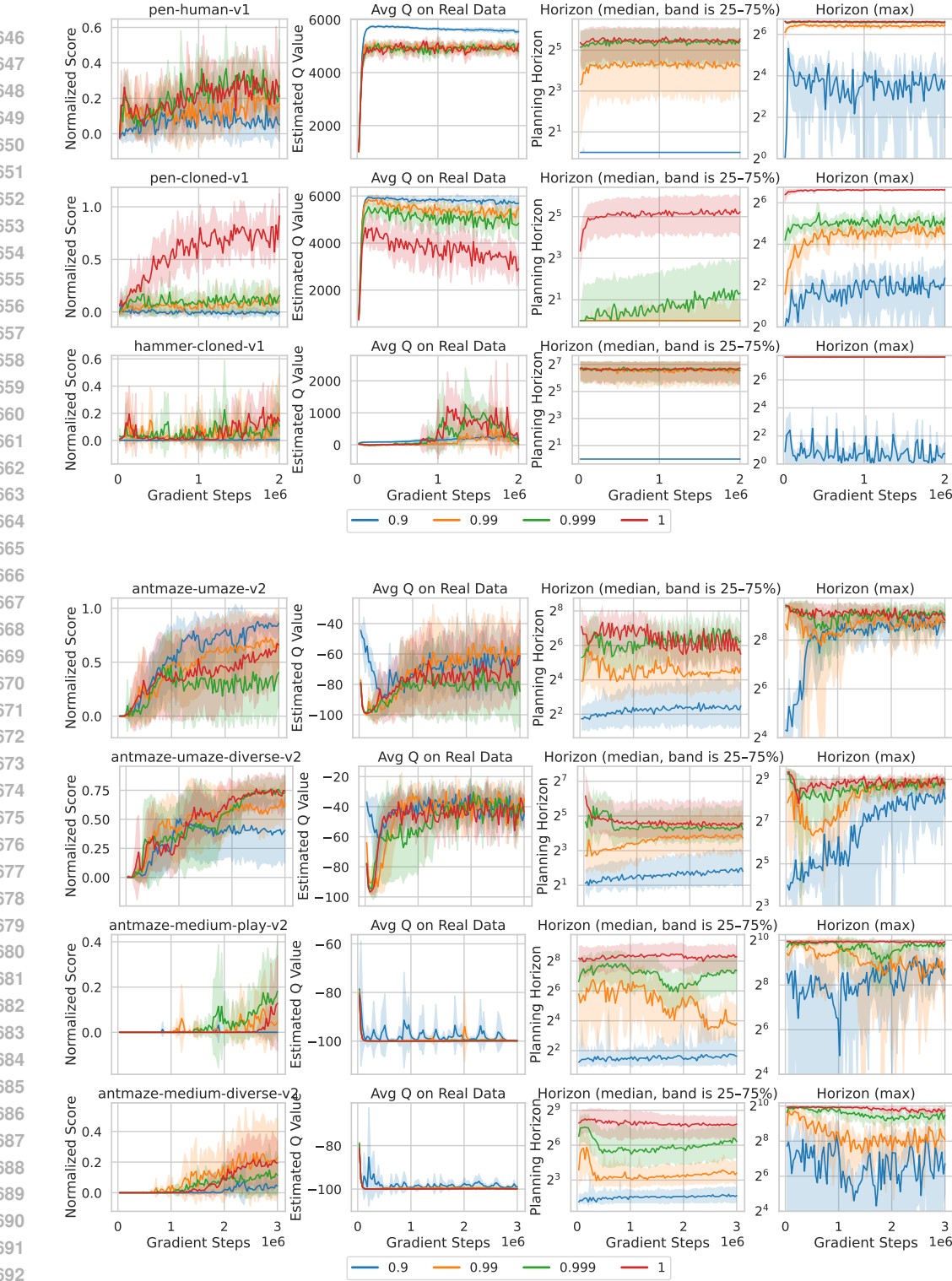

Figure 12: Ablation on the uncertainty quantile $\zeta$ for rollout truncation in three D4RL Adroit datasets and four D4RL AntMaze datasets. Adroit benchmark has short maximum episode steps: $T = 100 < 2^7$ in pen and $T = 200 < 2^8$ in hammer, which limits the rollout horizon. Maximum episode steps are $T = 700$ in umaze and $T = 1000$ in medium maze. Results on the remaining Adroit and AntMaze datasets are omitted as our algorithm has near-zero performance. Note that in AntMaze, successful episodes terminate early, so horizon lengths are partially confounded by this effect.

## H.4 SENSITIVITY AND ABLATION RESULTS

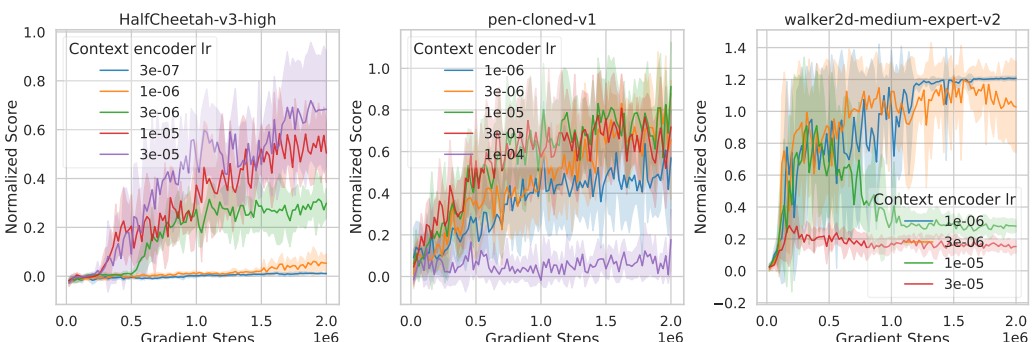

Figure 13: Selective learning curves on datasets where performance is *sensitive* to the **context encoder learning rate**, favoring high (left), medium (middle), and low (right) values.

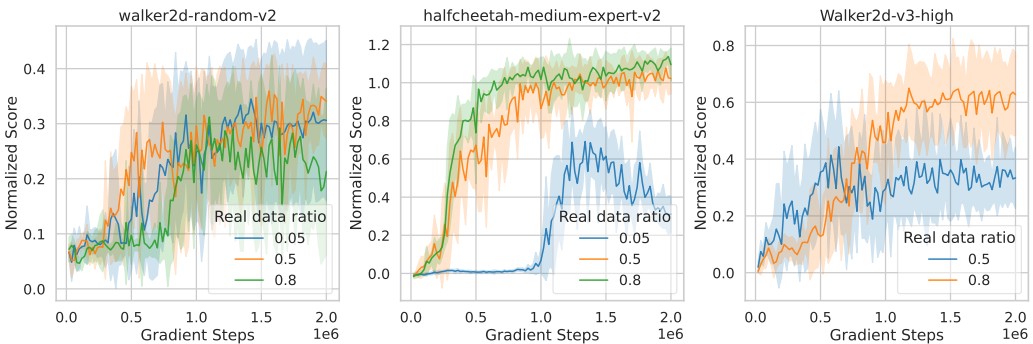

Figure 14: Selective learning curves on datasets where performance is *sensitive* to the **real data ratio**.

## H.5 FAILURE-CASE ANALYSIS IN ANTMAZE

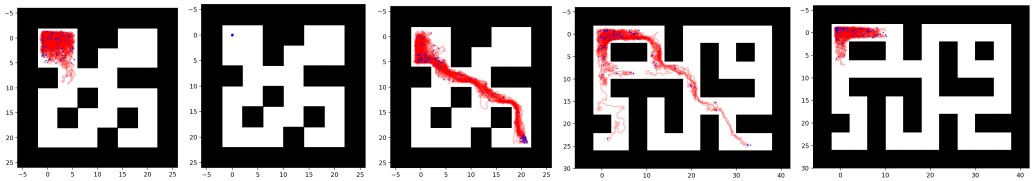

Figure 15: Failure cases in antmaze-medium-play-v2 (left three; different seeds) and antmaze-large-diverse-v2 (right two). The agent starts at the top-left corner and the goal is at the bottom-right. We show 100 evaluation trajectories in red and mark their endpoints in blue.

Fig. 15 shows typical failure cases for NEUBAY in AntMaze. Since the environment provides no reward until the goal, the agent receives almost no learning signal by interaction. As a result, NEUBAY frequently remains near the initial corner or repeatedly collides with nearby walls, resulting in degenerate behaviors. Importantly, introducing conservatism, such as adding uncertainty penalties, in fact degrades performance (see Tab. 13).

Based on these observations, we attribute the failures to two factors: (1) lack of effective exploration in large mazes, (2) long-horizon modeling challenges on contact-rich dynamics. We expect NEUBAY to benefit from stronger planning and modeling components, such as beam-search planning (Janner et al., 2021), more stable exploration strategy (Park & Lee, 2025), and multi-step world models (Lin et al., 2025).

## I THE USE OF LARGE LANGUAGE MODELS (LLMS)

We used OpenAI's ChatGPT as a general-purpose assistant. Specifically, it was used for (1) writing support (improving clarity, conciseness, and tone), and (2) technical coding assistance (code snippets, plotting utilities, and minor debugging). In all cases, the authors verified correctness and remained

Table 13: **Sensitivity and ablation results per dataset.** The highlighted setting ($N$=100, $\lambda$=0.0, $\zeta$=1.0, using the entire history as agent input) is the main result. Ablations vary one hyperparameter at a time, except for the Markov agent, where we sweep the real data ratio $\kappa$ for a fair comparison. Red shading shows **degradation** level: light (3–10), medium (10–30), dark (>30). Green shading shows **improvement** level: light (3–10), medium (10–30), dark (>30). Tab. 2 is a summary.

| Dataset | Ensemble size $N$ | | | Unc. penalty coef. $\lambda$ | | | | | Truncation threshold $\zeta$ | | | | Agent input | |
|---|---|---|---|---|---|---|---|---|---|---|---|---|---|---|
| | 100 | 20 | 5 | 0.0 | 0.04 | 0.2 | 1.0 | 5.0 | 1.0 | 0.999 | 0.99 | 0.9 | Hist. | Mark. |
| hc-random-v2 | 37.0 | 39.7 | 38.2 | 37.0 | 35.6 | 32.7 | 19.3 | 1.2 | 37.0 | 35.0 | 31.2 | 30.7 | 37.0 | 44.4 |
| hp-random-v2 | 24.5 | 15.1 | 13.3 | 24.5 | 48.2 | 40.1 | 31.6 | 18.8 | 24.5 | 8.0 | 7.1 | 8.7 | 24.5 | 21.7 |
| wk-random-v2 | 34.1 | 20.5 | 8.2 | 34.1 | 33.0 | 34.2 | 23.3 | 0.0 | 34.1 | 7.1 | 5.3 | 6.0 | 34.1 | 33.4 |
| hc-med-rep-v2 | 72.1 | 68.6 | 66.7 | 72.1 | 72.0 | 70.1 | 63.5 | 52.1 | 72.1 | 67.8 | 68.6 | 37.7 | 72.1 | 76.9 |
| hp-med-rep-v2 | 110.6 | 81.8 | 75.2 | 110.6 | 110.9 | 97.6 | 95.3 | 28.8 | 110.6 | 79.8 | 89.7 | 3.2 | 110.6 | 47.2 |
| wk-med-rep-v2 | 99.3 | 91.3 | 97.8 | 99.3 | 87.4 | 93.5 | 86.7 | 72.8 | 99.3 | 96.0 | 92.3 | 7.4 | 99.3 | 112.8 |
| hc-medium-v2 | 78.6 | 73.7 | 74.2 | 78.6 | 77.5 | 70.7 | 59.8 | 54.6 | 78.6 | 74.5 | 74.0 | 61.8 | 78.6 | 78.2 |
| hp-medium-v2 | 54.2 | 37.1 | 48.8 | 54.2 | 52.2 | 105.8 | 74.0 | 64.8 | 54.2 | 40.3 | 2.3 | 2.2 | 54.2 | 47.0 |
| wk-medium-v2 | 106.4 | 103.1 | 55.8 | 106.4 | 96.6 | 77.9 | 93.6 | 81.9 | 106.4 | 88.9 | 56.5 | 0.2 | 106.4 | 112.4 |
| hc-med-exp-v2 | 109.4 | 107.3 | 97.8 | 109.4 | 107.7 | 112.7 | 110.3 | 96.4 | 109.4 | 112.3 | 109.6 | 109.6 | 109.4 | 111.9 |
| hp-med-exp-v2 | 114.8 | 100.2 | 96.8 | 114.8 | 114.4 | 113.5 | 110.3 | 110.1 | 114.8 | 106.5 | 2.2 | 1.8 | 114.8 | 101.3 |
| wk-med-exp-v2 | 120.6 | 120.6 | 118.5 | 120.6 | 121.5 | 116.2 | 109.3 | 107.4 | 120.6 | 118.5 | 2.0 | 1.1 | 120.6 | 122.2 |
| hc-v3-Low | 53.0 | 51.4 | 52.2 | 53.0 | 52.3 | 47.3 | 37.8 | 26.3 | 53.0 | 53.3 | 52.7 | 50.1 | 53.0 | 55.4 |
| hp-v3-Low | 30.3 | 26.6 | 29.3 | 30.3 | 30.0 | 24.3 | 15.5 | 12.7 | 30.3 | 22.5 | 1.6 | 1.1 | 30.3 | 29.4 |
| wk-v3-Low | 43.9 | 41.9 | 35.4 | 43.9 | 46.1 | 45.7 | 53.0 | 53.2 | 43.9 | 39.8 | 22.5 | 0.2 | 43.9 | 57.4 |
| hc-v3-Med | 81.1 | 80.5 | 77.3 | 81.1 | 81.2 | 79.3 | 68.1 | 56.6 | 81.1 | 79.5 | 79.9 | 70.4 | 81.1 | 82.5 |
| hp-v3-Med | 95.7 | 88.6 | 94.1 | 95.7 | 89.3 | 81.6 | 72.1 | 33.4 | 95.7 | 86.8 | 35.2 | 0.6 | 95.7 | 94.9 |
| wk-v3-Med | 50.5 | 34.4 | 39.9 | 50.5 | 43.9 | 50.2 | 44.8 | 40.9 | 50.5 | 55.5 | 23.1 | 0.7 | 50.5 | 47.1 |
| hc-v3-High | 68.3 | 59.9 | 53.8 | 68.3 | 65.9 | 67.5 | 71.1 | 54.1 | 68.3 | 56.7 | 80.4 | 25.2 | 68.3 | 81.5 |
| hp-v3-High | 96.8 | 100.4 | 99.3 | 96.8 | 89.6 | 75.3 | 90.2 | 46.4 | 96.8 | 85.0 | 26.1 | 1.6 | 96.8 | 80.9 |
| wk-v3-High | 62.7 | 59.1 | 57.0 | 62.7 | 55.3 | 67.2 | 72.2 | 58.5 | 62.7 | 50.0 | 4.8 | 0.0 | 62.7 | 72.4 |
| pen-human-v1 | 20.8 | 19.3 | 13.7 | 20.8 | 25.8 | 24.0 | 35.9 | 34.8 | 20.8 | 27.2 | 16.8 | 5.0 | 20.8 | 28.0 |
| pen-cloned-v1 | 91.3 | 63.4 | 75.6 | 91.3 | 68.2 | 76.0 | 77.2 | 67.2 | 91.3 | 15.7 | 10.6 | 0.2 | 91.3 | 75.6 |
| hammer-cloned-v1 | 14.4 | 7.4 | 8.6 | 14.4 | 7.0 | 2.7 | 1.4 | 0.2 | 14.4 | 10.5 | 20.5 | 0.6 | 14.4 | 5.6 |
| umaze-v2 | 66.1 | 71.1 | 65.3 | 66.1 | 19 | 9.3 | 16.7 | 28.4 | 66.1 | 40.6 | 62.6 | 86.0 | 66.1 | 0.0 |
| umaze-diverse-v2 | 74.4 | 52.5 | 37.8 | 74.4 | 0.8 | 3.3 | 0.2 | 19.8 | 74.4 | 73.0 | 63.3 | 40.4 | 74.4 | 5.2 |
| medium-play-v2 | 12.8 | 2.2 | 0.8 | 12.8 | 0.0 | 0.0 | 0.0 | 0.0 | 12.8 | 18.2 | 4.5 | 0.0 | 12.8 | 0.0 |
| medium-diverse-v2 | 19.4 | 6.8 | 11.3 | 19.4 | 0.0 | 0.0 | 0.0 | 0.0 | 19.4 | 11.4 | 23.1 | 4.5 | 19.4 | 0.0 |

fully responsible for research ideation, algorithm design, code implementation, experimental results, and paper writing.

