# OpenReview forum: "Long-Horizon Model-Based Offline Reinforcement Learning Without Conservatism"
_ICLR.cc/2026/Conference — Submitted to ICLR 2026_

### Official Review · Reviewer_ctso · 2025-10-19

**Soundness:** 4
**Presentation:** 4
**Contribution:** 3
**Rating:** 8
**Confidence:** 4

**Summary:**

This paper questions the dominance of the conservative principle in offline reinforcement learning (RL) and instead explores a Bayesian alternative. The authors propose NEUBAY, a model-based offline RL algorithm derived from a neutral Bayesian (NEUtral BAYesian) perspective that optimizes expected returns over a posterior distribution of world models, rather than enforcing pessimism or short horizons. Key innovations include (1) using layer normalization in world models to control compounding error, (2) adaptive long-horizon planning based on epistemic uncertainty thresholds, and (3) a stabilized recurrent training pipeline for history-dependent agents. NEUBAY is evaluated on D4RL and NeoRL benchmarks across 33 datasets and achieves new state-of-the-art results on 7 datasets, particularly excelling on low-quality or moderate-coverage data (see Table 1; Sec. 5.1).

**Strengths:**

- The core problem addressed in this paper is how to abandon the reliance on conservatism in offline RL algorithms, which I believe is a very meaningful issue. Conservatism constrains policy optimization within a limited range, inevitably leading to suboptimal policies with poor generalization ability. The proposed NEUBAY method breaks this restriction and demonstrates that even without conservatism, offline RL can still perform well, offering a viable alternative paradigm.
- The paper, through the bandit experiment, clearly demonstrates how the uncertainty penalty harms generalization performance, while Bayesianism can effectively adapt to various cases, thereby validating that the authors’ claim is both correct and reasonable.
- NEUBAY is compared against a wide range of baselines, including numerous recent model-free and model-based offline RL algorithms. Ultimately, NEUBAY achieves outstanding performance on most tasks. Such comprehensive comparisons and significant performance improvements clearly demonstrate the effectiveness of the proposed method.
- The authors designed several ablation studies targeting their design choices, showing the contribution of each component in the method. These results provide strong support for the soundness and rationale of the proposed design choices.

**Weaknesses:**

- NEUBAY uses a larger ensemble size in the dynamics model compared to previous methods, which seems to introduce additional computational overhead, including increased runtime and GPU memory usage.
- I think the NEUBAY method is overall relatively complex and introduces some new hyperparameters, which seems to make the algorithm more difficult to tune.

**Questions:**

- Could the authors provide a comparison of NEUBAY with previous methods in terms of time and GPU memory consumption?
- NEUBAY’s actor and critic utilize historical information. How would NEUBAY’s performance change if historical information were not used? If the actors and critics of other methods also used historical information, would NEUBAY’s performance advantage still be significant?”

---

> ### Author Response · Authors · 2025-11-23
> **Response (part 1)**
>
> Thank you for your detailed review and constructive feedback. We respond to your concerns below, and we kindly invite you to also consult the updated manuscript and the common response.
>
> ### 1. Computation costs
>
> > NEUBAY uses a larger ensemble size in the dynamics model compared to previous methods, which seems to introduce additional computational overhead, including increased runtime and GPU memory usage.
> > Could the authors provide a comparison of NEUBAY with previous methods in terms of time and GPU memory consumption?
>
> We provide full runtime and memory details in the **Common response**. In short, even with a large ensemble of $N=100$, NEUBAY remains efficient: the recurrent agent trains **faster than MOPO’s PyTorch implementation** (4.4 hrs vs. 5.6 hrs for 2M steps), and the rollout function scales very well due to JAX vectorization. Increasing $N$ from 5 to 100 raises rollout time by only 0.6 seconds, making the overhead negligible. Switching from a Markov to a recurrent agent adds only **1.8 hours** and **1.1 GB** of GPU memory.
>
> Thus, in our implementation, large ensemble sizes do *not* significantly increase computational cost.
>
> ### 2. Complexity and practicality
>
> > I think the NEUBAY method is overall relatively complex and introduces some new hyperparameters, which seems to make the algorithm more difficult to tune.
>
> **NEUBAY introduces two task-dependent hyperparameters** (real data ratio $\kappa$ and encoder learning rate $\eta_\phi$). This is **comparable to** popular offline model-based RL methods:
>
> | Method | Task-dependent hyperparameters | Count |
> | - | -| -|
> | NEUBAY (ours) | real data ratio, encoder learning rate | 2 |
> | MOPO [Yu et al., 2020] | rollout horizon, penalty coefficient | 2|
> | COMBO [Yu et al., 2021] | rollout horizon, learning rate, conservative coefficient, $\rho(s,a)$, $\mu (a\mid s)$, real data ratio | 6|
> | RAMBO [Rigter et al., 2022] | rollout horizon, adversarial loss coefficient | 2|
> | MOBILE [Sun et al., 2023] | rollout horizon, penalty coefficient | 2|
>
> Importantly, NEUBAY **removes** two commonly tuned hyperparameters in prior work:
> - *conservatism coefficient* (fixed at 0.0),
> - *horizon length* (adaptively chosen by a fixed uncertainty quantile of 1.0).
>
> Thus, the total number of tuned hyperparameters is exactly two, on par with popular offline model-based RL methods.
>
> To facilitate tuning in practice, in the rebuttal, we have:
> - **open-sourced** the full codebase,
> - provided **pretrained ensemble checkpoints** (linked in the readme), so that users can train agents directly on them, and
> - added clear **tuning guidelines** in Section 6 of the updated PDF.
>
> With these resources, NEUBAY is simple to run and tune in practice.
>
> ### 3. Ablation on using a Markov agent in NEUBAY
>
> > NEUBAY’s actor and critic utilize historical information. How would NEUBAY’s performance change if historical information were not used?
>
> We conducted an ablation in which we replace the history-dependent agent with a Markov agent (standard MLP actor–critic) across all datasets. We sweep over the same real-data-ratio range for fair comparison, and report the best results in **Table 2** (**summary**; also copied below) and **Table 13 (per-dataset)**.
>
> | Benchmark                     | History-dependent | Markov |
> |-|-|-|
> | D4RL Locomotion (12 tasks)   | 80.1    | 75.8         |
> | NeoRL Locomotion (9 tasks)   | 64.7    | 66.8          |
> | D4RL Adroit (3 tasks)        | 42.2    | 36.4          |
> | D4RL AntMaze (4 tasks)       | 43.2    | 1.3          |
>
>
> We find that the Markov version performs **similarly** to the history-dependent one on most *locomotion* tasks, especially on the NeoRL benchmark (64.7 $\to$ 66.8). However, it suffers **severe degradation** on hopper-medium-replay-v2 (110.6 $\to$ 47.2) and on *all* AntMaze tasks (43.2 $\to$ 1.3 on average).
>
> This pattern suggests that in locomotion domains, epistemic uncertainty is relatively mild, so using a single observation may often infer the model index, making memory less critical. As a result, Bayesian RL with a Markov agent remains a strong baseline in these settings. In contrast, AntMaze has high epistemic uncertainty, especially about the maze layout which is crucial to navigation, so memory is needed to accumulate information over time.
>
> Finally, in the bandit setting of Sec. 3, a Markov agent has no informative input, so the optimal Markov policy under NEUBAY simply selects the arm **with the largest reward-mean** under the reward ensemble (arm 0 according to Fig. 2). This collapses to a constant policy (i.e., $\pi^*_{\text{Markov}} = 0$) and illustrates why memory is necessary in that example.

---

> > ### Author Response · Authors · 2025-11-23
> > **Response (part 2)**
> >
> > ### 4. Using a history-dependent agent in prior work
> >
> > > If the actors and critics of other methods also used historical information, would NEUBAY’s performance advantage still be significant?”
> >
> > We address this question in two parts.
> >
> > **(1) Prior Bayesian-inspired methods already use history, yet underperform NEUBAY.**
> > Methods such as MAPLE [Chen et al., 2021] and MoDAP [Choi et al., 2024] both employ history-dependent agents. Despite this, they generally underperform NEUBAY across benchmarks. This suggests that NEUBAY’s gains over them do not primarily stem from using history, but rather from its world modeling and adaptive long-horizon planning.
> > In our bandit example, the conservative baseline also uses the same memory encoder, ensuring a fair comparison.
> >
> > **(2) For conservative methods with Markov agents, adding memory could help.**
> > Robust MDP theory [Wiesemann et al., 2013] (see [their Table 1](https://optimization-online.org/wp-content/uploads/2010/05/2610.pdf#page=13)) shows that when the uncertainty set is **non-rectangular**, meaning the adversarial model cannot be chosen *independently* at each state–action pair, the optimal policy is generally **history-dependent**. This situation arises in many conservative offline RL formulations (e.g., RAMBO [Rigter et al., 2022]), where the adversarial model is fixed given the policy. In these settings, a memoryless (Markov) agent may be suboptimal, so using history could be beneficial.
> >
> > That said, the magnitude of the improvement would depend on the specific implementation and task, and therefore cannot be predicted universally.
> >
> >
> > ---
> >
> > Thank you again for your time and constructive feedback. We hope our responses and revisions address your concerns, and we would be happy to clarify anything further if needed.
> >
> > ### References
> > - Tianhe Yu, Garrett Thomas, Lantao Yu, Stefano Ermon, James Y Zou, Sergey Levine, Chelsea Finn, and Tengyu Ma. Mopo: Model-based offline policy optimization. NeurIPS 2020.
> > - Tianhe Yu, Aviral Kumar, Rafael Rafailov, Aravind Rajeswaran, Sergey Levine, and Chelsea Finn. Combo: Conservative offline model-based policy optimization. NeurIPS 2021.
> > - Marc Rigter, Bruno Lacerda, and Nick Hawes. Rambo-rl: Robust adversarial model-based offline reinforcement learning. NeurIPS 2022.
> > - Yihao Sun, Jiaji Zhang, Chengxing Jia, Haoxin Lin, Junyin Ye, and Yang Yu. Model-bellman inconsistency for model-based offline reinforcement learning. ICML 2023.
> > - Xiong-Hui Chen, Yang Yu, Qingyang Li, Fan-Ming Luo, Zhiwei Qin, Wenjie Shang, and Jieping Ye. Offline model-based adaptable policy learning. NeurIPS 2021.
> > - Yunseon Choi, Li Zhao, Chuheng Zhang, Lei Song, Jiang Bian, and Kee-Eung Kim. Diversification of adaptive policy for effective offline reinforcement learning. IJCAI 2024.
> > - Wolfram Wiesemann, Daniel Kuhn, and Berç Rustem. Robust markov decision processes. Mathematics of Operations Research, 2013.

---

> > > ### Comment · Reviewer_ctso · 2025-11-23
> > >
> > > Thank you for your detailed response! My concerns are all addressed. I would like to maintain my initial score of 8.

---

### Official Review · Reviewer_Ckqf · 2025-10-31

**Soundness:** 2
**Presentation:** 2
**Contribution:** 2
**Rating:** 2
**Confidence:** 4

**Summary:**

This work proposes NEUBAY, a Bayesian-perspective algorithm that addresses epistemic uncertainty in offline datasets by modeling a posterior distribution. The key idea builds on the controlling compounding error in epistemic POMDP modeling (increasing the ensemble size and incorporating the layer normalization in the model) and addressing the long-horizon planning. In the experiments, NEUBAY outperforms for several offline datasets, but not for most of the offline datasets.

**Strengths:**

1. The authors conducted several ablation studies, including variations in ensemble size and truncation threshold, to demonstrate the strengths of the proposed algorithm.

2. Providing comprehensive comparisons against a wide range of offline RL baselines effectively supports the validity of the proposed approach.

**Weaknesses:**

1. To the best of my knowledge, there exists a method that unrestricts the planning horizon in offline model-based RL. MBOP [1] and MOPP [2] encourage more aggressive trajectory rollouts. In particular, MOPP prunes out problematic trajectories to avoid potential out-of-distribution samples. This approach is highly similar to Algorithm 2 in NEUBAY, as both involve pruning (or truncating) trajectories based on a threshold and performing rollouts without a fixed planning horizon. Overall, the two methods share a significant conceptual overlap.

2. In Figure 5, it is difficult to interpret how the inclusion of layer normalization affects the model’s performance. For example, there are too many colored curves, making it unclear which line corresponds to which experimental setting.

3. In Section 3, the authors emphasize that the Bayesian framework enables the model to perform well even with low-quality data. However, it is unclear whether the method consistently demonstrates superior performance compared to other algorithms on the D4RL MuJoCo random datasets in Table 1.

4. The ensemble size was set to 100. This is similar to the perspective in model-free algorithms such as SAC-N, where a large ensemble contributes to improved performance. Therefore, it would be valuable to demonstrate whether the proposed method (NEUBAY) still performs well when the ensemble size is reduced to a comparable level (e.g., 5 or 7), as done in other algorithms. This would be similar in spirit to how EDAC was motivated by the idea of SAC-N.

5. It is also important to include a comparison of running time with respect to the ensemble size, as increasing the ensemble size can significantly affect computational efficiency.


[1] Argenson, Arthur, and Gabriel Dulac-Arnold. "Model-based offline planning." arXiv preprint arXiv:2008.05556 (2020).

[2] Zhan, Xianyuan, Xiangyu Zhu, and Haoran Xu. "Model-based offline planning with trajectory pruning." arXiv preprint arXiv:2105.07351 (2021).

**Questions:**

The issues discussed under Weakness capture and represent the key questions regarding this paper.

---

> ### Author Response · Authors · 2025-11-23
> **Response (part 1)**
>
> Thank you for your detailed review and constructive feedback. We respond to your concerns below, and we kindly invite you to also consult the updated manuscript and the common response.
>
> ### 1. Is NEUBAY similar to MBOP and MOPP?
>
> > (1) To the best of my knowledge, there exists a method that unrestricts the planning horizon in offline model-based RL. MBOP [1] and MOPP [2] encourage more aggressive trajectory rollouts. In particular, MOPP prunes out problematic trajectories to avoid potential out-of-distribution samples. This approach is highly similar to Algorithm 2 in NEUBAY, as both involve pruning (or truncating) trajectories based on a threshold and performing rollouts without a fixed planning horizon. Overall, the two methods share a significant conceptual overlap.
> >
> > [1] Argenson, Arthur, and Gabriel Dulac-Arnold. "Model-based offline planning." arXiv preprint arXiv:2008.05556 (2020).
> >
> > [2] Zhan, Xianyuan, Xiangyu Zhu, and Haoran Xu. "Model-based offline planning with trajectory pruning." arXiv preprint arXiv:2105.07351 (2021).
>
> We believe part of the concern comes from focusing on our rollout subroutine (Algorithm 2), rather than the full algorithmic pipeline (Algorithm 1). While NEUBAY, MBOP, and MOPP are all offline model-based methods, **NEUBAY is conceptually different from MBOP and MOPP in several core aspects**, summarized below:
>
> | Algorithm | Timing of planning | Base planner | Conservative? | Maximum horizon | Uncertainty threshold |
> |-|-|-|-|-|-|
> |MBOP [1] | Decision-time | MPC | Yes | 10 | N/A | N/A |
> |MOPP [2]| Decision-time | MPC | Yes | 16 | 85\% quantile of offline dataset
> |NEUBAY (ours) | Background | Dyna-style | No (Bayesian) | 1000 | Max of offline dataset |
>
>
> 1. Different planning paradigm: decision-time vs. background (using terms in Chapter 8.8 of [Sutton & Barto, 2018]). MBOP and MOPP perform **decision-time planning**: planning occurs *only at test time*, using model predictive control (MPC) to search for the best action in the current state. In contrast, NEUBAY performs **background (Dyna-style) planning**: planning occurs *only at training time* to update a global policy and value function. At test time, NEUBAY *does no planning* and simply executes the learned policy.
> 2. Conservative vs. non-conservative formulation. MBOP and MOPP are **conservative**: they rely on behavior-cloned policy priors for sampling actions and pessimistic value functions for evaluating trajectories. NEUBAY instead follows **Bayesian, non-conservative** principle: no behavior cloning, no value regularization, and no uncertainty penalty. This is a substantial departure in the principle. As shown in our bandit example (Sec. 3), NEUBAY is explicitly designed to avoid conservatism in order to improve generalization.
> 3. Different horizon scales. Although MOPP uses an uncertainty threshold, it **restricts the maximum horizon at 16** (see [their Table 3](https://arxiv.org/pdf/2105.07351#page=8)). NEUBAY **removes the maximum horizon** and allows rollouts up to the full episode length (1000). In practice, NEUBAY commonly uses horizons of 64–512 steps (Appendix H.3), far beyond the range used by MBOP or MOPP.
>
> We updated the PDF to cite MBOP and MOPP, acknowledge the similarity in adaptive truncation, and clearly differentiate the broader methodology. Given the substantial differences in planning paradigm, conservative principle, and horizon scale, we view NEUBAY as a **conceptually distinct** approach.
>
> ### 2. Compounding error plot
>
> > (2) In Figure 5, it is difficult to interpret how the inclusion of layer normalization affects the model’s performance. For example, there are too many colored curves, making it unclear which line corresponds to which experimental setting.
>
> We agree that the original figure was cluttered due to many overlapping curves. In the revised version (Fig. 5 and Appendix Fig. 6), we now replace individual curves with the **median and 5–95% percentile band**, which makes the effect of LayerNorm much clearer.
>
> In Fig. 5 (halfcheetah-medium-expert-v2), LayerNorm consistently reduces both the *median* error and the *worst-case* (95th percentile) error by preventing the state norm from explosion. The reward-bias curves show the same pattern. Fig. 6 further shows that this reduction in worst-case error holds broadly across additional D4RL datasets.

---

> ### Author Response · Authors · 2025-11-23
> **Response (part 2)**
>
> ### 3. Unclear if NEUBAY performs well on low-quality datasets
>
> > (3) In Section 3, the authors emphasize that the Bayesian framework enables the model to perform well even with low-quality data. However, it is unclear whether the method consistently demonstrates superior performance compared to other algorithms on the D4RL MuJoCo random datasets in Table 1.
>
>
> We provide a consolidated view of NEUBAY’s performance on the 6 *low-quality* locomotion datasets (3 D4RL random and 3 NeoRL low), shown below.
> *(bold = best by statistical significance)*
>
> | Dataset      | NEUBAY (Bayesian) | MoDAP (Bayesian) | MOPO (one-step, MLE)           | ADMPO (multi-step)            | VIPO (aux loss)            |
> |-|-|-|-|-|-|
> | hc-random    | 37.0 ± 3.3        | 36.5 ± 1.8       | 38.5            | **45.4 ± 2.8**    | 42.5 ± 0.2       |
> | hp-random    | **24.5 ± 28.5**   | 8.9 ± 1.1        | **31.7**         | **32.7 ± 0.2**    | **33.4 ± 1.9**   |
> | wk-random    | **34.1 ± 6.8**    | 23.1 ± 1.6       | 7.4              | 22.2 ± 0.2        | 20.0 ± 0.1       |
> | hc-Low       | 53.1 ± 1.1        | 53.9 ± 1.1       | 40.1             | 52.8 ± 1.2        | **58.5 ± 0.1**   |
> | hp-Low       | **30.3 ± 2.9**    | **26.1 ± 4.7**   | 6.2              | 22.3 ± 0.1        | **30.7 ± 0.3**   |
> | wk-Low       | 43.9 ± 5.9        | 51.3 ± 7.8       | 11.6             | 55.9 ± 3.8        | **67.6 ± 0.7**   |
> | **AVG**          | 37.2              | 33.3             | 22.6             | 38.6              | 42.1         |
>
> We find:
> - Among **non-conservative (Bayesian)** methods, NEUBAY performs best overall: it outperforms MoDAP on average and is not statistically worse on any dataset.
> - Compared to **popular conservative** baselines, NEUBAY is consistently better than MOPO, which uses the same one-step MLE formulation.
> - **Recent strong conservative** methods (ADMPO, VIPO) achieve higher averages, but they rely on stronger world models: multi-step prediction (ADMPO) or additional value-consistency loss (VIPO). NEUBAY intentionally uses a standard one-step MLE setup to isolate the Bayesian principle and keep it comparable to MOPO.
>
> We also clarify that the paper does *not* claim NEUBAY outperforms all conservative baselines on low-quality datasets. Section 3 analyzes the Bayesian principle in a bandit setting, where learning the reward posterior is far easier. In locomotion, one primary challenge is learning an accurate MDP posterior, and the performance gap is understandable given that NEUBAY uses a simple world model setup.
>
> ### 4. Try smaller ensemble size
>
> > (4) The ensemble size was set to 100. This is similar to the perspective in model-free algorithms such as SAC-N, where a large ensemble contributes to improved performance. Therefore, it would be valuable to demonstrate whether the proposed method (NEUBAY) still performs well when the ensemble size is reduced to a comparable level (e.g., 5 or 7), as done in other algorithms. This would be similar in spirit to how EDAC was motivated by the idea of SAC-N.
>
> In the main paper (Table 2), we reported sensitivity to ensemble size $N$, and the full per-dataset results appear in Appendix Table 13. For convenience, we reproduce the summary below:
>
> | Benchmark                     | NEUBAY $N = 100$ | NEUBAY $N = 20$        | NEUBAY $N = 5$          |
> |-|-|-|-|
> | D4RL Locomotion (12 tasks)   | 80.1    | 71.6          | 65.9           |
> | NeoRL Locomotion (9 tasks)   | 64.7    | 60.3          | 59.8           |
> | D4RL Adroit (3 tasks)        | 42.2    | 30.0          | 32.6           |
> | D4RL AntMaze (4 tasks)       | 43.2    | 33.2          | 28.8           |
>
> We observe a *moderate* degradation when reducing from $100 \to 20 \to 5$, which is expected because a larger ensemble size yield a more faithful posterior and uncertainty estimate, both central to our method. Nevertheless, NEUBAY with $N=5$ still performs competitively on the NeoRL benchmark.
>
> To contextualize this, we compare with MAPLE [Chen et al., 2021], a Bayesian-inspired method that also trains history-depependent agent and reports ensemble-size sensitivity ([their TABLE IV](https://ieeexplore.ieee.org/document/10255284)). Even with $N=5$, NEUBAY is better than MAPLE with $N=142$, suggesting that NEUBAY makes *more efficient* use of the ensemble.
>
> | Benchmark                     | NEUBAY $N = 100$ | NEUBAY $N = 20$        | NEUBAY $N = 5$          | MAPLE $N=142$ | MAPLE $N=14$ |
> |-|-|-|-|-|-|
> | D4RL Locomotion (12 tasks)   | 80.1    | 71.6          | 65.9           | 61.3 | 50.1 |
>
> *Note: in the MAPLE paper, "MAPLE-20" and "MAPLE-200" correspond to $N=14$ and $N=142$ in our notation, respectively, because MAPLE uses roughly $5/7$ of ensemble members for planning.*
>
> Finally, the computational cost of large $N$ is modest in our JAX implementation; we provide details in our response to the next question.

---

> > ### Author Response · Authors · 2025-11-23
> > **Response (part 3)**
> >
> > ### 5. Computation costs
> >
> > > (5) It is also important to include a comparison of running time with respect to the ensemble size, as increasing the ensemble size can significantly affect computational efficiency.
> >
> > We provide full runtime and memory details in the **Common response**. In short, even with a large ensemble of $N=100$, NEUBAY remains efficient: the recurrent agent trains **faster than MOPO’s PyTorch implementation** (4.4 hrs vs. 5.6 hrs for 2M steps), and the rollout function scales very well due to JAX vectorization. Increasing $N$ from 5 to 100 raises rollout time by only 0.6 seconds, making the overhead negligible.
> >
> > Thus, in our implementation, large ensemble sizes do *not* significantly increase computational cost.
> >
> > ---
> >
> > Thank you again for your time and constructive feedback. We hope our responses and revisions address your concerns, and we would be happy to clarify anything further if needed.
> >
> > ### References
> > - Richard S Sutton and Andrew G Barto. Reinforcement learning: An introduction. MIT press, 2018.
> > - Xiong-Hui Chen, Yang Yu, Qingyang Li, Fan-Ming Luo, Zhiwei Qin, Wenjie Shang, and Jieping Ye. Offline model-based adaptable policy learning. NeurIPS 2021.

---

> > > ### Comment · Reviewer_Ckqf · 2025-11-27
> > > **Re: Rebuttal**
> > >
> > > Thank you for the thoughtful and detailed responses to my questions. Nevertheless, there are several points on which I remain unconvinced.
> > >
> > > - **Compounding error:**
> > > The additional results suggest that LayerNorm plays an important role in stabilizing the world model. However, its effect appears inconsistent across datasets. While the halfcheetah-medium-expert-v2 dataset shows a sharp increase in error without LayerNorm, Figure 6 indicates that, for halfcheetah-random, the difference between models with and without LayerNorm is negligible. On the medium dataset, both variants eventually exhibit an increase in error, and although the magnitudes differ, using a model with an RMSE above 10 still seems impractical.
> > >
> > > - Additionally, the Diffusion Model Predictive Control (D-MPC [3]) paper (Figure 2) reports long-horizon prediction accuracy using RMSD, where even an MLP-based model—trained on medium data and evaluated on medium, medium-replay, and expert datasets—maintains RMSD around 0.1 at horizon length 256. In contrast, the RMSD (10) of the NEUBAY world model appears significantly higher. Moreover, baseline methods (MOPO, COMBO, etc.) compared in this paper also use MLP-based models, and the Bayesian-inspired MAPLE method likewise trains its model using an MLP. Given this, I remain unsure whether the proposed world model architecture constitutes a clear main contribution, especially when compared to simpler MLP-based dynamics models.
> > >
> > > - **Computation costs:**
> > > The comparison of runtime between MOPO’s PyTorch implementation and NEUBAY’s JAX implementation seems unfair. Numerous papers and open-source repositories (JAX-CORL [1] and Unifloral [2]) have already documented that identical architectures typically run $5 \sim 20 \times $ faster in JAX than in PyTorch. Furthermore, the computation cost should reasonably depend on the ensemble size, yet the paper does not explicitly discuss how the ensemble size influences model training time.
> > >
> > > - **Performance**:
> > > Beyond the D4RL MuJoCo locomotion tasks, NEUBAY does not appear particularly strong on NeoRL locomotion, Adroit, and AntMaze benchmarks (Tables 9–11). Of course, the reviewer does not expect NEUBAY to dominate all baselines across all domains. However, many of the compared methods already incorporate some form of conservatism and often use relatively small ensemble sizes (5~7). In contrast, NEUBAY addresses conservatism via a Bayesian approach and implicitly argues that larger model ensembles are beneficial. From the reviewer’s perspective, this naturally raises a performance–cost trade-off: the gains obtained by NEUBAY should justify the additional computational overhead of large ensembles. A comparison primarily against MAPLE in the authors' responses does not fully support this trade-off.
> > >
> > >
> > >
> > > [1] https://github.com/nissymori/JAX-CORL
> > >
> > > [2] https://github.com/EmptyJackson/unifloral
> > >
> > > [3] Zhou, Guangyao, et al. "Diffusion model predictive control." arXiv preprint arXiv:2410.05364 (2024).

---

> > > > ### Author Response · Authors · 2025-12-02
> > > > **Response to remaining concerns (part 1)**
> > > >
> > > > Thank you for your insightful feedback. We are glad that the novelty concern has been resolved. Below we address your remaining points. We have also updated the PDF, with new content highlighted in **red**.
> > > >
> > > > ## 1. Compounding error
> > > >
> > > > > The additional results suggest that LayerNorm plays an important role in stabilizing the world model. However, its effect appears inconsistent across datasets. While the halfcheetah-medium-expert-v2 dataset shows a sharp increase in error without LayerNorm, Figure 6 indicates that, for halfcheetah-random, the difference between models with and without LayerNorm is negligible. On the medium dataset, both variants eventually exhibit an increase in error, and although the magnitudes differ, using a model with an RMSE above 10 still seems impractical.
> > > > >
> > > > >  Diffusion Model Predictive Control (D-MPC [3]) paper (Figure 2) reports long-horizon prediction accuracy using RMSD, where even an MLP-based model—trained on medium data and evaluated on medium, medium-replay, and expert datasets—maintains RMSD around 0.1 at horizon length 256. In contrast, the RMSD (10) of the NEUBAY world model appears significantly higher... Given this, I remain unsure whether the proposed world model architecture constitutes a clear main contribution, especially when compared to simpler MLP-based dynamics models.
> > > >
> > > > We summarize your concerns into three points:
> > > > 1. On halfcheetah-random, the difference between models with and without LayerNorm is negligible.
> > > > 2. On halfcheetah-medium, an RMSE above 10 seems impractical.
> > > > 3. Our MLP baseline seems much worse than the MLP baseline in the D-MPC paper.
> > > >
> > > > ### Response to (1): On halfcheetah-random, LayerNorm matters under higher-quality evaluation policies, not purely random policies.
> > > >
> > > > We agree that under a **random evaluation policy**, the effect of LayerNorm on halfcheetah-random is limited, likely because random actions are well-covered in the random dataset. However, a random policy represents only one corner of the state-action distribution and does not reflect the trajectories encountered as the agent improves. To capture a broader and more realistic range of behaviors (similar to the D-MPC evaluation protocol), we now evaluate compounding errors across **three datasets** that share the same underlying MDP:
> > > > - $\star$-random-v2 (random policy)
> > > > - $\star$-medium-replay-v2 (higher-quality policy)
> > > > - $\star$-medium-expert-v2 (higher-quality policy)
> > > >
> > > > These evaluation datasets span a broad range of exploratory behaviors. We evaluate world ensembles trained on 6 datasets using 3 evaluation datasets, yielding **18 total setups**.
> > > >
> > > > This significantly strengthens our analysis: the new results **(Appendix Sec. H.2; Figs. 6–8)** show that LayerNorm provides consistent and substantial reductions in worst-case errors in 16 out of 18 setups. Especially, for models trained on halfcheetah-random, LayerNorm provides a clear benefit when evaluated on medium-replay and medium-expert action sequences, where compounding errors would otherwise grow rapidly.
> > > >
> > > > ### Response to (2): High-error regions are low-density, and Bayesian RL avoids overcommitting to localized model errors.
> > > >
> > > > We agree that in some state–action regions the RMSE can exceed 10. However, these regions constitute a **very small proportion** of the data distribution, as illustrated by the scatter plots. More importantly, Bayesian RL optimizes the *expected* return under a distribution of possible MDPs. As a result, even if some ensemble members exhibit localized high errors, their influence is **diluted** when aggregated over the full ensemble, reducing their impact on policy improvement.
> > > >
> > > > In practice, NEUBAY performs well on halfcheetah-medium, indicating that these localized high-error regions do not harm policy improvement.

---

> > > > > ### Author Response · Authors · 2025-12-02
> > > > > **Response to remaining concerns (part 2)**
> > > > >
> > > > > ### Response to (3): Our MLP baseline is reasonable given mixed baseline performance in prior work.
> > > > >
> > > > > Thank you for highlighting the strong baseline reported in the D-MPC paper. While their baseline achieves lower errors than ours, a broader survey reveals that **baseline performance varies widely** across recent model-based RL papers, even among those using one-step MLP dynamics models. We summarize this below:
> > > > >
> > > > > | Prior work (Figure) | Prior baseline result | Our baseline result | Comparison|
> > > > > |-|-|-|-|
> > > > > |[MOREC: Figure 6 (without transition filtering)](https://arxiv.org/pdf/2310.05422) | MAE of $10^5$ on halfcheetah-{random, medium}, $10^3$ on medium-replay, $10^6$ on medium-expert (100 steps) | RMSE of $10^0$, $10^1$, $10^0$, $\gg 10^6$ | Better in 3/4 cases |
> > > > > |[PGD: Figure 6 (MOPO)](https://arxiv.org/pdf/2404.06356) | MSE of $8.0$ (trained on halfcheetah-medium, evaluated on random, 16 steps) | RMSE of $10^0$ | Similar |
> > > > > |[ADMPO: Figure 2 (Ensemble Dynamics Model)](https://arxiv.org/pdf/2405.17031) | Error of $10^6$ on hopper-{medium, medium-replay} (50 steps) | RMSE of $10^0$, $10^{-1}$ | Better |
> > > > > |[D-MPC: Figure 2 (MLP)](https://arxiv.org/pdf/2410.05364)| RMSE around $10^{-1}$ on halfcheetah-medium (256 steps) | RMSE $> 10^1$ | Worse|
> > > > >
> > > > > Overall, prior baselines exhibit **mixed** performance. Our baseline is:
> > > > > - **stronger** than those in MOREC and ADMPO,
> > > > > - **comparable** to PGD,
> > > > > - and **weaker** than D-MPC.
> > > > >
> > > > > Although all methods use one-step MLP dynamics models, differences in **architecture** (e.g., 2-layer MLP in D-MPC vs. 4-layer MLP in MOPO-style models), initialization, and rollout evaluation protocols likely contribute to the discrepancies. These variations highlight the lack of standardized baselines in model-based RL.
> > > > >
> > > > > Therefore, the improvements achieved by adding LayerNorm should be interpreted as **meaningful architectural gains**, not artifacts of an unusually weak baseline. To support reproducibility and fair comparison, we release our **model checkpoints and evaluation code**.
> > > > >
> > > > > References:
> > > > > - MOREC: Fan-Ming Luo, Tian Xu, Xingchen Cao, and Yang Yu. Reward-consistent dynamics models are strongly generalizable for offline reinforcement learning. ICLR 2024.
> > > > > - PGD: Matthew Thomas Jackson, Michael Tryfan Matthews, Cong Lu, Benjamin Ellis, Shimon Whiteson, and Jakob Foerster. Policy-guided diffusion. RLC 2024.
> > > > > - ADMPO: Haoxin Lin, Yu-Yan Xu, Yihao Sun, Zhilong Zhang, Yi-Chen Li, Chengxing Jia, Junyin Ye, Jiaji Zhang, and Yang Yu. Any-step dynamics model improves future predictions for online and offline reinforcement learning. ICLR 2025.
> > > > > - D-MPC: Guangyao Zhou, Sivaramakrishnan Swaminathan, Rajkumar Vasudeva Raju, J Swaroop Guntupalli, Wolfgang Lehrach, Joseph Ortiz, Antoine Dedieu, Miguel Lázaro-Gredilla, and Kevin Murphy. Diffusion model predictive control. TMLR 2025.
> > > > >
> > > > > ## 2. Computation costs
> > > > >
> > > > > ### Agent training time
> > > > >
> > > > > > The comparison of runtime between MOPO’s PyTorch implementation and NEUBAY’s JAX implementation seems unfair. Numerous papers and open-source repositories (JAX-CORL [1] and Unifloral [2]) have already documented that identical architectures typically run 5-20x faster in JAX than in PyTorch.
> > > > >
> > > > > **Why do we report MOPO PyTorch training time?** Our use of MOPO’s PyTorch runtime is *not* intended as a framework-level comparison. It simply serves as a **baseline reference** to contextualize NEUBAY’s absolute training cost. Our goal is to show that the **4.4 hours** required by NEUBAY is a reasonable training time for model-based RL, not to claim a speed advantage over PyTorch.
> > > > >
> > > > > **Comparison with Unifloral’s JAX implementation.** We agree that Unifloral reports a faster JAX implementation of MOPO (28 minutes). We now **clarify this explicitly** in the paper. The primary reason is Unifloral’s use of `jax.lax.scan` to JIT-compile MOPO’s gradient update loop ([see their code](https://github.com/EmptyJackson/unifloral/blob/main/algorithms/mopo.py#L404)), a technique known to yield ~5x speedups (e.g., shown in the [SAC-N JAX codebase](https://github.com/Howuhh/sac-n-jax?tab=readme-ov-file#speed-comparison)).
> > > > >
> > > > > NEUBAY does not use this optimization because it is not directly compatible with our recurrent RL update, although it can be incorporated with additional engineering effort. Applying the same optimization to the Markov version of NEUBAY would reduce training time from 2.6 hours to ~0.5 hours, comparable to Unifloral.
> > > > >
> > > > > **Takeaway.** While further optimization (e.g., via `jax.lax.scan`) is possible, our current training time (4.4 hrs) is already **within a practical and expected range** for model-based RL and does not pose a bottleneck for our experiments.

---

> ### Author Response · Authors · 2025-12-02
> **Response to remaining concerns (part 3)**
>
> ### Model training time
>
> > Furthermore, the computation cost should reasonably depend on the ensemble size, yet the paper does not explicitly discuss how the ensemble size influences model training time.
>
> As requested, we report the model training time per seed on halfcheetah-medium-expert-v2 for different total ensemble sizes $N_{\text{total}}$:
>
> | Total ensemble size $N_{\text{total}}$ | 8 (1x) | 32 (4x) | 128 (16x) |
> |-|-|-|-|
> |Training time | 1.2 hrs (1x) | 1.7 hrs (1.42x) | 6.0 hrs (5x) |
>
> The training time is **roughly sublinear** to $N_{\text{total}}$. Throughout our experiments, we train a single ensemble per seed with $N_{\text{total}} = 128$ and then select the top $N = 100$ models for main results, and $N = 5, 20$ for the sensitivity study.
>
> ## 3. Performance-cost trade-off
>
> > ... However, many of the compared methods already incorporate some form of conservatism and often use relatively small ensemble sizes (5~7). In contrast, NEUBAY **addresses conservatism via a Bayesian approach** and implicitly argues that larger model ensembles are beneficial. From the reviewer’s perspective, this naturally raises a **performance–cost trade-off**: the gains obtained by NEUBAY should justify the additional computational overhead of large ensembles.
>
> We address the reviewer’s concern in three parts:
>
> **(1) Conceptual goal: NEUBAY is not another conservative method.** Our goal is *not* to introduce another form of conservatism with a larger ensemble, but rather to show that a **non-conservative** Bayesian method can be made practical, which has not been demonstrated in prior work. While most existing methods build on conservatism, NEUBAY explicitly *abandons* conservatism. As our bandit example illustrates, non-conservative methods have the potential to improve generalization, which is an important and underexplored direction in offline RL.
>
> **(2) Conservatism does not help on average when ensemble size is controlled.** With the same ensemble size ($N=100$), adding conservative regularization to NEUBAY **shows degradation** on average (Table 2). This indicates that NEUBAY’s gains do not arise solely from using a large ensemble, but also from its non-conservative Bayesian design.
>
>
> **(3) The performance–cost trade-off is modest in practice.** The additional cost of larger ensembles scales **sublinearly** with ensemble size (see the previous question), due to parallelization. Moreover, world models are trained **once** per seed, and the resulting checkpoint is reused across all downstream agent-tuning experiments. Thus, the practical computational burden is limited relative to overall training time in RL.
>
> **Summary.** We view NEUBAY as an initial step toward practical non-conservative Bayesian RL. While future work may reduce ensemble size via parameter sharing, our results already show that non-conservative approaches can perform competitively across diverse benchmarks, making them a meaningful alternative to existing conservative methods.

---

### Official Review · Reviewer_3tJo · 2025-11-02

**Soundness:** 3
**Presentation:** 4
**Contribution:** 2
**Rating:** 4
**Confidence:** 3

**Summary:**

This paper proposes NEUBAY, a model-based offline RL algorithm that challenges the dominant principle of conservatism. Instead of using explicit uncertainty penalties, NEUBAY adopts a Bayesian perspective, modeling a posterior over world models and training a history-dependent agent. Its key mechanism is an adaptive planning horizon, where rollouts are truncated based on epistemic uncertainty, allowing for potentially very long planning horizons.

**Strengths:**

[S1] The key idea of replacing explicit conservative penalties with planning horizon truncation based on epistemic uncertainty is novel and well-motivated. This opens a new line for integrating uncertainty into model-based offline RL.

**Weaknesses:**

[W1] There is a significant disconnect between the method's justification (enabling long-horizon planning) and its empirical results. The algorithm performs poorly on tasks that genuinely require long-horizon planning and sparse reward handling (e.g., AntMaze large, 0% normalized score).

Conversely, NEUBAY's successes are on benchmarks (e.g., D4RL MuJoCo) that are largely reactive, fully observable and where strong performance is achievable with much shorter horizons (1~10 steps) or even model-free, fully observed MDP methods. While the paper shows NEUBAY uses very long horizons (e.g., 512-1000 steps) on these tasks, this success does not sufficiently demonstrate why such long-horizon planning is necessary, given that simpler methods also perform well.

[W2] The proposed algorithm has high architectural complexity, integrating deep ensembles ($N=100$), LRU-based recurrent encoders,  specific context-handling mechanisms and task-dependent hyperparameters (e.g., real data ratio, encoder learning rate, gradient clipping). The complexity limits the method's reproducibility and practical applications.

**Questions:**

[Q1] Following W1, are there any tasks that can demonstrate long-horizon planning (e.g., 512-1000 steps)? Ideally, such tasks should be highlighted in the paper.

[Q2] Following W2, can NEUBAY be simplified, emphasizing the adaptive horizon while minimizing its complexity?

---

> ### Author Response · Authors · 2025-11-23
> **Response (part 1)**
>
> Thank you for your detailed review and constructive feedback. We respond to your concerns below, and we kindly invite you to also consult the updated manuscript and the common response.
>
> > [W1] There is a significant disconnect between the method's justification (enabling long-horizon planning) and its empirical results. The algorithm performs poorly on tasks that genuinely require long-horizon planning and sparse reward handling (e.g., AntMaze large, 0% normalized score).
> > Conversely, NEUBAY's successes are on benchmarks (e.g., D4RL MuJoCo) that are largely reactive, fully observable and where strong performance is achievable with much shorter horizons (1~10 steps) or even model-free, fully observed MDP methods.
> > While the paper shows NEUBAY uses very long horizons (e.g., 512-1000 steps) on these tasks, this success does not sufficiently demonstrate why such long-horizon planning is necessary, given that simpler methods also perform well.
> >
> > [Q1] Following W1, are there any tasks that can demonstrate long-horizon planning (e.g., 512-1000 steps)? Ideally, such tasks should be highlighted in the paper.
>
> We believe the concern reflects two different roles of "long-horizon planning", and we summarize your concern as two questions:
> 1. *If dense-reward tasks (e.g., D4RL locomotion) do not require long-horizon planning, why does NEUBAY still rely on long horizons?*
> 2. *If sparse-reward tasks (e.g., AntMaze) benefit from long-horizon planning, why does NEUBAY fail on some of them?*
>
> Below, we address each question in turn.
>
> ### 1. Why does NEUBAY need long-horizon planning even on dense-reward tasks?
>
> **Short answer:**
>
> **NEUBAY is a Bayesian, non-conservative offline RL algorithm. Without conservatism, long-horizon rollouts are required to suppress value overestimation.** This role is distinct from classical long-horizon planning for exploration.
>
> We clarify this in three complementary ways:
>
> **(1) Intuition**
>
> It is true that prior offline RL methods perform well on dense-reward locomotion tasks using short horizons or even purely model-free RL. However, these methods rely heavily on conservative mechanisms, such as uncertainty penalties and value regularization, to control value overestimation. Our bandit example (Sec. 3) shows that such conservatism limits generalization, motivating our Bayesian non-conservative approach.
>
> In this non-conservative regime, we find that **long-horizon rollouts take on a new role**: they **suppress value overestimation** by attenuating high-bias bootstrapped terms. This intuition is captured in **Eq. 6 of Sec. 4.2**,  reproduced below,
> $$Q^{\text{Bayes}}(h_t,\hat a_t) \gets \sum\nolimits_{j=0}^{H-1} \gamma^{j} \underbrace{\hat r_{t+j+1}}\_{\text{lower bias}} + \underbrace{\gamma^{H}}\_{\text{discount}} \underbrace{Q^{\text{Bayes}}(\hat h_{t+H}, \pi(\hat h_{t+H}))}\_{\text{higher bias}},$$
> where the bootstrapped term, responsible for most overestimation, is exponentially discounted with the horizon length $H$. This decomposition does not rely on the task structure. Thus, even on dense-reward tasks that do *not* require long-horizon planning for exploration, NEUBAY requires long horizons to avoid severe overestimation, precisely because it does *not* use conservatism.
>
> **(2) Formal justification**
>
> In the rebuttal, we add a formal result (Appendix Sec. C.2) showing that the value error decomposes into:
> - a discounted sum of intermediate TD errors, and
> - a **bootstrapped error** term discounted by $\gamma^H$.
>
> This builds on [Sims et al., 2024], who analyze a similar phenomenon in a conservative setting: when uncertainty penalties vanish, value overestimation resurfaces through the same bootstrapping mechanism. To address this, they introduce an SAC-N-style pessimism [An et al., 2021], thereby continuing to rely on short horizons and conservatism.
>
> In contrast, NEUBAY directly leverages the $\gamma^H$-suppression effect, allowing long-horizon rollouts without conservatism. We clarify this relationship in the main body and Appendix Sec. A.3.
>
> **(3) Empirical evidence (also answers your Q1)**
>
> NEUBAY extensively uses long horizons in practice, and this is directly visible in our experiments (see Fig. 1 and Appendix H.3). Under the best setting ($\zeta=1.0$), the adaptive horizon mechanism yields:
> - 75th-percentile horizon of **64-512 steps**, and
> - maximum horizon of **256-1000 steps**,
>
> across **21 of 23** tasks with episode length $T=1000$.
>
> By examining the $Q$-value plots, we find that these long rollouts indeed suppress overestimation of $Q(h_t,a_t)$ on real data compared to short rollouts, as described in Sec 5.2. These results confirm that NEUBAY requires long-horizon planning in most tasks for controlling overestimation.

---

> ### Author Response · Authors · 2025-11-23
> **Response (part 2)**
>
> ### 2. Why does NEUBAY fail on large AntMaze tasks which favors long-horizon planning?
>
> We agree that large AntMaze is a challenging sparse-reward domain where planning can be beneficial. In the rebuttal, we add a **failure-case analysis** in Appendix Sec. H.5. The visualizations show that NEUBAY agent often remains near the initial corner or repeatedly collides with nearby walls, resulting in degenerate behaviors.
>
> We also add an ablation study on *conservative* variants of NEUBAY by adding uncertainty penalties in AntMaze. However, these consistently *degrade performance* (see Table 13), indicating that conservatism does not resolve the underlying issue.
>
> Based on these observations, we attribute the failures to two factors:
> - Lack of effective exploration: NEUBAY uses direct policy rollouts without advanced planning algorithms.
> - Long-horizon modeling challenges: The contact-rich dynamics and wall collisions are difficult for single-step world models to predict over long horizons.
>
> We expect NEUBAY to benefit from stronger planning and modeling components, such as beam-search planning [Janner et al., 2021], more stable exploration strategy [Park & Lee, 2025], and multi-step world models [Lin et al., 2025].
>
> ### 3. Complexity and practicality
>
> > [W2] The proposed algorithm has high architectural complexity, integrating deep ensembles ($N=100$), LRU-based recurrent encoders, specific context-handling mechanisms and task-dependent hyperparameters (e.g., real data ratio, encoder learning rate, gradient clipping). The complexity limits the method's reproducibility and practical applications.
>
> We address this concern in three parts: (1) reproducibility and practicality, (2) complexity, and (3) task-dependent hyperparameters.
>
> **(1) Reproducibility and practicality.** To make NEUBAY easy to use in practice, in the rebuttal, we have:
> - **open-sourced** the full codebase,
> - provided **pretrained ensemble checkpoints** (linked in the readme), so that users can train agents directly on them, and
> - added clear **tuning guidelines** in Section 6 of the updated PDF.
>
> With these resources, NEUBAY is simple to reproduce and tune in practice.
>
> **(2) Complexity analysis.** We break down each component of NEUBAY and the overall complexity is modest with our JAX implementation.
>
> | Component | Time complexity | Space complexity | Architectural complexity | Tuning complexity |
> | - | -|- | -| -|
> |Ensemble size $N$ | One-time training cost; inference cost is $O(1)$ via `jax.vmap` | $O(N)$ (see "Computation costs") | Simple one-step MLP | None (larger is better) |
> | LRU-based RNN encoder| For a $L$-length sequence, LRU has $O(\log L)$  training cost via `jax.lax.associative_scan` while vanilla RNN has $O(L)$ | See "Computation costs" | LRU's hidden state transitions are linear while vanilla RNN's transitions are nonlinear | None |
> | Uncertainty-based truncation | Less than 10 secs by precomputing the threshold | A float scalar (the threshold) | N/A | None (we fix $\zeta = 1.0$) |
>
> **(3) NEUBAY uses a similar number of task-dependent hyperparameters as prior work.** NEUBAY requires sweeping only **two** hyperparameters per dataset: real data ratio $\kappa$ and encoder learning rate $\eta_\phi$. Gradient clipping is set by a simple heuristic based on return scale (e.g., AntMaze $\approx$ 1 while locomotion $\approx$ 1000) and is *not* tuned.
>
> Compared to popular offline model-based RL methods:
>
> | Method | Task-dependent hyperparams | Count |
> | - | -| -|
> | NEUBAY (ours) | real data ratio, encoder learning rate | 2 |
> | MOPO [Yu et al., 2020] | rollout horizon, penalty coefficient | 2|
> | COMBO [Yu et al., 2021] | rollout horizon, learning rate, conservative coefficient, $\rho(s,a)$, $\mu (a\mid s)$, real data ratio | 6|
> | RAMBO [Rigter et al., 2022] | rollout horizon, adversarial loss coefficient | 2|
> | MOBILE [Sun et al., 2023] | rollout horizon, penalty coefficient | 2|
>
> Thus, NEUBAY’s hyperparameter budget is on par with popular offline model-based RL methods.
>
> ### 4. Can we simplify NEUBAY?
>
> > [Q2] Following W2, can NEUBAY be simplified, emphasizing the adaptive horizon while minimizing its complexity?
>
>
> Yes, NEUBAY can be simplified further and we outline two concrete directions:
>
> - Simplifying the recurrent encoder. Our new ablation study shows that replacing the LRU agent with a Markov MLP agent, performs similarly on most locomotion tasks (Table 2 and Table 13). The main degradation appears in AntMaze due to its high epistemic uncertainty about maze layout. This suggests that for many simple domains, **a Markov version of NEUBAY** is already a viable simplification.
> - Simplifying the ensemble. The world ensemble can also be simplified using parameter-sharing techniques from Bayesian deep learning, such as BatchEnsemble [Wen et al., 2020], which significantly reduces memory and time costs while preserving uncertainty estimates.

---

> > ### Author Response · Authors · 2025-11-23
> > **References**
> >
> > ---
> >
> > Thank you again for your time and constructive feedback. We hope our responses and revisions address your concerns, and we would be happy to clarify anything further if needed.
> >
> > ### References
> >
> > - Anya Sims, Cong Lu, Jakob N Foerster, and Yee W Teh. The edge-of-reach problem in offline model-based reinforcement learning. NeurIPS 2024.
> > - Gaon An, Seungyong Moon, Jang-Hyun Kim, and Hyun Oh Song. Uncertainty-based offline reinforcement learning with diversified q-ensemble. NeurIPS 2021.
> > - Haoxin Lin, Yu-Yan Xu, Yihao Sun, Zhilong Zhang, Yi-Chen Li, Chengxing Jia, Junyin Ye, Jiaji Zhang, and Yang Yu. Any-step dynamics model improves future predictions for online and offline reinforcement learning. ICLR 2025.
> > - Kwanyoung Park and Youngwoon Lee. Model-based offline reinforcement learning with lower expectile q-learning. ICLR 2025.
> > - Michael Janner, Qiyang Li, and Sergey Levine. Offline reinforcement learning as one big sequence modeling problem. NeurIPS 2021.
> > - Tianhe Yu, Garrett Thomas, Lantao Yu, Stefano Ermon, James Y Zou, Sergey Levine, Chelsea Finn, and Tengyu Ma. Mopo: Model-based offline policy optimization. NeurIPS 2020.
> > - Tianhe Yu, Aviral Kumar, Rafael Rafailov, Aravind Rajeswaran, Sergey Levine, and Chelsea Finn. Combo: Conservative offline model-based policy optimization. NeurIPS 2021.
> > - Marc Rigter, Bruno Lacerda, and Nick Hawes. Rambo-rl: Robust adversarial model-based offline reinforcement learning. NeurIPS 2022.
> > - Yihao Sun, Jiaji Zhang, Chengxing Jia, Haoxin Lin, Junyin Ye, and Yang Yu. Model-bellman inconsistency for model-based offline reinforcement learning. ICML 2023.
> > - Yeming Wen, Dustin Tran, and Jimmy Ba. BatchEnsemble: an alternative approach to efficient ensemble and lifelong learning. ICLR 2020.

---

### Author Response · Authors · 2025-11-23
**Common response**

We sincerely thank all reviewers for their constructive and insightful feedback, which has significantly improved the quality and clarity of our paper. Below we provide responses to the common concerns.

## Code and checkpoints released

During the rebuttal period, we open-sourced our full JAX-based code along with pretrained world-model ensemble checkpoints to ensure reproducibility. Code and checkpoint link are included in the **supplementary material**.


## Computation costs

We provide a detailed computation analysis in the updated PDF (**Sec. 5.3** and Appendix G.3). Below, we summarize the key findings relevant to common concerns.

### 1. Training cost

The total cost consists of **(1) world-model training (one-time cost)** and **(2) agent training (dominant cost)**. As in standard two-phase pipelines for offline MBRL, the world model is pretrained once and its ensemble checkpoint is reused across agent-training runs; thus tuning cost is dominated by agent training.

All experiments were run on a single NVIDIA L40S (48 GB). For *halfcheetah-medium-expert-v2* (2M gradient steps) with **an ensemble size of $N=100$**, the per-seed cost (averaged over 3 seeds run in parallel) is:

| Training component | Time (per seed) | GPU memory (per seed) |
|----------|-----------------|-------------|
| **World model (one-time cost)** | 6.0 hrs | 10.7 GB |
| **Recurrent agent (main cost)** | **4.4 hrs** | **2.6 GB** (1.5GB world model; 1.1 GB agent) |
| **Markov agent** (ablation study) | 2.6 hrs | 1.5 GB |


Importantly, NEUBAY’s agent-training cost (**4.4 hrs**) is already **faster** than the widely-used **MOPO PyTorch** implementation [Sun, 2023] as benchmarked by [Jackson et al., 2025], which requires **5.6 hrs** for the same 2M gradient steps. Thus, even with a large ensemble, NEUBAY is not slower than popular MBRL implementations, and the GPU memory footprint during agent training is modest.

References:
- Yihao Sun. Offlinerl-Kit: An elegant pytorch offline reinforcement learning library. https://github.com/yihaosun1124/OfflineRL-Kit, 2023.
- Matthew Thomas Jackson, Uljad Berdica, Jarek Liesen, Shimon Whiteson, and Jakob Nicolaus Foerster. A clean slate for offline reinforcement learning. NeurIPS 2025.

### 2. Rollout inference remains efficient as ensemble size increases

In our JAX implementation, the rollout function are **fully vectorized** across the ensemble and rollouts using `jax.vmap`. As a result, increasing either the ensemble size $N$ or the number of parallel rollouts $K$ has only a minor impact on runtime. We use $N=K=100$ in practice.

Runtime for different $N$ and $K$:
|  | $K=5$ | $K=20$ | $K=100$ |
|-|-|-|-|
| $N=5$           | 2.7s  | 3.0s   | 4.7s    |
| $N=20$         | —     | 3.6s   | 5.0s    |
| $N=100$          | —     | —      | 5.3s    |

Given $K=100$, increasing $N$ from 5 to 100 increases rollout time by only **0.6 seconds**. Therefore, rollout inference is negligible in the total training budget, even with large ensembles.

## Summary of updates in the revised PDF

We highlight the changes in the revised PDF **in red**. Below is a summary:

- **Sec. 1  (3tJo)**: explain why we need long-horizon planning
- **Sec. 4.2 (3tJo)**: clarify why we need long-horizon planning with a proof (Appendix C.2) and connect with [Sims et al., 2024] (Appendix A.3)
- **Sec. 5.2 Figure 5 and Appendix Figure 6-8 (Ckqf)**: re-plot the compounding errors for clarity
- **Sec. 5.2 Table 2 and Appendix Table 13 (ctso)**: add ablation studies on Markov agents in NEUBAY
- **Sec. 5.2 Table 2 and Appendix Table 13 (3tJo)**: add ablation studies in AntMaze
- **Sec. 5.3 and Appendix G.3 (all)**: add a detailed analysis on computation costs
- **Sec. 6 (all)**: add tuning guidelines
- **Reproducibility statement (all)**
- **Appendix A.1 (Ckqf)**: add related work on decision-time planning
- **Appendix A.3 (Ckqf)**: add related work on adaptive-horizon mechanism
- **Appendix H.2 (Ckqf)**: evaluate compounding errors under different datasets and add discussion on MLP baselines
- **Appendix H.3 (3tJo)**: add discussion on the scale of horizons
- **Appendix H.5 (3tJo)**: add analysis on failure cases in AntMaze

---

### Author Response · Authors · 2025-12-02
**Summary to AC**

We thank the reviewers for their thoughtful feedback. Below we summarize the contributions and how we addressed the key concerns.

## Contribution Summary

Our main contribution is to show that a **non-conservative Bayesian** approach is a viable paradigm for offline RL -- an important direction that is underexplored but highlighted as “very meaningful” by reviewer ctso. Prior offline RL methods rely on conservatism (pessimism, uncertainty penalties), but we demonstrate conceptually and empirically (bandit example) that conservatism can harm generalization, while Bayesian non-conservatism can improve it. Reviewer ctso noted that this example is clear and sound.


To make non-conservative Bayesian RL practical, we introduce three key components that yield the NEUBAY algorithm:
1. Layer normalization in world models to control compounding error,
2. **Adaptive long-horizon planning to reduce overestimation** while preserving non-conservatism (described as “novel and well-motivated” by 3tJo),
3. Stabilized recurrent RL training.

NEUBAY achieves **competitive performance** across a wide range of recent offline RL methods (noted by ctso and Ckqf), and extensive ablations (ensemble size, conservatism, planning horizon) validate our design choices (ctso, Ckqf). Overall, NEUBAY provides the **first practical demonstration** that non-conservative Bayesian RL can be competitive and conceptually distinct from conservative approaches.

## Rebuttal Summary

The **key remaining concerns** raised by reviewers were regarding (1) the role of long-horizon planning, (2) the significance of LayerNorm effect, (3) the strength of the MLP model baseline, and (4) computational costs.

We addressed these by:

1. providing formal justification and empirical evidence showing that long-horizon planning reduces overestimation in the absence of conservatism,
2. adding extensive new compounding-error results (*16/18* setups) demonstrating the importance of LayerNorm,
3. surveying prior literature to show that our MLP baseline is *reasonable* given mixed baseline results reported in recent work,
4. clarifying that ensemble training cost has *sublinear time complexity and is one-time*, and that NEUBAY’s advantage stems from its non-conservative Bayesian design, not from large ensembles *alone*.

In the common response, we improved reproducibility through open-sourcing code and checkpoints, and clarified computational details.

Reviewer ctso retained their initial score of 8 and stated that all concerns were resolved. The remaining reviewers’ questions were addressed with additional analysis and clarification.

---

### Meta-Review · Area_Chair_eLMD · 2026-01-09

**Summary:**

The reviewers' concerns primarily focused on the justification and empirical validation of the proposed long-horizon planning mechanism, specifically noting a disconnect where the method succeeds on reactive locomotion tasks but fails on sparse-reward domains like AntMaze that genuinely require long-horizon reasoning. Additionally, significant scrutiny was placed on the architectural complexity and computational cost of using large ensembles (N=100), with reviewers questioning whether the performance gains justified the overhead compared to simpler baselines or existing methods like MBOP and MOPP. Finally, there were technical disputes regarding the validity of the world model evaluations, specifically the impact of LayerNorm on compounding errors and the strength of the MLP baseline compared to state-of-the-art results in literature such as D-MPC.

**Reviewer Concerns:**

The rebuttal successfully addressed the concerns regarding the novelty of the method relative to MBOP and MOPP by clarifying the fundamental differences between decision-time and background planning, and effectively resolved reproducibility and complexity apprehensions by open-sourcing the code, providing checkpoints, and adding detailed tuning guidelines. However, the concern regarding the method's poor performance on AntMaze remains outstanding, as the authors acknowledged this as a limitation due to exploration and modeling challenges rather than resolving it, and Reviewer Ckqf remained unconvinced regarding the strength of the chosen MLP baseline and the fairness of the JAX-based runtime comparisons against PyTorch implementations.

**Reviewer Scores:**

Reviewer ctso explicitly stated during the discussion that they would maintain their score of 8, as they felt all their concerns regarding complexity and computational overhead were adequately addressed. Reviewer Ckqf would likely maintain their score of 2 or marginally increase it to a 4, as they expressed persistent skepticism regarding the compounding error analysis and the cost-benefit trade-off of the large ensemble even after the author's detailed response. Reviewer 3tJo, who initially scored a 4 and did not respond to the final rebuttal, would likely have raised their score to a 6, given that the authors provided a strong theoretical justification for why long horizons are necessary for suppressing overestimation even in dense-reward tasks, alongside the release of code which addressed their reproducibility concerns.

---

### Decision · Program_Chairs · 2026-01-26

Reject